# FEDSHUFFLE: **Recipes for Better Use of Local Work in Federated Learning**

**Samuel Horváth**                                                          *samuel.horvath@mbzuai.ac.ae*
*MBZUAI**

**Maziar Sanjabi**                                                                  *maziars@fb.com*
*Meta AI*

**Lin Xiao**                                                                          *linx@fb.com*
*Meta AI*

**Peter Richtárik**                                                          *richtarik@gmail.com*
*KAUST*

**Michael Rabbat**                                                              *mikerabbat@fb.com*
*META AI*

**Reviewed on OpenReview:** *https://openreview.net/forum?id=Lgs5pQ1v30*

## Abstract

The practice of applying several local updates before aggregation across clients has been empirically shown to be a successful approach to overcoming the communication bottleneck in Federated Learning (FL). Such methods are usually implemented by having clients perform one or more epochs of local training per round, while randomly reshuffling their finite dataset in each epoch. Data imbalance, where clients have different numbers of local training samples, is ubiquitous in FL applications, resulting in different clients performing different numbers of local updates in each round. In this work, we propose a general recipe, FEDSHUFFLE, that better utilizes the local updates in FL, especially in this regime encompassing random reshuffling and heterogeneity. FEDSHUFFLE is the first local update method with theoretical convergence guarantees that incorporates random reshuffling, data imbalance, and client sampling — features that are essential in large-scale cross-device FL. We present a comprehensive theoretical analysis of FEDSHUFFLE and show, both theoretically and empirically, that it does not suffer from the objective function mismatch that is present in FL methods that assume homogeneous updates in heterogeneous FL setups, such as FEDAVG (McMahan et al., 2017). In addition, by combining the ingredients above, FEDSHUFFLE improves upon FEDNOVA (Wang et al., 2020), which was previously proposed to solve this mismatch. Similar to MIME (Karimireddy et al., 2020), we show that FEDSHUFFLE with momentum variance reduction (Cutkosky & Orabona, 2019) improves upon non-local methods under a Hessian similarity assumption.

## 1 Introduction

*Federated learning* (FL) aims to train models in a decentralized manner, preserving the privacy of client data by leveraging edge-device computational capabilities. Clients' data never leaves their devices; instead, the clients coordinate with a server to train a global model. Due to such advantages and promises, FL is now deployed in a variety of applications (Hard et al., 2018; Apple, 2019).

---

*Majority of work completed during an internship at Meta.

In this paper, we consider the standard FL problem formulation of solving an empirical risk minimization problem over the data available from all devices; i.e.,

$$\min_{x \in \mathbb{R}^d} \left[ f(x) \stackrel{\text{def}}{=} \sum_{i=1}^{n} w_i f_i(x) \right], \text{ where } \forall i \in \{1, \ldots, n\}: \ f_i \stackrel{\text{def}}{=} \frac{1}{|\mathcal{D}_i|} \sum_{j=1}^{|\mathcal{D}_i|} f_{ij}(x). \tag{1}$$

Here $f_{ij}$ corresponds to the loss of a model with parameters $x$ evaluated on the $j$-th data point of the $i$-th client. The weight $w_i$ assigned to device $i$'s empirical risk is $w_i = {}^{|\mathcal{D}_i|}/{}_{|\mathcal{D}|}$ where $|\mathcal{D}_i|$ is the size of the training dataset at device $i$ and $|\mathcal{D}| = \sum_{i=1}^{n} |\mathcal{D}_i|$. This choice of weights $w_i$ places equal weight on all training data.[1]

To solve (1), optimization methods must contend with several challenges that are unique to FL: *heterogeneity* with respect to data and compute capabilities of each device, *data imbalance* across devices, and *limited device availability*. Moreover, in cross-device FL, the number of participating devices $n$ can be on the order of millions. At this scale, *client sampling* (using a subset of clients for each update) is a necessity since it is impractical for all devices to participate in every round. Furthermore, each device may only participate once or a few times during the entire training process, so *stateless* methods (those which do not rely on each client maintaining and updating local state throughout training) are of particular interest.

The most widely studied and used methods in this challenging setting have devices perform multiple steps on their data locally, before communicating updates to the server (i.e., local update methods *a la* local SGD/ FedAvg) (Kairouz et al., 2019). Most existing analyses of local update methods assume that all participating clients perform the same number of local *steps* in each round, and that clients sample new, independent gradients at every local step. In contrast, most practical implementations, going back to the original description of FedAvg (McMahan et al., 2017), have devices perform one or more local *epochs* over their finite training dataset, while *randomly reshuffling* the data at each epoch. The number of training samples per device may vary by many orders of magnitude (Kairouz et al., 2019). Thus, performing local epochs results in different clients performing differing numbers of local steps per round.

Although it is now well-understood that random reshuffling has a variance-reducing effect in centralized training, provably obtaining this benefit in federated training has been challenging because of the dependence induced by random reshuffling. In the next section, we provide a detailed discussion on random reshuffling as a part of related work. Mishchenko et al. (2021), Yun et al. (2021) and Malinovsky et al. (2021) analyze random reshuffling in the context of FL, while assuming that all devices have the same number of training samples and all devices participate in every round (i.e., no client sampling). Previous work of Wang et al. (2020) identified that performing different numbers of local steps per device per round in FedAvg leads to the *objective inconsistency* problem — effectively minimizing a different objective than (1), where terms are reweighted by the number of samples per device. Wang et al. (2020) propose the FedNova method to address this by rescaling updates during aggregation. FedLin (Mitra et al., 2021) addresses objective inconsistency by scaling local learning rates, while assuming full participation (all devices participate in every round) which is not practical for cross-device FL. Neither FedNova nor FedLin incorporate reshuffling of device data. Improved convergence rates for FL optimization have been achieved by Karimireddy et al. (2020) by combining a Hessian similarity assumption with momentum variance reduction (Cutkosky & Orabona, 2019), but without accounting for random reshuffling or data imbalance.

In this paper we aim to provide a unified view of local update methods while accounting for random reshuffling, data imbalance (through non-identical numbers of local steps), and client subsampling. Furthermore, we obtain faster convergence by incorporating momentum variance reduction. Table 1 summarizes the key differences mentioned above.

**Contributions.** We make the following contributions in this paper:

- **New algorithm:** FedShuffle, **an improved way to remove objective inconsistency.** In Section 4 we introduce and analyze FedShuffle to account for random reshuffling, client sampling,

---

[1] Although we focus on the setting with weights $w_i = {}^{|\mathcal{D}_i|}/{}_{|\mathcal{D}|}$ so that the overall objective is equivalent to a standard, centralized empirical risk minimization problem using the data from all devices, this is not essential to our analysis, which could accommodate any choice of $w_i$. Of course, using a different choice of $w_i$ will change the solution.

Table 1: Comparison of characteristics considered in previous work and the methods analyzed in this paper. Notation: *HD* = Heterogeneous Data, *CS* = Client Sampling, *RR* = Random Reshuffling, *NL* = Non-identical Local Steps, *VR* = Variance Reduction, *GM* = Global Momentum, *SS* = Server Step Size. The bottom four methods are proposed and/or analyzed in this paper.

| | HD | CS | RR | NL | VR | GM | SS |
|---|---|---|---|---|---|---|---|
| FEDNOVA (Wang et al., 2020) | ✓ | ✓ | ✗ | ✓ | ✗ | ✗ | ✓ |
| FEDLIN (Mitra et al., 2021) | ✓ | ✗ | ✗ | ✓ | ✓ | ✗ | ✗ |
| MIME (Karimireddy et al., 2020) | ✓ | ✓ | ✗ | ✗ | ✓ | ✓ | ✗ |
| FEDRR (Mishchenko et al., 2021) / LOCALRR (Yun et al., 2021) | ✓ | ✗ | ✓ | ✗ | ✗ | ✗ | ✗ |
| *(Contributions in this work)* | | | | | | | |
| FEDAVGRR | ✓ | ✓ | ✓ | ✗* | ✗ | ✗ | ✓ |
| FEDNOVARR | ✓ | ✓ | ✓ | ✓ | ✗ | ✗ | ✓ |
| FEDSHUFFLE | ✓ | ✓ | ✓ | ✓ | ✗ | ✗ | ✓ |
| FEDSHUFFLEMVR | ✓ | ✓ | ✓ | ✓ | ✓ | ✓ | ✓ |

*With NL, FEDAVGRR optimizes the wrong objective.

and address the objective inconsistency problem. FEDSHUFFLE fixes the objective inconsistency problem by adjusting the local step size for each client and redesigning the aggregation step, enabling a larger theoretical step size than either FEDAVG or FEDNOVA, while also benefiting from lower variance from random reshuffling.

- **New algorithm: FEDSHUFFLEMVR, beating non-local methods.** In Section 5.1 we extend the results of (Karimireddy et al., 2020) by accounting for random reshuffling and data imbalance. Under a Hessian similarity assumption, we show that incorporating momentum variance reduction (MVR) with FEDSHUFFLE leads to better convergence rates than the lower bounds for methods that do not use local updates.

- **General framework: FedShuffleGen.** The above results are obtained by first considering a general framework (see Algorithm 4 in Appendix E.4) that accounts for data heterogeneity, different numbers of local updates per device, arbitrary client sampling, different local learning rates, and heterogeneity in the aggregation weights. To the best of our knowledge, this work is the first to tackle the challenge of random reshuffling in FL at this level of generality. Similar to Wang et al. (2020), our analysis reveals how heterogeneity in the number of local updates can lead to objective inconsistency. Within this framework, we obtain the first analysis of FEDAVG[2] and FEDNOVA with random reshuffling (FEDAVGRR and FEDNOVARR respectively in Table 1).

- **Theoretical analysis.** To our knowledge, we are the first to analyze a very general setup, where we consider all the standard components that are commonly used in practical implementations of FL. Furthermore, our results are tight compared to the best-known guarantees for components analyzed in isolation. The main challenge of our analysis comes especially from combining biased random reshuffling (RR) with other techniques. On top of that, our analysis is simpler when compared to the state-of-the-art analysis of FEDAVG (Karimireddy et al., 2019a) as it does not require to upper bound local client drift using recursive estimates. Finally, our general variance bound (see Appendix E.4) is the first result that allows the incorporation of non-deterministic aggregation rules based on client sampling. To our knowledge, such results are impossible to obtain with any previously known analysis, despite this being standard practice for FEDAVG, where the update of each client is scaled by $w_i/(\sum_{j \in S} w_j)$, e.g., the default way to aggregate in Tensorflow Federated[3] and other frameworks.

- **Experiments.** Finally, our theoretical results and insights are corroborated by experiments, both in a controlled setting with synthetic data, and using commonly-used real datasets for benchmarking and comparison with other methods from the literature.

---

[2]We acknowledge the earlier work of Malinovsky et al. (2021), which analyzes random reshuffling of local data in the context of federated learning, with a specific focus on the role of client and server stepsizes. Our main results were obtained independently in early Fall 2021, and at that time we also learned about the results of Malinovsky et al. (2021) through personal communication, which were obtained somewhat earlier, but were not available online at that time.

[3]https://www.tensorflow.org/federated

## 2 Related Work

**Federated optimization and local update methods.** As we mentioned earlier local update methods are at the heart of FL. As a result, many prior works have analyzed various aspects of local update methods, e.g. (Wang & Joshi, 2021; Stich, 2018; Zhou & Cong, 2017; Yu et al., 2019; Li et al., 2019; Haddadpour & Mahdavi, 2019; Haddadpour et al., 2019a;b; Khaled et al., 2020; Stich & Karimireddy, 2020; Wang & Joshi, 2019; Woodworth et al., 2020; Koloskova et al., 2020; Khaled et al., 2019; Woodworth et al., 2018; Xie et al., 2019; Lin et al., 2018). These analyses are done under the assumption that every client performs the same number of local updates in each round. However this is usually not the case in practice, due to the heterogeneity of the data and system in FL; note that practical FL algorithms run a fixed number of *epochs* (not steps) per device. Moreover, forcing the fast and slow devices to run the same number of iterations would slow-down the training. This problem was also noted by (Wang et al., 2020), where it is shown that having a heterogeneous number of updates, which is inevitable in FL, leads to an inconsistency between the target loss (1) and the loss that the methods optimize. Moreover, as shown in (Wang et al., 2020), other approaches such as FEDPROX (Li et al., 2020), VRLSGD (Liang et al., 2019) and SCAFFOLD (Karimireddy et al., 2019a) that are designed for heterogeneous data can partially alleviate the problem but not completely eliminate it.

In this work, we propose FEDSHUFFLE, a method that combines update weighting and learning rate adjustments to deal with this issue. Our approach is more general than FEDNOVA (Wang et al., 2020), which only uses update weighting,[4] and we show both analytically and experimentally that it outperforms FEDNOVA and does not slow down the convergence of FEDAVG.

**Random reshuffling.** A particularly successful technique to optimize the empirical risk minimization objective is randomly permute (i.e., reshuffle) the training data at the beginning of every epoch (Bottou, 2012) instead of randomly sampling a data point (or a subset of data points) with replacement at each step, as in the standard analysis of SGD. This process is repeated several times and the resulting method is usually referred to as Random Reshuffling (RR). RR is often observed to exhibit faster convergence than sampling with replacement, which can be intuitively attributed to the fact that RR is guaranteed to process each training sample exactly once every epoch, while with-replacement sampling needs more steps than the equivalent of one epoch to see every sample with high probability. Properly understanding the random reshuffling trick, and why it works, has been a challenging open problem (Bottou, 2009; Ahn & Sra, 2020; Gürbüzbalaban et al., 2021) until recent advances in Mishchenko et al. (2020) introduced a significant simplification of the convergence analysis technique.

The difficulty of analysing RR stems from the fact that step-to-step dependence results in biased gradient estimates, unlike in with-replacement sampling. Apart from this, RR in FL involves an additional challenge: imbalance in number of samples that leads to the heterogeneity in number of local updates. To the best of our knowledge, analyzing RR in FL and local update methods remains largely unexplored in the literature despite RR being the default implementation used in simulations and practical deployments of FL; e.g., it is a default option in TensorFlow Federated.

We are only aware of two previous papers analyzing RR for FL (Mishchenko et al., 2021; Yun et al., 2021) and both of these works rely on two assumptions that are usually violated in cross-device FL: (i) that all clients participate in every round, and (ii) that all clients have the same number of training samples. In addition, Mishchenko et al. (2021) only analyze the (strongly) convex setting and Yun et al. (2021) require the Polyak-Łoyasiewicz condition to hold for the global function. In this work, building on shoulders of the recent advances (Mishchenko et al., 2020), we address all the challenges that come from applying RR for FL.

## 3 Notation and Assumptions

Firstly, recall that the portion of the loss function that belongs to client $i$ is composed of single losses $f_{ij}(x)$, where $j$ corresponds to $j$-th data point, and $x$ is a parameter we aim to optimize. We assume that client $i$

---

[4]We note that the analysis of Wang et al. (2020) could potentially accommodate clients using different learning rates (in their notation, balancing $\{\|a_i\|_1\}$ instead of aggregation weights). Still, this is not an obvious extension of the FEDNOVA analysis and it has been neither considered nor analyzed in theory or practice in previous work.

has access to an oracle that takes $(j, x)$ as an input and returns the gradient $\nabla f_{ij}(x)$ as an output. In order to provide convergence guarantees, we make the following standard assumptions and will discuss how they relate to other commonly used assumptions in the literature. We provide convergence guarantees for three common classes of smooth objectives: strongly-convex, general convex, and non-convex.

**Assumption 3.1.** The functions $\{f_{ij}\}$ are $\mu$-**convex** for $\mu \geq 0$; i.e., for any $i, x, y$

$$\langle \nabla f_{ij}(x), y - x \rangle \leq -\left( f_{ij}(x) - f_{ij}(y) + \frac{\mu}{2} \|x - y\|^2 \right). \tag{2}$$

We say that $f_{ij}$ is $\mu$-strongly convex if $\mu > 0$, and otherwise $f_{ij}$ is (general) convex.

**Assumption 3.2.** The functions $\{f_{ij}\}$ are $L$-**smooth**; i.e., there is an $L > 0$ such that for any $i, j, x, y$

$$\|\nabla f_{ij}(x) - \nabla f_{ij}(y)\| \leq L\|x - y\|. \tag{3}$$

Next, we state two standard assumptions which quantify heterogeneity. The first bounds the gradient dissimilarity among local functions $\{f_i\}$ at different clients, and the second controls the variance of local gradients at each client. The same or more restrictive versions of these assumptions appeared in (Karimireddy et al., 2019b; 2020; Wang et al., 2020; Mitra et al., 2021).

**Assumption 3.3** (Gradient Similarity)**.** The local gradients $\{\nabla f_i\}$ are $(G, B)$-**bounded**, i.e., for all $x \in \mathbb{R}^d$,

$$\sum_{i=1}^{n} w_i \|\nabla f_i(x)\|^2 \leq G^2 + B^2 \|\nabla f(x)\|^2. \tag{4}$$

If $\{f_i\}$ are convex, then we can relax the assumption to

$$\sum_{i=1}^{n} w_i \|\nabla f_i(x)\|^2 \leq G^2 + 2LB^2(f(x) - f^\star). \tag{5}$$

**Assumption 3.4** (Bounded Variance)**.** The local stochastic gradients $\{\nabla f_{ij}\}$ have $(\sigma_i, P_i)$-**bounded variance**, i.e., for all $x \in \mathbb{R}^d$,

$$\frac{1}{|\mathcal{D}_i|} \sum_{j=1}^{|\mathcal{D}_i|} \|\nabla f_{ij}(x) - \nabla f_i(x)\|^2 \leq \sigma_i^2 + P_i^2 \|\nabla f_i(x)\|^2. \tag{6}$$

Note that we do not require gradient norms to be bounded by constant. Moreover, we do not require the global or local variance to be bounded by constants either, but we allow them to be proportional to the gradient norms. In stochastic optimization, these assumptions are referred to as relaxed growth condition (Bottou et al., 2018). Furthermore, one can show that for smooth and convex $\{f_{ij}\}$, these are not actually assumptions, but rather properties (Stich, 2019). While for non-convex functions, these are critical assumptions to show convergence under partial participation.

Following (Karimireddy et al., 2020), we also characterize the variance in the Hessian. This is an important assumption that helps us to understand and showcase the benefits of local steps.

**Assumption 3.5** (Hessian Similarity)**.** The local gradients $\{\nabla f_i\}$ have $\delta$-**Hessian similarity**, i.e., for all $x \in \mathbb{R}^d$ and $i \in [n]$, ($\|\cdot\|$ represents spectral norm for matrices)

$$\|\nabla^2 f_i(x) - \nabla^2 f(x)\|^2 \leq \delta^2. \tag{7}$$

Note that if $\{f_i\}$ are $L$-smooth then it must hold that $\|\nabla^2 f_i(x)\| \leq L$ for all for all $x \in \mathbb{R}^d$ and $i \in [n]$ and, therefore, Assumption 3.5 is satisfied with $\delta \leq 2L$. In realistic examples, one might expect the clients to be similar and hence it could happen that $\delta \ll L$.

We work with a fixed *arbitrary participation framework* (Horváth & Richtárik, 2020), where one assumes that the subset of participating clients is determined by an arbitrary random set-valued mapping $\mathcal{S}$ (a "sampling")

---

**Algorithm 1** FEDSHUFFLE

---

1: **Input:** initial global model $x^0$, global and local step sizes $\eta_g^r$, $\eta_l^r$, proper distribution $\mathcal{S}$
2: **for** each round $r = 0, \ldots, R - 1$ **do**
3:     server broadcasts $x$ to all clients $i \in \mathcal{S}^r \sim S$
4:     **for** each client $i \in \mathcal{S}^r$ (in parallel) **do**
5:         initialize local model $y_i \leftarrow x$
6:         **for** $e = 1, \ldots, E$ **do**
7:             Sample permutation $\{\Pi_0, \ldots, \Pi_{|\mathcal{D}_i|-1}\}$ of $\{1, \ldots, |\mathcal{D}_i|\}$
8:             **for** $j = 1, \ldots, |\mathcal{D}_i|$ **do**
9:                 update $y_i \leftarrow y_i - \frac{\eta_l^r}{|\mathcal{D}_i|}\nabla f_{i\Pi_{j-1}}(y_i)$
10:            **end for**
11:        **end for**
12:        send $\Delta_i = y_i - x$ to server
13:    **end for**
14:    server computes $\Delta = \sum_{i \in \mathcal{S}^r} \frac{w_i}{p_i}\Delta_i$
15:    server updates global model $x \leftarrow x - \eta_g^r \Delta$
16: **end for**

---

with values in $2^{[n]}$. A sampling $\mathcal{S}$ is uniquely defined by assigning probabilities to all $2^n$ subsets of $[n]$. With each sampling $\mathcal{S}$ we associate a *probability matrix* $\mathbf{P} \in \mathbb{R}^{n \times n}$ defined by $\mathbf{P}_{ij} \overset{\text{def}}{=} \Pr[\{i, j\} \subseteq \mathcal{S}]$. The *probability vector* associated with $\mathcal{S}$ is the vector composed of the diagonal entries of $\mathbf{P}$: $p = (p_1, \ldots, p_n) \in \mathbb{R}^n$, where $p_i \overset{\text{def}}{=} \Pr[i \in \mathcal{S}]$. We say that $\mathcal{S}$ is *proper* if $p_i > 0$ for all $i$. It is easy to show that $b \overset{\text{def}}{=} \mathrm{E}[|\mathcal{S}|] = \mathrm{Trace}(\mathbf{P}) = \sum_{i=1}^n p_i$, and hence $b$ can be seen as the expected number of clients participating in each communication round. We associate every proper sampling with a vector $s = [s_1, \ldots, s_n]^\top$ for which it holds

$$\mathbf{P} - pp^\top \preceq \mathbf{Diag}(p_1 s_1, p_2 s_2, \ldots, p_n s_n), \tag{8}$$

which is a quantity that appears in the convergence rate. For instance, one can show that uniform sampling with $b$ participating clients admits $s_i = (n-b)/(n-1)$ and full participation allows to set $s_i = 0$ as $\mathbf{P}$ is all ones matrix, see (Horváth & Richtárik, 2019) for details. Finally, we note that a fixed *arbitrary participation framework* is only for an ease of exposition and our framework can handle non-fixed distributions with minimal adjustments in the analysis.

## 4 The FEDSHUFFLE Algorithm

We now formally introduce our FEDSHUFFLE method. Its pseudocode is provided in Algorithm 1 (simple) and Algorithm 3 (precise). The main inspiration for FEDSHUFFLE is the default optimization strategy used in Federated Learning: FEDAVG. As described in McMahan et al. (2017), in FEDAVG one starts each communication round by sampling $b$ clients uniformly at random to participate. These clients then receive the global model from the server and update it by training the model for $E$ epochs on their local data. The model updates are communicated back to the server, which aggregates them and updates the global model before proceeding to the next round. We provide the pseudocode for this procedure in Algorithm 2 in the Appendix.

Unfortunately, we show that FEDAVG (as implemented in practice) does not converge to the exact solution due to inconsistency caused by unbalanced local steps and biased aggregation. We discuss each of these issues in details in Sections 4.1 and 4.2, respectively. Therefore, we propose a new algorithm–FEDSHUFFLE, to address these limitations of local methods. It involves two modifications compared to FEDAVG: we scale the local step size by $\frac{1}{|\mathcal{D}_i|}$, and we also adjust the aggregation step. In addition, our analysis allows each client to run different number of epochs $\{E_i\}$, in that case, the local step size is scaled proportionally to $\frac{1}{E_i|\mathcal{D}_i|}$; see Section E in the Appendix.

### 4.1 Heterogeneity in the Number of Local Updates

We introduce the first adjustment: step size scaling. We consider the same example as Wang et al. (2020), the quadratic minimization problem

$$\min_{x \in \mathbb{R}^d} \frac{1}{|\mathcal{D}|} \sum_{i=1}^{|\mathcal{D}|} \|x - e_i\|^2, \tag{9}$$

where $\{e_i\}_{i=1}^{|\mathcal{D}|}$ are given vectors. Clearly, this is a strongly convex objective with the unique minimizer $x^\star = \frac{1}{|\mathcal{D}|} \sum_{i=1}^{|\mathcal{D}|} e_i$. For simplicity, let us assume that we solve this objective using standard FEDAVG with local shuffling and full client participation, i.e., $b = |\mathcal{D}|$. Since each local function has only one element, this is equivalent to running Gradient Descent (GD), and therefore for small enough step size this algorithm converges linearly to the optimal solution $x^\star$. Now, suppose instead that only $\{e_i\}_{i=1}^n$ are unique and each client $i$ has $|\mathcal{D}_i|$ copies of $e_i$ locally. Then, we can write the objective as

$$\min_{x \in \mathbb{R}^d} \sum_{i=1}^n \frac{|\mathcal{D}_i|}{|\mathcal{D}|} f_i(x), \quad \text{where} \quad f_i(x) \stackrel{\text{def}}{=} \frac{1}{|\mathcal{D}_i|} \sum_{j=1}^{|\mathcal{D}_i|} \|x - e_i\|^2. \tag{10}$$

Applying FEDAVG with local shuffling is equivalent to running FEDAVG with $E|\mathcal{D}_i|$ local steps since all the local data are the same. Similarly to Wang et al. (2020), we show that FEDAVG with unbalanced $E|\mathcal{D}_i|$ local steps introduces bias/inconsistency is the optimized objective and converges linearly to the sub-optimal solution $\tilde{x} = (1/\sum_{i=1}^n |\mathcal{D}_i|^2) \sum_{i=1}^n |\mathcal{D}_i|^2 e_i$ for sufficiently small step size $\eta_l$ (this statement is a direct consequence of Theorem E.1 that can be found in Appendix E). We note that one can choose $\{|\mathcal{D}_i|\}$ and $\{e_i\}$ arbitrarily; thus, the difference between $\tilde{x}$ and $x^\star$ can be arbitrary large. To tackle this first issue that causes the objective inconsistency, we propose to scale the step size proportionally to $1/|\mathcal{D}_i|$, which removes the aforementioned inconsistency.

In fact, Appendix E contains more general results. We introduce and analyze a general shuffling algorithm—FEDSHUFFLEGEN, that encapsulates FEDAVG, FEDNOVA and our FEDSHUFFLE as special cases due to its general parametrization by local and global step sizes, step size normalization, aggregation weights and the aggregation normalization constants; see Algorithm 4 in the appendix. As a byproduct, we obtain a detailed theoretical comparison of FEDNOVA and FEDSHUFFLE. In a nutshell, we show that FEDSHUFFLE balances the progress made by each client and keeps the aggregation weights unaffected while FEDNOVA diminishes the weights for the client that makes the most progress. As a consequence, FEDSHUFFLE allows larger theoretical local step sizes than both FEDAVG and FEDNOVA while preserving the worst-case convergence rate. We refer the reader to Appendix E, particularly Section E.2, for the extended discussion and a detailed comparison of all three methods.

Lastly, one might fix the inconsistencies in the FEDAVG by running the same number of local steps $K$ at each client. Note that universally choosing a fixed number of steps for all clients is not straightforward. We will compare heuristics based on a fixed number $K$ of steps with our proposed approaches in the experiments. To be comparable to other baselines, we use two heuristics to select $K$ : (1) Set $K$ based on the client with minimum number of data points in the round (FEDAVGMIN), which ensures that such a round will not result in any additional stragglers compared to other baselines. As we will see, FEDAVGMIN does not result in great performance as it does not utilize all the data on most of the clients. (2) Set $K$ to be the average number of steps that the selected clients would have taken in that round if they were running other baselines (FEDAVGMEAN). This makes sure that the total number of local steps for all clients is the same across all baselines. Note that FEDAVGMIN and FEDAVGMEAN are not practical since they require additional coordination among the selected clients to determine the number of local steps to take; we consider them as a heuristic to show the difficulty of choosing a fixed number of local steps for all clients. As we will see, FEDAVGMEAN under-performs FEDSHUFFLE and even in some cases FEDAVG, especially in terms of test accuracy in heterogeneous settings.

### 4.2 Removing Bias in Aggregation

The second algorithmic change compared to FEDAVG has been, to the best of our knowledge, overlooked and it is related to the aggregation step. The original aggregation that is widely used in practice, see Algorithm 2

for FEDAVG practical implementation, contains the step (line 15) where the local weights from the client $i \in \mathcal{S}$ are normalized to sum to one by $w_i / \sum_{j \in \mathcal{S}} w_j$; we refer to this as the *Sum One (SO)* aggregation. Such aggregation can lead to a biased contribution from workers and therefore to an inconsistent solution that optimizes a different objective as we show in the following example.

Suppose that there are three clients and they hold, respectively, 1, 2 and 3 data points. In each round, we sample two clients uniformly at random. Then, the expected contribution from client $i$ is $\mathbf{E}_i [w_i / \Delta_i]$, where $\Delta_i = \sum_{j \in \mathcal{S} \text{ s.t. } i \in \mathcal{S}} w_j$. It is easy to verify that this is equal to $7/36$, $16/45$ and $9/20$, respectively. One can note that this is not proportional to the weights $\{w_i\}$ of the objective (1). Furthermore, this proposed aggregation cannot be simply fixed by changing the client sampling scheme, e.g., by sampling clients with probability proportional to the number of examples they hold, since one can always find a simple counterexample. The problem of the aggregation scheme is the sample dependent normalization $\sum_{i \in \mathcal{S}} w_i$ that makes sampling biased in the presence of non-uniformity with respect to the number of data samples per client. To solve this issue, we use $w_i / p_i$ in the scaling step, where $p_i$ is the probability that client $i$ is selected. This a very standard aggregation scheme (Wang et al., 2018; Wangni et al., 2018; Horváth & Richtárik, 2019) that results in unbiased aggregation with respect to the worker contribution since $\mathbf{E}_i [w_i / p_i] = w_i$. It is easy to see that if $\{p_i\}$ are proportional to $\{w_i\}$ then the aggregation step would be simply taking a sum. This can be achieved by each client being sampled independently using a probability proportional to its weight $w_i$, i.e., its dataset size if the central server has access to this information.[5] If not all clients are available at all times, one can use Approximate Independent Sampling (Horváth & Richtárik, 2019) that leads to the same effect.

### 4.3 Extensions

As mentioned previously, we introduce FEDSHUFFLEGEN (Algorithm 4) in Appendix E which encapsulates FEDAVG, FEDNOVA and FEDSHUFFLE as special cases and unifies the convergence analysis of these three methods. As an advantage, we use this unified framework to show that it is better to handle objective inconsistency by scaling the step sizes rather than scaling the updates, i.e., it is better to run FEDSHUFFLE rather than FEDNOVA as FEDSHUFFLE allows for larger theoretical step sizes, see Remark E.2 for details.

In addition, our general analysis allows for different extensions such as each client running different arbitrary number of local epochs. FEDSHUFFLEGEN also allows us to run and analyze hybrid approaches of mixing step size scaling with update scaling to overcome the objective inconsistency. These hybrid approaches would be efficient when applying step size scaling only, i.e. FEDSHUFFLE, might not overcome objective inconsistency due to system challenges. For example, such a scenario could happen when some clients cannot finish their predefined number of epochs due to a time-out, e.g., large variance in computing time, random drop-off, or interruption during local training. In such scenarios, FEDSHUFFLEGEN allows additional adjustments through update scaling.

## 5 Convergence guarantees

In the theorem below, we establish the convergence guarantees for Algorithm 1. Before proceeding with the theorem, we define several quantities derived from the constants that appear in Assumptions 3.3 and 3.4

$$M \stackrel{\text{def}}{=} \max_{i \in [n]} \left\{ \frac{s_i}{p_i} w_i \right\}, \quad P^2 \stackrel{\text{def}}{=} \max_{i \in [n]} \frac{P_i^2}{|\mathcal{D}_i|}, \quad \sigma^2 \stackrel{\text{def}}{=} \frac{1}{|\mathcal{D}|} \sum_{i \in [n]} \sigma_i^2, \quad \beta \stackrel{\text{def}}{=} 1 + (1+P)B + MB^2,$$

and the ones that reflect the quality of the initial solution $D \stackrel{\text{def}}{=} \|x^0 - x^\star\|^2$ and $F \stackrel{\text{def}}{=} f(x^0) - f^\star$.

**Theorem 5.1.** *Suppose that the Assumptions 3.2-3.4 hold. Then, in each of the following cases, there exist weights $\{v_r\}$, local step sizes $\eta_l^r \stackrel{\text{def}}{=} \eta_l$ and effective step sizes $\tilde{\eta}^r \stackrel{\text{def}}{=} \tilde{\eta} = E\eta_g\eta_l$ such that for any $\eta_g^r \stackrel{\text{def}}{=} \eta_g \geq 1$ the output of FEDSHUFFLE (Algorithm 1)*

$$\bar{x}^R = x^r \quad \text{with probability} \quad \frac{v_r}{\sum_\tau v_\tau} \quad \text{for} \quad r \in \{0, \dots, R-1\} \tag{11}$$

---

[5]It may not be possible for the server to know the number of samples per client because of privacy constraints, in which case one can always default a uniform sampling scheme with $p_i = 1/n$.

*satisfies*

- **Strongly convex:** $\{f_{ij}\}$ *satisfy* (2) *for* $\mu > 0$, $\tilde{\eta} \leq \frac{1}{4\beta L}$, $R \geq \frac{4\beta L}{\mu}$ *then*

$$\mathbf{E}\left[f(\bar{x}^R) - f(x^\star)\right] \leq \tilde{\mathcal{O}}\left(\frac{MG^2}{\mu R} + \frac{(E^2 + P^2)G^2 + \sigma^2}{\mu^2 R^2 \eta_g^2 E^2} + \mu D^2 \exp\left(-\frac{\mu}{8\beta L}R\right)\right),$$

- **General convex:** $\{f_{ij}\}$ *satisfy* (2) *for* $\mu = 0$,

$$\mathbf{E}\left[f(\bar{x}^R) - f(x^\star)\right] \leq \mathcal{O}\left(\frac{\sqrt{DM}G}{\sqrt{R}} + \frac{D^{2/3}((E^2 + P^2)G^2 + \sigma^2)^{1/3}}{R^{2/3}\eta_g^{2/3}E^{2/3}} + \frac{LD\beta}{R}\right),$$

- **Non-convex:** $\tilde{\eta} \leq \frac{1}{4\beta L}$, *then*

$$\mathbf{E}\left[\|\nabla f(\bar{x}^R)\|^2\right] \leq \mathcal{O}\left(\frac{\sqrt{FML}G}{\sqrt{R}} + \frac{F^{2/3}L^{1/3}((E^2 + P^2)G^2 + \sigma^2)^{1/3}}{R^{2/3}\eta_g^{2/3}E^{2/3}} + \frac{LF\beta}{R}\right).$$

Let us discuss the obtained rates. First, note that for a sufficiently large number of communication rounds, the first term is the leading term. This term together with the last term correspond to the rate of Distributed GD with partial participation, where each sampled client returns its gradient as the update. If each client participates then $M = 0$ and the first term vanishes. The second term comes from local steps using random reshuffling. Note here that the dependency of the noise term $\sigma^2$ on the number of communication rounds is $R^2$ and $R^{2/3}$, respectively, while for local steps with unbiased stochastic gradients, this would be $R$ and $R^{1/3}$. This shows that the variance is decreased when one employs random reshuffling instead of with-replacement sampling. We further note that the middle term can be completely removed in the limit where $\eta_g \to \infty$, and the local variance $\sigma^2$ vanishes when $E \to \infty$. We note that such property was not observed for FEDNOVA. The limit $\eta_g \to \infty$ implies $\eta_l \to 0$ and thus FEDSHUFFLE reduces to GD with partial participation. To analyze the effect of the cohort size $b$ (number of sampled clients) on the convergence rate, we look at the special case where each client is sampled independently with probability $p_i = bw_i$ (assume $bw_i \leq 1$ for simplicity) for all $i \in [n]$. We refer to this sampling as *importance sampling* as it is easy to see that the $M$ term is minimized for this sampling (Horváth & Richtárik, 2019). In this particular case, $M = {}^{(1-\min\{w_i\})}/b$ and, thus, we obtain theoretical linear speed with respect to the expected cohort size $b$.

Lastly, we note that the obtained rates do not asymptotically improve upon distributed GD with partial participation, but this is the case for every local method with local steps based only on the local dataset, i.e., no global information is exploited.

## 5.1 Improving upon Non-Local Methods

Contrary to the relatively negative worst-case results presented in the previous section, local methods have been observed to perform significantly better in practice (McMahan et al., 2017) when compared to non-local (i.e., one local step) methods. To overcome this issue, Karimireddy et al. (2019b) proposed to use a Hessian similarity assumption (Arjevani & Shamir, 2015), and they showed that local steps bring improvement when the objective is *quadratic*, *all clients* participate in each round and the local steps are corrected using SAGA-like *variance reduction* (Defazio et al., 2014). Later, Karimireddy et al. (2020) proposed MIMEMVR that uses the Momentum Variance Reduction (MVR) technique (Cutkosky & Orabona, 2019; Tran-Dinh et al., 2019) and extended the prior results to smooth non-convex functions with uniform partial participation. In our work, we build upon these results and show that FEDSHUFFLE can also improve in terms of communication rounds complexity. To achieve this, we introduce FEDSHUFFLEMVR, a FEDSHUFFLE type algorithm that is extended with MIMEMVR's momentum technique. Each local update of FEDSHUFFLEMVR has the following form

$$y_{i,e,j}^r = y_{i,e,j-1}^r - \frac{\eta_l^r}{|\mathcal{D}_i|}d_{i,e,j-1}, \tag{12}$$

where

$$d_{i,e,j} = a\nabla f_{i\Pi^r_{i,e,j}}(y^r_{i,e,j}) + (1-a)m^r + (1-a)\Big(\nabla f_{i\Pi^r_{i,e,j}}(y^r_{i,e,j}) - \nabla f_{i\Pi^r_{i,e,j}}(x^r)\Big) \tag{13}$$

where the momentum term $m^r$ is updated at the beginning of each communication round as

$$m^r = a\sum_{i\in\mathcal{S}^r}\frac{w_i}{p_i}\nabla f_i(x^r) + (1-a)m^{r-1} + (1-a)\left(\sum_{i\in\mathcal{S}^r}\frac{w_i}{p_i}(\nabla f_i(x^r) - \nabla f_i(x^{r-1}))\right). \tag{14}$$

For the notation details, we refer the reader to Algorithm 3 in the Appendix. The above equations can be seen as the standard momentum (first two terms) with an extra correction term (the last term). For a detailed explanation about the motivation behind this momentum technique, we refer the reader to Cutkosky & Orabona (2019). Note that the momentum term is only updated *once* in each communication round, this is to reduce the local drift as proposed by Karimireddy et al. (2020). A convergence guarantee of FEDSHUFFLEMVR in the non-convex regime follows.

**Theorem 5.2.** *Let us run* FEDSHUFFLEMVR *with step sizes* $\eta_l = \frac{1}{40E}\min\left\{\frac{1}{\delta}, \left(\frac{f(x^0)-f^\star}{R\delta^2(G^2+\sigma^2)}\right)^{1/3}\right\}$, $\eta_g = 1$, *momentum parameter* $a = \max\big(1152E^2\delta^2\eta_l^2, \frac{1}{R}\big)$, *and local epochs* $E \geq \frac{L}{\delta}$. *Then, given that Assumptions 3.2-3.5 hold with* $P_i = 0$ *for all* $i \in [n]$, $B = 1$, $\delta > 0$ *and one client is sampled with probabilities* $\{w_i\}$, *we have*

$$\frac{1}{RE}\sum_{r=0}^{R-1}\sum_{e=0}^{E-1}\mathbf{E}\left[\left\|\nabla f(y^r_{i^r,e})\right\|^2\right] \leq \mathcal{O}\left(\frac{\delta^{2/3}F^{2/3}(G^2+\sigma^2)^{1/3}}{R^{2/3}} + \frac{\delta F + G^2}{R}\right).$$

Note that our rate is independent of $L$ and only depends on the Hessian similarity constant $\delta$. Because $\delta \leq L$, this rate improves upon the rate of the centralized MVR $\mathcal{O}(L^{2/3}/R^{2/3})$. We note that the improvement is only for the number of communication rounds, and the number of gradient calls is at least the same as for the non-local centralized MVR since $E\delta \leq L$, but our main concern here is the communication efficiency. It is worth noting that our results are qualitatively similar to those provided by MIMEMVR Karimireddy et al. (2020), but in our work, we consider a more challenging and practical setting. Namely, FEDSHUFFLEMVR does not require that all clients perform the same number of local updates, and each client runs local epochs using random reshuffling. Therefore, we work with biased gradients, and, in addition, we allow for heterogeneity in the number of samples per client, which brings another challenge that needs to be adequately addressed in the analysis to avoid objective inconsistency. Lastly, we note that our step size scaling is essential in the provided FEDSHUFFLEMVR convergence theory as it requires balancing the progress made by each client. Therefore, it is not clear whether a combination of FEDNOVA and MVR can lead to a similar improvement.

## 6 Experimental Evaluation

For the experimental evaluation, we compare three methods — FEDAVG, FEDNOVA, and our FEDSHUFFLE— with different extensions such as random reshuffling or momentum. We run two sets of experiments. In the first, we perform an ablation study on a simple distributed quadratic problem to verify the improvements predicted by our theory. In the second part, we compare all of the methods for training deep neural networks on the CIFAR100 and Shakespeare datasets. Details of the experimental setup can be found in Appendix F. As expected, our findings are that FEDSHUFFLE consistently outperforms other baselines. Moreover, global momentum leads to an improved performance of all methods as predicted by our theory.

### 6.1 Results on quadratic functions

In this section, we verify our theoretical findings on a convex quadratic objective; see (36) in the appendix for details. Figure 1 summarizes the results. The left-most plot showcases the comparison of FEDAVG, FEDAVG with reshuffling (FEDAVGRR), FEDNOVA as analyzed in (Wang et al., 2020) (with sampling), FEDNOVA

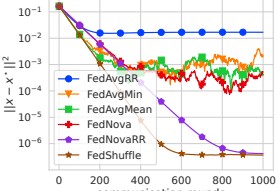 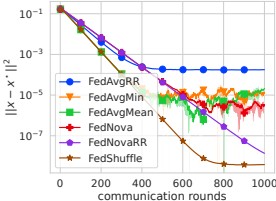 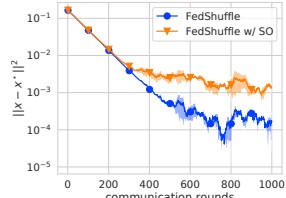 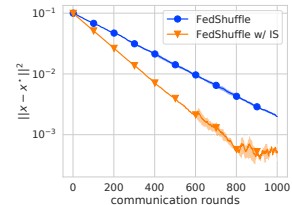

Figure 1: Quadratic objective as defined in (36). Each client runs one local epoch. Left: A comparison of FEDAVG, FEDAVG with reshuffling (FEDAVGRR), FEDNOVA and FEDNOVA with reshuffling (FEDNOVA RR) and FEDSHUFFLE with full participation. Middle Left: the same baselines with the global momentum 0.9. Middle Right: FEDSHUFFLE w/ Sum One and plain FEDSHUFFLE with partial participation (two clients sampled uniformly at random). Right: FEDSHUFFLE with uniform sampling and importance sampling (IS) with partial participation (one client per round).

Table 2: Shakespeare dataset.

| Accuracy | No Momentum | With Momentum |
|---|---|---|
| FEDAVGMIN | $35.79 \pm 0.28$ | $51.13 \pm 0.27$ |
| FEDAVGMEAN | $36.95 \pm 0.55$ | $55.38 \pm 0.27$ |
| FEDAVG | $48.61 \pm 0.56$ | $64.64 \pm 0.10$ |
| FEDNOVA | $43.29 \pm 0.40$ | $61.65 \pm 0.06$ |
| FEDSHUFFLE | $\mathbf{59.57 \pm 0.14}$ | $\mathbf{67.63 \pm 0.35}$ |

Table 3: CIFAR100 dataset.

| Accuracy | No Momentum | With Momentum |
|---|---|---|
| FEDAVGMIN | $48.49 \pm 0.28$ | $63.62 \pm 0.20$ |
| FEDAVGMEAN | $49.24 \pm 0.92$ | $\mathbf{64.75 \pm 0.20}$ |
| FEDAVG | $50.29 \pm 0.31$ | $63.04 \pm 0.51$ |
| FEDNOVA | $50.63 \pm 0.66$ | $62.83 \pm 0.19$ |
| FEDSHUFFLE | $\mathbf{51.97 \pm 0.20}$ | $\mathbf{64.52 \pm 0.48}$ |

with reshuffling (FEDNOVARR) as we analyze it in Appendix E, and FEDSHUFFLE. All clients participate in each round and run one local epoch with batch size 1.

As expected, FEDAVGRR saturates at a higher loss, since it optimizes the wrong objective, see Appendix E. FEDAVGMIN fixes the objective inconsistency of the FEDAVG, but it is still dominated by other baselines due to decreased amount of local work per client. FEDNOVA provides a better performance since it does not contain any inconsistency by construction, but its performance is later dominated by noise coming from stochastic gradients. The same holds for FEDAVGMEAN. As predicted by our theory, random reshuffling decreases the stochastic noise and FEDNOVARR improves upon the performance of FEDNOVA. FEDSHUFFLE dominates all the baselines since it does not have any objective inconsistency, it uses a superior method to remove inconsistencies compared to FEDNOVA, and it also incorporates random reshuffling which itself provides some variance reduction.

In the second plot, we use the same baselines but include global momentum as defined in (13) and (14). We can see that this technique helps FEDAVG to reduce its objective inconsistency since the momentum in (14) is unbiased. For other methods, we can see that momentum has a beneficial variance reduction effect as expected, and we observe convergence to a solution with higher precision. Note that FEDSHUFFLE still performs the best as predicted by our theory.

In the third plot, we analyse the difference between the default implementation of the aggregation step, that is denoted as "sum one" (FEDSHUFFLE w/ SO) since the sum of weights during aggregation is normalized to be one, and our unbiased version, where the weights are scaled by the probability of sampling the given client. We sample two clients in each step uniformly at random and each client runs one local epoch. As was discussed in Section 4, we observe that FEDSHUFFLE w/ SO converges to a worse solution due to objective inconsistency resulting from the biased aggregation.

Finally, we compare FEDSHUFFLE with uniform and importance sampling, where we sample one client per round and the sampled client runs one local epoch. For importance sampling (IS), each client is sampled proportionally to its dataset size. To better showcase the effect of importance sampling, we use a slightly different objective, where $d = 10$, the first client holds 8 data points, and other two clients hold one. As predicted by our theory (decrease of the $M$ term in Theorem 5.1), importance sampling leads to a substantial improvement.

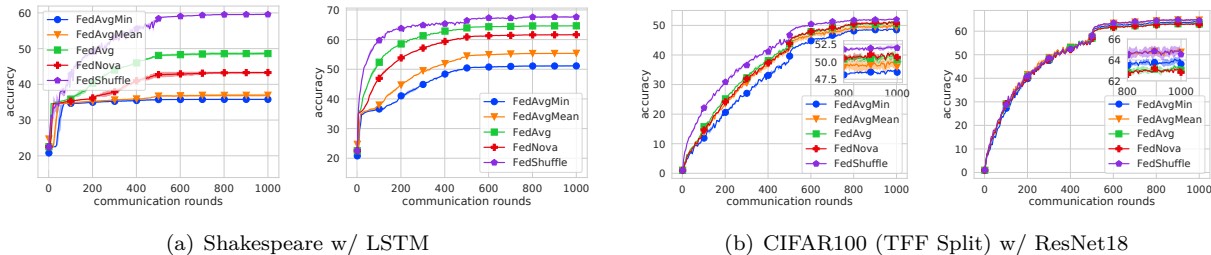

(a) Shakespeare w/ LSTM          (b) CIFAR100 (TFF Split) w/ ResNet18

Figure 2: Comparison of FEDAVGMIN, FEDAVG, FEDNOVA, FEDSHUFFLE on real-world datasets. Partial participation: in each round 16 client is sampled uniformly at random. All methods use random reshuffling. For Shakespeare, number of local epochs is 2 and for CIFAR100, it is 2 to 5 sampled uniformly at random at each communication round for each client. Left: Plain methods, without momentum. Right: Global momentum 0.9.

## 6.2 Training Deep Neural Networks

In the next experiments, we evaluate the same methods on the CIFAR100 Krizhevsky et al. (2009) and Shakespeare McMahan et al. (2017) datasets. The results already showcased theoretically (see Sections 4 and Appendix E) and empirically in the previous experiments that reshuffling leads to a substantial improvement over random sampling with replacement. Therefore, in these experiments we focus on other aspects of FEDSHUFFLE and show its superiority over other methods that use random reshuffling. To do that, we only consider random reshuffling methods in this section; thus FEDNOVA, FEDAVG, FEDAVGMIN and FEDAVGMEAN refer to FEDAVGRR, FEDNOVARR, FEDAVGMIN and FEDAVGMEAN with (partial) random reshuffling, respectively. For each task, we run 1000 rounds and we sample 16 clients in each round. For Shakespeare, each client runs two local epochs. For CIFAR100, all clients have the same number of data points. Thus, we follow (Wang et al., 2020) to create heterogeneity and test our FEDSHUFFLEGEN framework we assume each client runs 2 to 5 epochs uniformly at random. We investigate which method performs the best and, furthermore, we look at the effects of the momentum. and importance sampling, where the description of the used importance sampling strategy can be found in (Horváth & Richtárik, 2019, Section 2.3). We report the test accuracies in Figure 2 and Tables 2 and 3. We observe that FEDSHUFFLE outperforms all the baselines, for the Shakespeare dataset with a large margin. With respect to the global fixed momentum, we can see that this technique helps all the methods and substantially improves their performance. We note that FEDAVGMEAN and FEDAVGMIN perform exceptionally well for CIFAR100 (w/ momentum) but not for the Shakespeare dataset. We conjecture this is due to the significant heterogeneity of the Shakespeare dataset in terms of samples per client. In this setting, FEDAVGMIN does not utilize most of the data on clients with larger data-sets, while FEDAVGMEAN over-uses the data on clients with smaller data-size. Note that for CIFAR100, this is not the case as we use an equal-sized split. For completeness, we include the train loss corresponding to Figure 2 in Section F in the appendix.

## 7 Conclusion

This paper introduces and analyzes FEDSHUFFLE, which incorporates the practice of running local epochs with reshuffling in common FL implementations while also accounting for data imbalance across clients, and correcting for the resulting objective inconsistency problem that arises in FEDAVG-type methods. FEDSHUFFLE involves adjusting local learning rates based on the amount of local data, in addition to modified aggregation weights compared to prior work like FEDAVG or FEDNOVA. Under an additional Hessian smoothness assumption, incorporating momentum variance reduction leads to order-optimal rates, in the sense of matching the lower bounds achieved by non-local-update methods. The theoretical contributions of this work are verified in controlled experiments using quadratic functions, and the superiority of FEDSHUFFLE is also demonstrated in experiments training deep neural networks on standard FL benchmark problems.

Promising directions for future work include generalizing the analysis of FEDSHUFFLE to cover asynchronous execution (Huba et al., 2022; Nguyen et al., 2022), incorporating compression mechanisms to reduce communication overhead (Karimireddy et al., 2019c; Mishchenko et al., 2019), and to account for computational and communication heterogeneity across clients (Caldas et al., 2018b; Diao et al., 2020; Horváth et al., 2021).

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

# Appendix

## A   FEDAVG **with Random Reshuffling**

---

**Algorithm 2** FEDAVG with RR

---

 1: **Input:** initial global model $x^0$, global and local step sizes $\eta_g^r$, $\eta_l^r$
 2: **for** each round $r = 0, \ldots, R - 1$ **do**
 3:     server broadcasts $x^r$ to all clients $i \in \mathcal{S}^r \subset [n]$ sampled uniformly at random with $|\mathcal{S}^r| = b$
 4:     **for** each client $i \in \mathcal{S}^r$ (in parallel) **do**
 5:         initialize local model $y_{i,0,0}^r \leftarrow x^r$
 6:         **for** $e = 1, \ldots, E$ **do**
 7:             Sample permutation $\{\Pi_{i,e,0}^r, \ldots, \Pi_{i,e,|\mathcal{D}_i|-1}^k\}$ of $\{1, \ldots, |\mathcal{D}_i|\}$
 8:             **for** $j = 1, \ldots, |\mathcal{D}_i|$ **do**
 9:                 update $y_{i,e,j}^r = y_{i,e,j-1}^r - \eta_l^r \nabla f_{i\Pi_{i,e,j-1}^r}(y_{i,e,j-1}^r)$
10:             **end for**
11:             $y_{i,e+1,0}^r = y_{i,e,|\mathcal{D}_i|}^r$
12:         **end for**
13:         send $\Delta_i^r = y_{i,E,|\mathcal{D}_i|}^r - x^r$ to server
14:     **end for**
15:     server computes $\Delta^r = \frac{n}{b} \frac{1}{\sum_{i \in \mathcal{S}^r} w_i} \sum_{i \in \mathcal{S}^r} w_i \Delta_i^r$
16:     server updates global model $x^{r+1} = x^r - \eta_g^r \Delta^r$
17: **end for**

---

## B   Technicalities

### B.1   Convex Smooth Functions

We first state some implications of Assumption 3.2. It implies the following quadratic upper bound on $f_{ij}$

$$f_{ij}(y) \leq f_{ij}(x) + \langle \nabla f_{ij}(x), y - x \rangle + \frac{L}{2} \|y - x\|^2 . \tag{15}$$

In addition, if the functions $\{f_{ij}\}$ are convex and $x^\star$ is an optimum of $f$, then

$$\frac{1}{2L|\mathcal{D}|} \sum_{i,j} \|\nabla f_{ij}(x) - \nabla f_{ij}(x^\star)\|^2 \leq f(x) - f^\star . \tag{16}$$

Further, if $f_{ij}$ is twice-differentiable, then Assumption 3.2 implies that $\|\nabla^2 f_{ij}(x)\| \leq L$ for all $x$ ; see, e.g., (Nesterov et al., 2018, Theorem 2.1.5), where $\|\cdot\|$ refers here to the spectral norm.

Next, we include lemma that is useful to provide bounds for any smooth and strongly-convex functions. It can be seen as a generalization of the standard strong convexity inequality (2), but this bound can handle gradients computed at slightly perturbed points using smoothness assumption. This lemma turns out to be especially useful for local methods and was also used in the analysis of `FedAVG` in (Karimireddy et al., 2019b).

**Lemma B.1** (perturbed strong convexity)**.** *The following holds for any $L$-smooth and $\mu$-strongly convex function $h$, and any $x, y, z$ in the domain of $h$:*

$$\langle \nabla h(x), z - y \rangle \geq h(z) - h(y) + \frac{\mu}{4} \|y - z\|^2 - L \|z - x\|^2 .$$

*Proof.* Given any $x$, $y$, and $z$, one gets the following two inequalities using smoothness and strong convexity of the function $h$:

$$\langle \nabla h(x), z - x \rangle \geq h(z) - h(x) - \frac{L}{2}\|z - x\|^2$$

$$\langle \nabla h(x), x - y \rangle \geq h(x) - h(y) + \frac{\mu}{2}\|y - x\|^2.$$

Further, applying the relaxed triangle inequality gives

$$\frac{\mu}{2}\|y - x\|^2 \geq \frac{\mu}{4}\|y - z\|^2 - \frac{\mu}{2}\|x - z\|^2.$$

Combining all the inequalities together we have

$$\langle \nabla h(x), z - y \rangle \geq h(z) - h(y) + \frac{\mu}{4}\|y - z\|^2 - \frac{L + \mu}{2}\|z - x\|^2.$$

The lemma follows since it has to hold $L \geq \mu$. $\qquad\square$

## B.2 Convergence Derivations

In this section, we cover some technical lemmas which are useful to unroll recursions and derive convergence rates. computations later on. The provided Lemmas correspond to Lemma 1 and 2 in (Karimireddy et al., 2019b).

**Lemma B.2** (linear convergence rate)**.** *For every non-negative sequence $\{d_{r-1}\}_{r\geq 1}$ and any parameters $\mu > 0$, $\eta_{\max} \in (0, 1/\mu]$, $c \geq 0$, $R \geq \frac{1}{2\eta_{\max}\mu}$, there exists a constant step size $\eta \leq \eta_{\max}$ and weights $v_r \stackrel{def}{=} (1 - \mu\eta)^{1-r}$ such that for $V_R \stackrel{def}{=} \sum_{r=1}^{R+1} v_r$,*

$$\Psi_R \stackrel{def}{=} \frac{1}{V_R} \sum_{r=1}^{R+1} \left( \frac{v_r}{\eta}(1 - \mu\eta) d_{r-1} - \frac{v_r}{\eta} d_r + c\eta v_r \right) = \tilde{\mathcal{O}}\left( \mu d_0 \exp(-\mu\eta_{\max}R) + \frac{c}{\mu R} \right).$$

*Proof.* By substituting the value of $v_r$, we observe that we end up with a telescoping sum and estimate

$$\Psi_R = \frac{1}{\eta V_R} \sum_{r=1}^{R+1} (v_{r-1} d_{r-1} - v_r d_r) + \frac{c\eta}{V_R} \sum_{r=1}^{R+1} v_r \leq \frac{d_0}{\eta V_R} + c\eta.$$

When $R \geq \frac{1}{2\mu\eta}$, $(1 - \mu\eta)^R \leq \exp(-\mu\eta R) \leq \frac{2}{3}$. For such an $R$, we can lower bound $\eta V_R$ using

$$\eta V_R = \eta(1 - \mu\eta)^{-R} \sum_{r=0}^{R} (1 - \mu\eta)^r = \eta(1 - \mu\eta)^{-R} \frac{1 - (1 - \mu\eta)^R}{\mu\eta} \geq (1 - \mu\eta)^{-R} \frac{1}{3\mu}.$$

This proves that for all $R \geq \frac{1}{2\mu\eta}$,

$$\Psi_R \leq 3\mu d_0(1 - \mu\eta)^R + c\eta \leq 3\mu d_o \exp(-\mu\eta R) + c\eta.$$

The lemma now follows by carefully tuning $\eta$. Consider the following two cases depending on the magnitude of $R$ and $\eta_{\max}$:

- Suppose $\frac{1}{2\mu R} \leq \eta_{\max} \leq \frac{\log(\max(1, \mu^2 R d_0/c))}{\mu R}$. Then we can choose $\eta = \eta_{\max}$,

$$\Psi_R \leq 3\mu d_0 \exp\left[-\mu\eta_{\max}R\right] + c\eta_{\max} \leq 3\mu d_0 \exp\left[-\mu\eta_{\max}R\right] + \tilde{\mathcal{O}}\left(\frac{c}{\mu R}\right).$$

- Instead if $\eta_{\max} > \frac{\log(\max(1,\mu^2 R d_0 / c))}{\mu R}$, we pick $\eta = \frac{\log(\max(1,\mu^2 R d_0 / c))}{\mu R}$ to claim that

$$\Psi_R \leq 3\mu d_0 \exp\left[-\log(\max(1, \mu^2 R d_0 / c))\right] + \tilde{\mathcal{O}}\left(\frac{c}{\mu R}\right) \leq \tilde{\mathcal{O}}\left(\frac{c}{\mu R}\right).$$

$\square$

**Lemma B.3** (sub-linear convergence rate)**.** *For every non-negative sequence $\{d_{r-1}\}_{r\geq 1}$ and any parameters $\eta_{\max} \geq 0$, $c \geq 0$, $R \geq 0$, there exists a constant step size $\eta \leq \eta_{\max}$ and weights $v_r = 1$ such that,*

$$\Psi_R \overset{def}{=} \frac{1}{R+1} \sum_{r=1}^{R+1} \left(\frac{d_{r-1}}{\eta} - \frac{d_r}{\eta} + c_1 \eta + c_2 \eta^2\right) \leq \frac{d_0}{\eta_{\max}(R+1)} + \frac{2\sqrt{c_1 d_0}}{\sqrt{R+1}} + 2\left(\frac{d_0}{R+1}\right)^{\frac{2}{3}} c_2^{\frac{1}{3}}.$$

*Proof.* Unrolling the sum, we can simplify

$$\Psi_R \leq \frac{d_0}{\eta(R+1)} + c_1 \eta + c_2 \eta^2.$$

Similar to the strongly convex case (Lemma B.2), we distinguish the following cases:

- When $R+1 \leq \frac{d_0}{c_1 \eta_{\max}^2}$, and $R+1 \leq \frac{d_0}{c_2 \eta_{\max}^3}$ we pick $\eta = \eta_{\max}$ to claim

$$\Psi_R \leq \frac{d_0}{\eta_{\max}(R+1)} + c_1 \eta_{\max} + c_2 \eta_{\max}^2 \leq \frac{d_0}{\eta_{\max}(R+1)} + \frac{\sqrt{c_1 d_0}}{\sqrt{R+1}} + \left(\frac{d_0}{R+1}\right)^{\frac{2}{3}} c_2^{\frac{1}{3}}.$$

- In the other case, we have $\eta_{\max}^2 \geq \frac{d_0}{c_1(R+1)}$ or $\eta_{\max}^3 \geq \frac{d_0}{c_2(R+1)}$. We choose $\eta = \min\left\{\sqrt{\frac{d_0}{c_1(R+1)}}, \sqrt[3]{\frac{d_0}{c_2(R+1)}}\right\}$ to prove

$$\Psi_R \leq \frac{d_0}{\eta(R+1)} + c\eta = \frac{2\sqrt{c_1 d_0}}{\sqrt{R+1}} + 2\sqrt[3]{\frac{d_0^2 c_2}{(R+1)^2}}.$$

$\square$

## B.3 Variance bounds

In this section, we provide bounds that are useful to bound the variance of the gradient estimators used in this work. We first introduce standard variance decomposition and then state two lemmas.

For random variable $X$ and any $y \in \mathbb{R}^d$, the variance can be decomposed as

$$\mathbf{E}\left[\|X - \mathbf{E}[X]\|^2\right] = \mathbf{E}\left[\|X - y\|^2\right] - \|\mathbf{E}[X] - y\|^2. \tag{17}$$

The first lemma captures bounds for the variance of unbiased estimator with arbitrary proper sampling. This lemma is adapted from (Chen et al., 2020) originally introduced in (Horváth & Richtárik, 2019).

**Lemma B.4.** *Let $\zeta_1, \zeta_2, \ldots, \zeta_n$ be vectors in $\mathbb{R}^d$ and $w_1, w_2, \ldots, w_n$ be non-negative real numbers such that $\sum_{i=1}^n w_i = 1$. Define $\tilde{\zeta} \overset{def}{=} \sum_{i=1}^n w_i \zeta_i$. Let $\mathcal{S}$ be a proper sampling. If $s \in \mathbb{R}^n$ is such that*

$$\mathbf{P} - pp^\top \preceq \mathbf{Diag}(p_1 s_1, p_2 s_2, \ldots, p_n s_n), \tag{18}$$

*then*

$$\mathbf{E}\left[\left\|\sum_{i \in \mathcal{S}} \frac{w_i \zeta_i}{p_i} - \tilde{\zeta}\right\|^2\right] \leq \sum_{i=1}^n w_i^2 \frac{s_i}{p_i} \|\zeta_i\|^2, \tag{19}$$

*where the expectation is taken over $\mathcal{S}$.*

*Proof.* Let $1_{i \in \mathcal{S}} = 1$ if $i \in \mathcal{S}$ and $1_{i \in \mathcal{S}} = 0$ otherwise. Likewise, let $1_{i,j \in \mathcal{S}} = 1$ if $i, j \in \mathcal{S}$ and $1_{i,j \in \mathcal{S}} = 0$ otherwise. Note that $\mathbf{E}\left[1_{i \in \mathcal{S}}\right] = p_i$ and $\mathbf{E}\left[1_{i,j \in \mathcal{S}}\right] = p_{ij}$. Next, let us compute the mean of $X \stackrel{\text{def}}{=} \sum_{i \in \mathcal{S}} \frac{w_i \zeta_i}{p_i}$:

$$\mathbf{E}\left[X\right] = \mathbf{E}\left[\sum_{i \in \mathcal{S}} \frac{w_i \zeta_i}{p_i}\right] = \mathbf{E}\left[\sum_{i=1}^{n} \frac{w_i \zeta_i}{p_i} 1_{i \in \mathcal{S}}\right] = \sum_{i=1}^{n} \frac{w_i \zeta_i}{p_i} \mathbf{E}\left[1_{i \in \mathcal{S}}\right] = \sum_{i=1}^{n} w_i \zeta_i = \tilde{\zeta}.$$

Let $\boldsymbol{A} = [a_1, \ldots, a_n] \in \mathbb{R}^{d \times n}$, where $a_i = \frac{w_i \zeta_i}{p_i}$, and let $e$ be the vector of all ones in $\mathbb{R}^n$. We now write the variance of $X$ in a form which will be convenient to establish a bound:

$$\begin{aligned}
\mathbf{E}\left[\|X - \mathbf{E}\left[X\right]\|^2\right] &= \mathbf{E}\left[\|X\|^2\right] - \|\mathbf{E}\left[X\right]\|^2 \\
&= \mathbf{E}\left[\|\sum_{i \in \mathcal{S}} \frac{w_i \zeta_i}{p_i}\|^2\right] - \|\tilde{\zeta}\|^2 \\
&= \mathbf{E}\left[\sum_{i,j} \frac{w_i \zeta_i^\top}{p_i} \frac{w_j \zeta_j}{p_j} 1_{i,j \in \mathcal{S}}\right] - \|\tilde{\zeta}\|^2 \\
&= \sum_{i,j} p_{ij} \frac{w_i \zeta_i^\top}{p_i} \frac{w_j \zeta_j}{p_j} - \sum_{i,j} w_i w_j \zeta_i^\top \zeta_j \\
&= \sum_{i,j} (p_{ij} - p_i p_j) a_i^\top a_j \\
&= e^\top ((\boldsymbol{P} - pp^\top) \circ \boldsymbol{A}^\top \boldsymbol{A}) e.
\end{aligned} \tag{20}$$

Since, by assumption, we have $\boldsymbol{P} - pp^\top \preceq \mathbf{Diag}(p \circ s)$, we can further bound

$$e^\top ((\boldsymbol{P} - pp^\top) \circ \boldsymbol{A}^\top \boldsymbol{A}) e \le e^\top (\mathbf{Diag}(p \circ s) \circ \boldsymbol{A}^\top \boldsymbol{A}) e = \sum_{i=1}^{n} p_i s_i \|a_i\|^2.$$

$\square$

The second lemma bounds the variance of the estimator obtained using the sampling without replacement. The provided lemma is adopted from (Mishchenko et al., 2020).

**Lemma B.5.** *Let $\zeta_1, \ldots, \zeta_n \in \mathbb{R}^d$ be fixed vectors,*

$$\tilde{\zeta} \stackrel{\text{def}}{=} \frac{1}{n} \sum_{i=1}^{n} \zeta_i$$

*be their average and*

$$\sigma^2 \stackrel{\text{def}}{=} \frac{1}{n} \sum_{i=1}^{n} \|\zeta_i - \tilde{\zeta}\|^2$$

*be the population variance. Fix any $k \in \{1, \ldots, n\}$, let $\zeta_{\pi_1}, \ldots \zeta_{\pi_k}$ be sampled uniformly without replacement from $\{\zeta_1, \ldots, \zeta_n\}$ and $\tilde{\zeta}_\pi^k$ be their average. Then, the sample average and variance are given by*

$$\mathbf{E}\left[\tilde{\zeta}_\pi^k\right] = \tilde{\zeta}$$
$$\mathbf{E}\left[\|\tilde{\zeta}_\pi^k - \tilde{\zeta}\|^2\right] = \frac{n-k}{k(n-1)} \sigma^2. \tag{21}$$

*Proof.* The first claim follows by linearity of expectation and uniformity of sampling:

$$\mathbf{E}\left[\tilde{\zeta}_\pi^k\right] = \frac{1}{k} \sum_{i=1}^{k} \mathbf{E}\left[\zeta_{\pi_i}\right] = \frac{1}{k} \sum_{i=1}^{k} \tilde{\zeta} = \tilde{\zeta}.$$

To prove the second claim, let us first establish that the identity $\mathrm{cov}(\zeta_{\pi_i}, \zeta_{\pi_j}) = -\frac{\sigma^2}{n-1}$ holds for any $i \neq j$. Indeed,

$$\mathrm{cov}(\zeta_{\pi_i}, \zeta_{\pi_j}) = \mathbf{E}\left[\langle \zeta_{\pi_i} - \tilde{\zeta}, \zeta_{\pi_j} - \tilde{\zeta} \rangle\right] = \frac{1}{n(n-1)} \sum_{l=1}^{n} \sum_{m=1, m \neq l}^{n} \langle \zeta_l - \tilde{\zeta}, \zeta_m - \tilde{\zeta} \rangle$$

$$= \frac{1}{n(n-1)} \sum_{l=1}^{n} \sum_{m=1}^{n} \langle \zeta_l - \tilde{\zeta}, \zeta_m - \tilde{\zeta} \rangle - \frac{1}{n(n-1)} \sum_{l=1}^{n} \|\zeta_l - \tilde{\zeta}\|^2$$

$$= \frac{1}{n(n-1)} \sum_{l=1}^{n} \langle \zeta_l - \tilde{\zeta}, \sum_{m=1}^{n} (\zeta_m - \tilde{\zeta}) \rangle - \frac{\sigma^2}{n-1}$$

$$= -\frac{\sigma^2}{n-1}.$$

This identity helps us to establish the formula for sample variance:

$$\mathbf{E}\left[\|\tilde{\zeta}_\pi^k - \tilde{\zeta}\|^2\right] = \frac{1}{k^2} \sum_{i=1}^{k} \sum_{j=1}^{k} \mathrm{cov}(\zeta_{\pi_i}, \zeta_{\pi_j})$$

$$= \frac{1}{k^2} \mathbf{E}\left[\sum_{i=1}^{k} \|\zeta_{\pi_i} - \tilde{\zeta}\|^2\right] + \sum_{i=1}^{k} \sum_{j=1, j \neq i}^{k} \mathrm{cov}(\zeta_{\pi_i}, \zeta_{\pi_j})$$

$$= \frac{1}{k^2}\left(k\sigma^2 - k(k-1)\frac{\sigma^2}{n-1}\right) = \frac{n-k}{k(n-1)}\sigma^2. \qquad \square$$

### B.4 Technical Lemmas

Next, we state a relaxed triangle inequality for the squared $\ell_2$ norm.

**Lemma B.6** (relaxed triangle inequality). *Let $\{v_1, \ldots, v_\tau\}$ be $\tau$ vectors in $\mathbb{R}^d$. Then the following are true*

$$\|v_i + v_j\|^2 \leq (1+a)\|v_i\|^2 + \left(1 + \frac{1}{a}\right)\|v_j\|^2 \text{ for any } a > 0 \tag{22}$$

*and*

$$\left\|\sum_{i=1}^{\tau} v_i\right\|^2 \leq \tau \sum_{i=1}^{\tau} \|v_i\|^2. \tag{23}$$

*Proof.* The proof of the first statement for any $a > 0$ follows from the identity:

$$\|v_i + v_j\|^2 = (1+a)\|v_i\|^2 + \left(1 + \frac{1}{a}\right)\|v_j\|^2 - \left\|\sqrt{a}v_i + \frac{1}{\sqrt{a}}v_j\right\|^2.$$

For the second inequality, we use the convexity of $x \to \|x\|^2$ and Jensen's inequality

$$\left\|\frac{1}{\tau} \sum_{i=1}^{\tau} v_i\right\|^2 \leq \frac{1}{\tau} \sum_{i=1}^{\tau} \|v_i\|^2.$$

$\square$

---

**Algorithm 3** FEDSHUFFLE

---

1: **Input:** initial global model $x^0$, global and local step sizes $\eta_g^r$, $\eta_l^r$, proper distribution $\mathcal{S}$
2: **for** each round $r = 0, \ldots, R-1$ **do**
3:     server broadcasts $x^r$ to all clients $i \in \mathcal{S}^r \sim S$
4:     **for** each client $i \in \mathcal{S}^r$ (in parallel) **do**
5:         initialize local model $y_{i,0,0}^r \leftarrow x^r$
6:         **for** $e = 1, \ldots, E$ **do**
7:             Sample permutation $\{\Pi_{i,e,0}^r, \ldots, \Pi_{i,e,|\mathcal{D}_i|-1}^k\}$ of $\{1, \ldots, |\mathcal{D}_i|\}$
8:             **for** $j = 1, \ldots, |\mathcal{D}_i|$ **do**
9:                 update $y_{i,e,j}^r = y_{i,e,j-1}^r - \eta_l^r/|\mathcal{D}_i|\nabla f_{i\Pi_{i,e,j-1}^r}(y_{i,e,j-1}^r)$
10:             **end for**
11:             $y_{i,e+1,0}^r = y_{i,e,|\mathcal{D}_i|}^r$
12:         **end for**
13:         send $\Delta_i^r = y_{i,E,|\mathcal{D}_i|}^r - x^r$ to server
14:     **end for**
15:     server computes $\Delta^r = \sum_{i \in \mathcal{S}^r} \frac{w_i}{p_i}\Delta_i^r$
16:     server updates global model $x^{r+1} = x^r - \eta_g^r \Delta^r$
17: **end for**

---

## C   Algorithm 3: Convergence Analysis (Proof of Theorem 5.1)

The style of our proof technique is related to the analysis of FEDAVG of (Karimireddy et al., 2019b). We start with proof for convex functions. By $\mathbf{E}_r[\cdot]$, we denote the expectation conditioned on the all history prior to communication round $r$. We first establish the bound on the progress in a single communication round.

**Lemma C.1. (one round progress)** *Suppose Assumptions 3.1 – 3.4 hold. For any constant step sizes* $\eta_l^r \overset{def}{=} \eta_l$ *and* $\eta_l^r \overset{def}{=} \eta_l$ *satisfying* $\eta_l \leq \frac{1}{(1+MB^2)4LE\eta_g}$ *and effective step size* $\tilde{\eta} \overset{def}{=} E\eta_g\eta_l$, *the updates of* FEDSHUFFLE *satisfy*

$$\mathbf{E}\left[\|x^r - x^\star\|^2\right] \leq \left(1 - \frac{\mu\tilde{\eta}}{2}\right)\mathbf{E}\left[\|x^{r-1} - x^\star\|^2\right] - \tilde{\eta}\mathbf{E}_{r-1}\left[f(x^{r-1}) - f^\star\right] + 3L\tilde{\eta}\xi^r + 2\tilde{\eta}^2 MG^2,$$

*where $\xi_r$ is the drift caused by the local updates on the clients defined to be*

$$\xi^r \overset{def}{=} \frac{1}{|\mathcal{D}|E}\sum_{e=1}^{E}\sum_{i=1}^{n}\sum_{j=1}^{|\mathcal{D}_i|}\mathbf{E}_{r-1}\left[\|y_{i,e,j-1}^r - x^{r-1}\|^2\right]$$

*and* $M \overset{def}{=} \max_{i \in [n]}\left\{\frac{s_i}{p_i}w_i\right\}$.

*Proof.* For a better readability of the proofs in one round progress, we drop the superscript that represents the current completed communication round $r - 1$.

By the definition in Algorithm 3, the update $\Delta$ can be written as

$$\Delta = -\eta_g\sum_{i \in \mathcal{S}}\frac{w_i}{p_i}\Delta_i = -\frac{\tilde{\eta}}{E|\mathcal{D}|}\sum_{i \in \mathcal{S}}\sum_{e=1}^{E}\sum_{j=1}^{|\mathcal{D}_i|}\frac{1}{p_i}\nabla f_{i\Pi_{i,e,j-1}}(y_{i,e,j-1}).$$

We adopt the convention that summation $\sum_{i \in \mathcal{M},e,j}$ ($\mathcal{M}$ is either $[n]$ or $\mathcal{S}$) refers to the summations $\sum_{i \in \mathcal{M}}\sum_{e=1}^{E}\sum_{j=1}^{|\mathcal{D}_i|}$ unless otherwise stated. Furthermore, we denote $g_{i,e,j} \overset{def}{=} \nabla f_{i\Pi_{i,e,j-1}}(y_{i,e,j-1})$. Using

above, we proceed as

$$\mathbf{E}_{r-1}\left[\|x+\Delta-x^\star\|^2\right] = \|x-x^\star\|^2 \underbrace{-2\mathbf{E}_{r-1}\left[\frac{\tilde{\eta}}{E|\mathcal{D}|}\sum_{i\in\mathcal{S},e,j}\frac{1}{p_i}\langle g_{i,e,j},x-x^\star\rangle\right]}_{\mathcal{A}_1} + \underbrace{\tilde{\eta}^2\mathbf{E}_{r-1}\left[\left\|\frac{1}{E|\mathcal{D}|}\sum_{i\in\mathcal{S},e,j}\frac{1}{p_i}g_{i,e,j}\right\|^2\right]}_{\mathcal{A}_2}.$$

To bound the term $\mathcal{A}_1$, we apply Lemma B.1 to each term of the summation with $h=f_{ij}$, $x=y_{i,e,j-1}$, $y=x^\star$, and $z=x$. Therefore,

$$\mathcal{A}_1 = -\mathbf{E}_{r-1}\left[\frac{2\tilde{\eta}}{E|\mathcal{D}|}\sum_{i\in\mathcal{S},e,j}\frac{1}{p_i}\langle g_{i,e,j},x-x^\star\rangle\right]$$

$$\leq \mathbf{E}_{r-1}\left[\frac{2\tilde{\eta}}{E|\mathcal{D}|}\sum_{i\in\mathcal{S},e,j}\frac{1}{p_i}\left(f_{i\Pi_{i,e,j-1}}(x^\star)-f_{i\Pi_{i,e,j-1}}(x)+L\|y_{i,e,j-1}-x\|^2-\frac{\mu}{4}\|x-x^\star\|^2\right)\right]$$

$$= -2\tilde{\eta}\left(f(x)-f^\star+\frac{\mu}{4}\|x-x^\star\|^2\right)+2L\tilde{\eta}\xi.$$

For the second term $\mathcal{A}_2$, we have

$$\mathcal{A}_2 = \tilde{\eta}^2\mathbf{E}_{r-1}\left[\left\|\frac{1}{E|\mathcal{D}|}\sum_{i\in\mathcal{S},e,j}\frac{1}{p_i}g_{i,e,j}\right\|^2\right]$$

$$\stackrel{(17)}{\leq} \frac{\tilde{\eta}^2}{E^2|\mathcal{D}|^2}\mathbf{E}_{r-1}\left[\left\|\sum_{i\in\mathcal{S},e,j}\frac{1}{p_i}g_{i,e,j}-\sum_{i\in[n],e,j}g_{i,e,j}\right\|^2+\left\|\sum_{i\in[n],e,j}g_{i,e,j}\right\|^2\right]$$

$$\stackrel{(19)}{\leq} \frac{\tilde{\eta}^2}{E^2|\mathcal{D}|^2}\mathbf{E}_{r-1}\left[\sum_{i\in[n]}\frac{s_i}{p_i}\left\|\sum_{e,j}g_{i,e,j}\right\|^2+\left\|\sum_{i\in[n],e,j}g_{i,e,j}\right\|^2\right]$$

$$\stackrel{(23)}{\leq} \frac{2\tilde{\eta}^2}{E^2|\mathcal{D}|^2}\mathbf{E}_{r-1}\left[\sum_{i\in[n]}\frac{s_i}{p_i}\left\|\sum_{e,j}g_{i,e,j}-\nabla f_{i\Pi_{i,e,j-1}}(x)\right\|^2\right]+2\tilde{\eta}^2\sum_{i\in[n]}\frac{s_i}{p_i}w_i^2\|\nabla f_i(x)\|^2$$

$$+\frac{2\tilde{\eta}^2}{E^2|\mathcal{D}|^2}\mathbf{E}_{r-1}\left[\left\|\sum_{i\in[n],e,j}g_{i,e,j}-\nabla f_{i\Pi_{i,e,j-1}}(x)\right\|^2\right]+2\tilde{\eta}^2\|\nabla f(x)\|^2$$

$$\stackrel{(23)}{\leq} \frac{2\tilde{\eta}^2}{E|\mathcal{D}|}\mathbf{E}_{r-1}\left[\sum_{i\in[n],e,i}\frac{s_i}{p_i}w_i\|g_{i,e,j}-\nabla f_{i\Pi_{i,e,j-1}}(x)\|^2\right]+2\tilde{\eta}^2\sum_{i\in[n]}\frac{s_i}{p_i}w_i^2\|\nabla f_i(x)\|^2$$

$$+\frac{2\tilde{\eta}^2}{E|\mathcal{D}|}\sum_{i\in[n],e,j}\mathbf{E}_{r-1}\left[\|g_{i,e,j}-\nabla f_{i\Pi_{i,e,j-1}}(x)\|^2\right]+2\tilde{\eta}^2\|\nabla f(x)\|^2$$

$$\stackrel{(3)}{\leq} 2\max_{i\in[n]}\left\{\frac{s_i}{p_i}w_i\right\}\tilde{\eta}^2L^2\xi+2\tilde{\eta}^2\max_{i\in[n]}\left\{\frac{s_i}{p_i}w_i\right\}\sum_{i\in[n]}w_i\|\nabla f_i(x)\|^2$$

$$+2\tilde{\eta}^2L^2\xi+2\tilde{\eta}^2\|\nabla f(x)\|^2$$

$$\stackrel{(4)}{\leq} 2\left(1+\max_{i\in[n]}\left\{\frac{s_i}{p_i}w_i\right\}\right)\tilde{\eta}^2L^2\xi+2\tilde{\eta}^2\max_{i\in[n]}\left\{\frac{s_i}{p_i}w_i\right\}G^2+2\tilde{\eta}^2\left(1+\max_{i\in[n]}\left\{\frac{s_i}{p_i}w_i\right\}B^2\right)\|\nabla f(x)\|^2$$

$$\stackrel{(16)}{\leq} 2\left(1+\max_{i\in[n]}\left\{\frac{s_i}{p_i}w_i\right\}\right)\tilde{\eta}^2L^2\xi+2\tilde{\eta}^2\max_{i\in[n]}\left\{\frac{s_i}{p_i}w_i\right\}G^2+4L\tilde{\eta}^2\left(1+\max_{i\in[n]}\left\{\frac{s_i}{p_i}w_i\right\}B^2\right)(f(x)-f^\star).$$

Recall $M \stackrel{\text{def}}{=} \max_{i \in [n]} \left\{ \frac{s_i}{p_i} w_i \right\}$, therefore by plugging back the bounds on $\mathcal{A}_1$ and $\mathcal{A}_2$,

$$\mathbf{E}_{r-1} \left[ \|x + \Delta - x^\star\|^2 \right] \leq \left(1 - \frac{\mu\tilde{\eta}}{2}\right) \|x - x^\star\|^2 - (2\tilde{\eta} - 4L\tilde{\eta}^2(MB^2+1))(f(x) - f^\star)$$
$$+ (1 + (1+M)\tilde{\eta}L)2L\tilde{\eta}\xi + 2\tilde{\eta}^2 MG^2 \, .$$

The lemma now follows by observing that $4L\tilde{\eta}(MB^2+1) \leq 1$ and that $B \geq 1$. $\qquad\square$

The next step is to bound client drift caused by the heterogeneity of clients' data. Unlike (Karimireddy et al., 2019b), we do not upper bound local client drift using recursive estimates, but we directly exploit smoothness combined with the relaxed triangle inequality.

**Lemma C.2. (bounded drift)** *Suppose Assumptions 3.2 – 3.4 hold. Then the updates of* FEDSHUFFLE *for any step size satisfying* $\eta_l \leq \frac{1}{(1+(P+1)B+MB^2))4LE\eta_g}$ *have bounded drift:*

$$3L\tilde{\eta}\xi_r \leq \frac{9}{10}\tilde{\eta}(f(x^r) - f^\star) + 9\frac{\tilde{\eta}^3}{\eta_g^2 E^2} \left( (E^2 + P^2)G^2 + \sigma^2 \right) \, ,$$

*where* $P^2 \stackrel{\text{def}}{=} \max_{i \in [n]} \frac{P_i^2}{3|\mathcal{D}_i|}$, $\sigma^2 \stackrel{\text{def}}{=} \frac{1}{3|\mathcal{D}|} \sum_{i \in [n]} \sigma_i^2$ *and* $M \stackrel{\text{def}}{=} \max_{i \in [n]} \left\{ \frac{s_i}{p_i} w_i \right\}$.

*Proof.* We adopt the same convention as for the previous proof, i.e., dropping superscripts, simplifying sum notation and having $g_{i,e,j} \stackrel{\text{def}}{=} \nabla f_{i\Pi_{i,e,j-1}}(y_{i,e,j-1})$. Therefore,

$$\xi = \frac{1}{|\mathcal{D}|E} \sum_{i \in [n], i, j} \mathbf{E}_{r-1} \left[ \|y_{i,e,j-1}^r - x^{r-1}\|^2 \right]$$

$$= \frac{\eta_l^2}{|\mathcal{D}|E} \sum_{i \in [n], e, j} \frac{1}{|\mathcal{D}_i|^2} \mathbf{E}_{r-1} \left[ \left\| \sum_{l=0}^{e-1} \sum_{c=1}^{\mathcal{D}_i} g_{i,l,c} + \sum_{c=1}^{j} g_{i,e,c} \right\|^2 \right]$$

$$\stackrel{(22)}{\leq} \frac{2\eta_l^2}{|\mathcal{D}|E} \sum_{i \in [n], e, j} \frac{1}{|\mathcal{D}_i|^2} \mathbf{E}_{r-1} \left[ \left\| \sum_{l=0}^{e-1} \sum_{c=1}^{\mathcal{D}_i} g_{i,l,c} - (e-1)|\mathcal{D}_i|\nabla f_i(x) + \sum_{c=1}^{j} g_{i,e,c} - \nabla f_{i\Pi_{i,e,c-1}}(x) \right\|^2 \right]$$

$$+ \frac{2\eta_l^2}{|\mathcal{D}|E} \sum_{i \in [n], e, j} \frac{1}{|\mathcal{D}_i|^2} \mathbf{E}_{r-1} \left[ \left\| (e-1)|\mathcal{D}_i|\nabla f_i(x) + \sum_{c=1}^{j} \nabla f_{i\Pi_{i,e,c-1}}(x) \right\|^2 \right]$$

Next, we upper bound each term separately. For the first term, we have

$$\frac{2\eta_l^2}{|\mathcal{D}|E} \sum_{i \in [n], e, j} \frac{1}{|\mathcal{D}_i|^2} \mathbf{E}_{r-1} \left[ \left\| \sum_{l=0}^{e-1} \sum_{c=1}^{\mathcal{D}_i} g_{i,l,c} - (e-1)|\mathcal{D}_i|\nabla f_i(x) + \sum_{c=1}^{j} g_{i,e,c} - \nabla f_{i\Pi_{i,e,c-1}}(x) \right\|^2 \right]$$

$$\stackrel{(23)}{\leq} \frac{2\eta_l^2}{|\mathcal{D}|E} \sum_{i \in [n], e, j} \frac{((e-1)|\mathcal{D}_i| + j)}{|\mathcal{D}_i|^2} \mathbf{E}_{r-1} \left[ \left( \sum_{l=0}^{e-1} \sum_{c=1}^{\mathcal{D}_i} \|g_{i,l,c} - \nabla f_{i\Pi_{i,e,j-1}}(x)\|^2 + \sum_{c=1}^{j} \|g_{i,e,c} - \nabla f_{i\Pi_{i,e,c-1}}(x)\|^2 \right) \right]$$

$$\stackrel{(3)}{\leq} 2\eta_l^2 L^2 E^2 \xi \, .$$

For the second term we obtain

$$\frac{2\eta_l^2}{|\mathcal{D}|E} \sum_{i\in[n],e,j} \frac{1}{|\mathcal{D}_i|^2} \mathbf{E}_{r-1} \left[ \left\| (e-1)|\mathcal{D}_i|\nabla f_i(x) + \sum_{c=1}^{j} \nabla f_{i\Pi_{i,e,c-1}}(x) \right\|^2 \right]$$

$$\overset{(17)}{\leq} \frac{2\eta_l^2}{|\mathcal{D}|E} \sum_{i\in[n],e,j} \frac{1}{|\mathcal{D}_i|^2} \mathbf{E}_{r-1} \left[ \|((e-1)|\mathcal{D}_i|+j)\nabla f_i(x)\|^2 + \left\| \sum_{c=1}^{j} \nabla f_{i\Pi_{i,e,c-1}}(x) - j\nabla f_i(x) \right\|^2 \right]$$

$$\overset{(21)}{\leq} \frac{2\eta_l^2}{|\mathcal{D}|E} \sum_{i\in[n],e,j} \frac{1}{|\mathcal{D}_i|^2} \left( ((e-1)|\mathcal{D}_i|+j)^2 \|\nabla f_i(x)\|^2 + \frac{j(|\mathcal{D}_i|-j)}{(|\mathcal{D}_i|-1)} \frac{1}{|\mathcal{D}_i|} \sum_{c=1}^{\mathcal{D}_i} \|\nabla f_{ic}(x) - \nabla f_i(x)\|^2 \right)$$

$$\overset{(6)}{\leq} \frac{2\eta_l^2}{|\mathcal{D}|E} \sum_{i\in[n],e,j} \frac{1}{|\mathcal{D}_i|^2} \left( ((e-1)|\mathcal{D}_i|+j)^2 \|\nabla f_i(x)\|^2 + \frac{j(|\mathcal{D}_i|-j)}{(|\mathcal{D}_i|-1)} (\sigma^2 + P^2 \|\nabla f_i(x)\|^2) \right)$$

$$\leq \frac{2\eta_l^2}{|\mathcal{D}|E} \sum_{i\in[n]} \frac{|\mathcal{D}_i|^3 E^3}{|\mathcal{D}_i|^2} \|\nabla f_i(x)\|^2 + \frac{E(|\mathcal{D}_i|+1)|\mathcal{D}_i|}{6|\mathcal{D}_i|^2} (\sigma_i^2 + P_i^2 \|\nabla f_i(x)\|^2)$$

$$\leq 2\eta_l^2 E^2 \sum_{i\in[n]} \left( 1 + \frac{P_i^2}{3|\mathcal{D}_i|E^2} \right) w_i \|\nabla f_i(x)\|^2 + \frac{\sigma_i^2}{3|\mathcal{D}|E^2}$$

$$\overset{(5)}{\leq} 4LB^2 \eta_l^2 E^2 (1+P^2)(f(x)-f^\star) + 2\eta_l^2 ((E^2+P^2)G^2 + \sigma^2).$$

Combining the upper bounds

$$\xi \leq 2\eta_l^2 L^2 E^2 \xi + 4LB^2 \eta_l^2 E^2 (1+P^2)(f(x)-f^\star) + 2\eta_l^2 ((E^2+P^2)G^2 + \sigma^2).$$

Since $2\eta_l^2 L^2 E^2 \leq \frac{1}{8}$ and $4LB^2 \eta_l^2 E^2 (1+P^2) \leq \frac{1}{4L}$, therefore

$$3L\tilde{\eta}\xi \leq \frac{9}{10}\tilde{\eta}(f(x)-f^\star) + 9\tilde{\eta}\eta_l^2 ((E^2+P^2)G^2 + \sigma^2),$$

which concludes the proof. $\qquad\square$

Adding the statements of the above Lemmas C.1 and C.2, we get

$$\mathbf{E}\left[\|x^r - x^\star\|^2\right] \leq \left(1 - \frac{\mu\tilde{\eta}}{2}\right) \mathbf{E}\left[\|x^{r-1} - x^\star\|^2\right] - \frac{\tilde{\eta}}{10}\mathbf{E}_{r-1}\left[f(x^{r-1}) - f^\star\right] + 2\tilde{\eta}^2 MG^2 + \frac{9\tilde{\eta}^3}{\eta_g^2 E^2} ((E^2+P^2)G^2 + \sigma^2)$$

$$= \left(1 - \frac{\mu\tilde{\eta}}{2}\right) \mathbf{E}\left[\|x^{r-1} - x^\star\|^2\right] - \frac{\tilde{\eta}}{10}\mathbf{E}_{r-1}\left[f(x^{r-1}) - f^\star\right] + 2\tilde{\eta}^2 \left(MG^2 + \frac{9\tilde{\eta}}{2\eta_g^2 E^2} ((E^2+P^2)G^2 + \sigma^2)\right),$$

Moving the $(f(x^{r-1}) - f(x^\star))$ term and dividing both sides by $\frac{\tilde{\eta}}{10}$, we get the following bound for any $\tilde{\eta} \leq \frac{1}{(1+(P+1)B+MB^2))4L}$

$$\mathbf{E}_{r-1}\left[f(x^{r-1}) - f^\star\right] \leq \frac{10}{\tilde{\eta}}\left(1 - \frac{\mu\tilde{\eta}}{2}\right) \mathbf{E}\left[\|x^{r-1} - x^\star\|^2\right] + 20\tilde{\eta}\left(MG^2 + \frac{9\tilde{\eta}}{2\eta_g^2 E^2} ((E^2+P^2)G^2 + \sigma^2)\right).$$

If $\mu = 0$ (weakly-convex), we can directly apply Lemma B.3. Otherwise, by averaging using weights $v_r = (1 - \frac{\mu\tilde{\eta}}{2})^{1-r}$ and using the same weights to pick output $\bar{x}^R$, we can simplify the above recursive bound (see proof of Lem. B.2) to prove that for any $\tilde{\eta}$ satisfying $\frac{1}{\mu R} \leq \tilde{\eta} \leq \frac{1}{(1+(P+1)B+MB^2))4L}$

$$\mathbf{E}\left[f(\bar{x}^R)\right] - f(x^\star) \leq \underbrace{10\|x^0 - x^\star\|^2}_{\overset{\text{def}}{=}d}\mu \exp\left(-\frac{\tilde{\eta}}{2}\mu R\right) + \tilde{\eta}(\underbrace{4MG^2}_{\overset{\text{def}}{=}c_1}) + \tilde{\eta}^2 \underbrace{\left(\frac{18}{\eta_g^2 E^2} ((E^2+P^2)G^2 + \sigma^2)\right)}_{\overset{\text{def}}{=}c_2}.$$

Now, the choice of $\tilde{\eta} = \min\left\{ \frac{\log(\max(1,\mu^2 Rd/c_1))}{\mu R}, \frac{1}{(1+(P+1)B+MB^2))4L} \right\}$ yields the desired rate.

For the non-convex case, one first exploits the smoothness assumption (Assumption 3.2) (extra smoothness term $L$ in the first term in the convergence guarantee) and the rest of the proof follows essentially in the same steps as the provided analysis. The only difference is that distance to an optimal solution is replaced by functional difference, i.e., $\|x^0 - x^\star\|^2 \to f(x^0) - f^\star$. The final convergence bound also relies on Lemma B.3. For completeness, we provide the proof below.

We adapt the same notation simplification as for the prior cases, see proof of Lemma C.1. Since $\{f_{ij}\}$ are $L$-smooth then $f$ is also $L$ smooth. Therefore,

$$
\begin{aligned}
\mathbf{E}_{r-1}\left[f(x+\Delta)\right] &\leq f(x) + \mathbf{E}_{r-1}\left[\langle\nabla f(x), \Delta\rangle\right] + \frac{L}{2}\mathbf{E}_{r-1}\left[\|\Delta\|^2\right] \\
&= f(x) - \mathbf{E}_{r-1}\left[\left\langle\nabla f(x), \frac{\tilde{\eta}}{E|\mathcal{D}|}\sum_{i\in[n],e,j}g_{i,e,j}\right\rangle\right] + \frac{L\tilde{\eta}^2}{2}\mathbf{E}_{r-1}\left[\left\|\frac{1}{E|\mathcal{D}|}\sum_{i\in\mathcal{S},e,j}\frac{1}{p_i}g_{i,e,j}\right\|^2\right] \\
&\stackrel{(22)}{\leq} f(x) - \frac{\tilde{\eta}}{2}\|\nabla f(x)\|^2 + \frac{\tilde{\eta}}{2}\mathbf{E}_{r-1}\left[\left\|\frac{1}{E|\mathcal{D}|}\sum_{i\in[n],e,j}g_{i,e,j} - \nabla f_{i\Pi_{i,e,j-1}}(x)\right\|^2\right] \\
&\quad + \frac{L\tilde{\eta}^2}{2}\mathbf{E}_{r-1}\left[\left\|\frac{1}{E|\mathcal{D}|}\sum_{i\in\mathcal{S},e,j}\frac{1}{p_i}g_{i,e,j}\right\|^2\right] \\
&\stackrel{(3)+(23)}{\leq} f(x) - \frac{\tilde{\eta}}{2}\|\nabla f(x)\|^2 + \frac{\tilde{\eta}L^2}{2}\xi + \frac{L\tilde{\eta}^2}{2}\mathbf{E}_{r-1}\left[\left\|\frac{1}{E|\mathcal{D}|}\sum_{i\in\mathcal{S},e,j}\frac{1}{p_i}g_{i,e,j}\right\|^2\right] .
\end{aligned}
$$

We upper-bound the last term using the bound of $\mathcal{A}_2$ in the proof of Lemma C.1 (note that this proof does not rely on the convexity assumption). Thus, we have

$$
\mathbf{E}_{r-1}\left[f(x+\Delta)\right] \leq f(x) - \frac{\tilde{\eta}}{2}\|\nabla f(x)\|^2 + \frac{\tilde{\eta}L^2}{2}\xi + (1+M)\tilde{\eta}^2 L^3\xi + \tilde{\eta}^2 MG^2 L + \tilde{\eta}^2\left(1+MB^2\right)L\|\nabla f(x)\|^2 .
$$

The bound on the step size $\tilde{\eta} \leq \frac{1}{(1+(P+1)B+MB^2))4L}$ implies

$$
\mathbf{E}_{r-1}\left[f(x+\Delta)\right] \leq f(x) - \frac{\tilde{\eta}}{4}\|\nabla f(x)\|^2 + \frac{3\tilde{\eta}L^2}{4}\xi + \tilde{\eta}^2 MG^2 L .
$$

Next, we reuse the partial result of Lemma C.2 that does not require convexity, i.e., we replace $f(x) - f^\star$ with $\frac{1}{2L}\|\nabla f(x)\|^2$. Therefore,

$$
\mathbf{E}_{r-1}\left[f(x+\Delta)\right] \leq f(x) - \frac{\tilde{\eta}}{8}\|\nabla f(x)\|^2 + \frac{9\tilde{\eta}^3 L}{4\eta_g^2 E^2}\left((E^2+P^2)G^2+\sigma^2\right) + \tilde{\eta}^2 MG^2 L .
$$

By adding $f^\star$ to both sides, reordering, dividing by $\tilde{\eta}$ and taking full expectation, we obtain

$$
\mathbf{E}\left[\|\nabla f(x^r)\|^2\right] \leq \frac{8}{\tilde{\eta}}\left(\mathbf{E}\left[f(x^r)-f^\star\right] - \mathbf{E}\left[f(x^{r+1})-f^\star\right]\right) + \tilde{\eta}\underbrace{8MG^2 L}_{c_1} + \tilde{\eta}^2\underbrace{\frac{18L}{\eta_g^2 E^2}\left((E^2+P^2)G^2+\sigma^2\right)}_{c_2} .
$$

Applying Lemma B.3 concludes the proof.

# D  FEDSHUFFLEMVR: **Convergence Analysis (Proof of Theorem 5.2)**

The style of our proof technique is related to the convergence analysis of MIMELITEMVR (Karimireddy et al., 2020) and non-convex Random Reshuffling (Mishchenko et al., 2020). By $\mathbf{E}_r\left[\cdot\right]$, we denote the expectation conditioned on the all history prior to communication round $r$. For the sake of notation, we drop the superscript that represents the communication round $r$ in the proofs. Furthermore, we use superscripts $^+$ and $^-$ to denote $^{r+1}$ and $^{r-1}$, respectively.

Firstly, we analyse a single local epoch. For the ease of presentation, we denote $y_{i,e,0} \overset{\text{def}}{=} y_{i,e}$ and

$$g_{i,e} \overset{\text{def}}{=} \frac{1}{|\mathcal{D}_i|} \sum_{j=1}^{|\mathcal{D}_i|} \nabla f_{i\Pi_{i,e,j-1}}(y_{i,e,j-1}) \tag{24}$$

and

$$d_{i,e} \overset{\text{def}}{=} a\nabla f_i(y_{i,e}) + (1-a)m + (1-a)(\nabla f_i(y_{i,e}) - \nabla f_i(x)). \tag{25}$$

In the first lemma, we investigate how $g_{i,e}$ update differs from the full local gradient update.

**Lemma D.1.** *Suppose Assumptions 3.2-3.4 hold with $P_i = 0$ for all $i \in [n]$ and $B = 1$ and only one client is sampled. Then, for $\eta_l \leq \frac{1}{2L}$*

$$\mathbf{E}_e\left[\|g_{i,e} - \nabla f_i(y_{i,e})\|^2\right] \leq 2\eta_l^2 L^2 \frac{\sigma_i^2}{|\mathcal{D}_i|} + \eta_l^2 L^2 \|d_{i,e}\|^2. \tag{26}$$

*Proof.* By definition,

$$\mathbf{E}_e\left[\|g_{i,e} - \nabla f_i(y_{i,e})\|^2\right] = \mathbf{E}_e\left[\left\|\frac{1}{|\mathcal{D}_i|}\sum_{j=1}^{|\mathcal{D}_i|}\nabla f_{i\Pi_{i,e,j-1}}(y_{i,e,j-1}) - \nabla f_{i\Pi_{i,e,j-1}}(y_{i,e})\right\|^2\right]$$

$$\overset{(16),(23)}{\leq} \frac{L^2}{|\mathcal{D}_i|}\mathbf{E}_e\left[\sum_{j=1}^{|\mathcal{D}_i|}\|y_{i,e,j-1} - y_{i,e}\|^2\right].$$

Let us now analyze $\sum_{j=1}^{|\mathcal{D}_i|}\|y_{i,e,j-1} - y_{i,e}\|^2$,

$$\sum_{j=1}^{|\mathcal{D}_i|}\|y_{i,e,j-1} - y_{i,e}\|^2 = \frac{\eta_l^2}{|\mathcal{D}_i|^2}\sum_{j=1}^{|\mathcal{D}_i|}\left\|j(1-a)(m - \nabla f_i(x)) - \sum_{k=0}^{j-1}\nabla f_{i\Pi_{i,e,k}}(y_{i,e,k})\right\|^2$$

$$\overset{(23)}{\leq} \frac{2\eta_l^2}{|\mathcal{D}_i|^2}\sum_{j=1}^{|\mathcal{D}_i|}\left\|\sum_{k=0}^{j-1}\nabla f_{i\Pi_{i,e,k}}(y_{i,e,k}) - \nabla f_{i\Pi_{i,e,k}}(y_{i,e})\right\|^2$$

$$+ \frac{2\eta_l^2}{|\mathcal{D}_i|^2}\sum_{j=1}^{|\mathcal{D}_i|}\left\|j(1-a)(m - \nabla f_i(x)) + \sum_{k=0}^{j-1}\nabla f_{i\Pi_{i,e,k}}(y_{i,e})\right\|^2$$

$$\overset{(16),(23)}{\leq} \frac{2\eta_l^2 L^2}{|\mathcal{D}_i|^2}\sum_{j=1}^{|\mathcal{D}_i|}j\sum_{k=0}^{j-1}\|y_{i,e,k} - y_{i,e}\|^2$$

$$+ \frac{2\eta_l^2}{|\mathcal{D}_i|^2}\sum_{j=1}^{|\mathcal{D}_i|}\left\|j(1-a)(m - \nabla f_i(x)) + \sum_{k=0}^{j-1}\nabla f_{i\Pi_{i,e,k}}(y_{i,e})\right\|^2$$

$$\leq \eta_l^2 L^2\sum_{j=1}^{|\mathcal{D}_i|}\|y_{i,e,j-1} - y_{i,e}\|^2$$

$$+ \frac{2\eta_l^2}{|\mathcal{D}_i|^2}\sum_{j=1}^{|\mathcal{D}_i|}\left\|j(1-a)(m - \nabla f_i(x)) + \sum_{k=0}^{j-1}\nabla f_{i\Pi_{i,e,k}}(y_{i,e})\right\|^2.$$

By $\eta_l \leq \frac{1}{2L}$, we have

$$
\mathbf{E}_e \left[ \sum_{j=1}^{|\mathcal{D}_i|} \|y_{i,e,j-1} - y_{i,e}\|^2 \right] \leq \frac{8\eta_l^2}{3|\mathcal{D}_i|^2} \sum_{j=1}^{|\mathcal{D}_i|} j^2 \mathbf{E}_e \left[ \left\| (1-a)(m - \nabla f_i(x)) + \frac{1}{j} \sum_{k=0}^{j-1} \nabla f_{i\Pi_{i,e,k}}(y_{i,e}) \right\|^2 \right]
$$

$$
\overset{(21),(6)}{\leq} \frac{8\eta_l^2}{3|\mathcal{D}_i|^2} \sum_{j=1}^{|\mathcal{D}_i|} \left( \frac{(|\mathcal{D}_i| - j)j}{(|\mathcal{D}_i| - 1)} \sigma_i^2 + j^2 \|d_{i,e}\|^2 \right)
$$

$$
\leq 2\eta_l^2 \sigma_i^2 + \eta_l^2 |\mathcal{D}_i| \|d_{i,e}\|^2.
$$

Combining this bound with the previous results concludes the proof. $\qquad\square$

Next, we examine the variance of our update in each local epoch $d_{i,e}$.

**Lemma D.2.** *For the client update* (13)*, given Assumption 3.5 and assuming that one client is sampled, the following holds for any $a \in [0, 1]$, where $h \overset{def}{=} m - \nabla f(x)$:*

$$
\|d_{i,e} - \nabla f(y_{i,e})\|^2 \leq 3\|h\|^2 + 3\delta^2 \|y_{i,e} - x\|^2 + 3a^2 \|\nabla f_i(x) - \nabla f(x)\|^2. \tag{27}
$$

*Proof.* Starting from the client update (13), we can rewrite it as

$$
d_{i,e} - \nabla f(y_{i,e}) = (1-a)h
$$
$$
+ (\nabla f_i(y_{i,e}) - \nabla f_i(x) - \nabla f(y_{i,e}) + \nabla f(x))
$$
$$
+ a(\nabla f_i(x) - \nabla f(x)).
$$

We can use the relaxed triangle inequality Lemma B.6 to claim

$$
\|d_{i,e} - \nabla f(y_{i,e})\| = (1-a)^2 \|h\|^2
$$
$$
+ 3\|\nabla f_i(y_{i,e}) - \nabla f_i(x) - \nabla f(y_{i,e}) + \nabla f(x)\|^2
$$
$$
+ 3a^2 \|\nabla f_i(x) - \nabla f(x)\|^2.
$$
$$
\overset{(7)}{\leq} 3(1-a)^2 \|h\|^2 + 3\delta^2 \|y_{i,e} - x\|^2 + 3a^2 \|\nabla f_i(x) - \nabla f(x)\|^2.
$$

It remains to use that $(1-a)^2 \leq 1$ since $a \in [0, 1]$. $\qquad\square$

In the following lemma, we introduce a bound that controls the distance moved by a client in each step during the client update.

**Lemma D.3.** *For the client update updates* (13) *with $\eta_l \leq \min\{\frac{1}{4E\delta}, \frac{1}{2L}\}$ and given Assumptions 3.2, 3.4 and 3.5 with $P_i = 0$ for all $i \in [n]$ and one client is sampled, the following holds*

$$
\mathbf{E} \left[ \sum_{e=0}^{E-1} \|y_{i,e} - x\|^2 \right] \leq 16E^2 \eta_l^2 \sum_{e=0}^{E-1} \mathbf{E} \left[ \|\nabla f(y_{i,e})\|^2 \right]
$$
$$
+ 16E^3 \eta_l^2 \left( 3\|h\|^2 + 3a^2 \|\nabla f_i(x) - \nabla f(x)\|^2 + \eta_l^2 L^2 \frac{\sigma_i^2}{|\mathcal{D}_i|} \right). \tag{28}
$$

*Proof.* Starting from the FEDSHUFFLEMVR update (13),

$$\sum_{e=0}^{E-1} \mathbf{E}\left[\|y_{i,e} - x\|^2\right] = \eta_l^2 \sum_{e=0}^{E-1} \mathbf{E}\left[\left\|\sum_{l=0}^{e-1} \frac{1}{|\mathcal{D}_i|} \sum_{j=1}^{|\mathcal{D}_i|} d_{i,e,j-1}\right\|^2\right]$$

$$= \eta_l^2 \sum_{e=0}^{E-1} \mathbf{E}\left[\left\|\sum_{l=0}^{e-1}(d_{i,e} - \nabla f(y_{i,e}) + \nabla f(y_{i,e}) - (g_{i,e} - \nabla f_i(y_{i,e})))\right\|^2\right]$$

$$\overset{(23)}{\leq} 3\eta_l^2 \sum_{e=0}^{E-1}\sum_{l=0}^{e-1} e\Big(\mathbf{E}\left[\|d_{i,e} - \nabla f(y_{i,e})\|^2\right] + \mathbf{E}\left[\|\nabla f(y_{i,e})\|^2\right] + \mathbf{E}\left[\|g_{i,e} - \nabla f_i(y_{i,e})\|^2\right]\Big)$$

$$\leq 2E^2\eta_l^2 \sum_{e=0}^{E-1}\Big(\mathbf{E}\left[\|d_{i,e} - \nabla f(y_{i,e})\|^2\right] + \mathbf{E}\left[\|\nabla f(y_{i,e})\|^2\right] + \mathbf{E}\left[\|g_{i,e} - \nabla f_i(y_{i,e})\|^2\right]\Big)$$

$$\overset{(26)}{\leq} 2E^2\eta_l^2 \sum_{e=0}^{E-1}\Big(\mathbf{E}\left[\|d_{i,e} - \nabla f(y_{i,e})\|^2\right] + \mathbf{E}\left[\|\nabla f(y_{i,e})\|^2\right] + 2\eta_l^2 L^2 \frac{\sigma_i^2}{|\mathcal{D}_i|} + \eta_l^2 L^2 \mathbf{E}\left[\|d_{i,e}\|^2\right]\Big)$$

$$\overset{(23)}{\leq} 4E^2\eta_l^2 \sum_{e=0}^{E-1}\Big(\mathbf{E}\left[\|d_{i,e} - \nabla f(y_{i,e})\|^2\right] + \mathbf{E}\left[\|\nabla f(y_{i,e})\|^2\right] + \eta_l^2 L^2 \frac{\sigma_i^2}{|\mathcal{D}_i|}\Big),$$

where the last inequality also uses the step size bound $\eta_l \leq \frac{1}{2L}$. By exploiting (27), we can further bound

$$\mathbf{E}\left[\sum_{e=0}^{E-1}\|y_{i,e} - x\|^2\right] \leq 12E^2\eta_l^2\delta^2 \sum_{e=0}^{E-1} \mathbf{E}\left[\|y_{i,e} - x\|^2\right] + 4E^2\eta_l^2 \sum_{e=0}^{E-1} \mathbf{E}\left[\|\nabla f(y_{i,e})\|^2\right]$$

$$+ 12E^3\eta_l^2\|h\|^2 + 12E^3\eta_l^2 a^2\|\nabla f_i(x) - \nabla f(x)\|^2 + 4E^3\eta_l^4 L^2 \frac{\sigma_i^2}{|\mathcal{D}_i|}.$$

$\eta_l \leq \frac{1}{4E\delta}$ implies that $12E^2\eta_l^2\delta^2 \leq \frac{3}{4}$, therefore

$$\mathbf{E}\left[\sum_{e=0}^{E-1}\|y_{i,e} - x\|^2\right] \leq 16E^2\eta_l^2 \sum_{e=0}^{E-1} \mathbf{E}\left[\|\nabla f(y_{i,e})\|^2\right]$$

$$+ 16E^3\eta_l^2\left(3\|h\|^2 + 3a^2\|\nabla f_i(x) - \nabla f(x)\|^2 + \eta_l^2 L^2 \frac{\sigma_i^2}{|\mathcal{D}_i|}\right).$$

$\square$

We compute the error of the server momentum $m$ defined as $h \overset{\text{def}}{=} m - \nabla f(x)$. Its expected norm can be bounded as follows.

**Lemma D.4.** *For the momentum update (14), given that Assumptions 3.2-3.5 with $P_i = 0$ for all $i \in [n]$ and $B = 1$ hold and one client is sampled, the following holds for any $\eta_l \leq \left\{\frac{1}{4L} \frac{1}{40\delta E}\right\}$ and $1 \geq a \geq 1152E^2\delta^2\eta_l^2$,*

$$\mathbf{E}\left[\|h^+\|^2\right] \leq \left(1 - \frac{23a}{24}\right)\|h\|^2 + 3a^2\mathbf{E}\left[\|\nabla f_i(x) - \nabla f(x)\|^2\right]$$

$$+ 16\eta_l^4 E^2\delta^2 L^2 \frac{\sigma_i^2}{|\mathcal{D}_i|} + 8\eta_l^2\delta^2 E \sum_{e=0}^{E-1} \mathbf{E}\left[\|\nabla f(y_{i,e})\|^2\right]. \tag{29}$$

*Proof.* Starting from the momentum update (14),

$$h^+ = (1-a)h$$
$$+ (1-a)\big(\nabla f_i(x^+) - \nabla f_j(x)\big) - \nabla f(x^+) + \nabla f(x)\big)$$
$$+ a\big(\nabla f_i(x^+) - \nabla f(x)\big).$$

Now, the first term does not have any information from current round and hence is statistically independent of the rest of the terms. Further, the rest of the terms have mean 0. Hence, we can separate out the zero mean noise terms from the $h$ and then the relaxed triangle inequality Lemma B.6 to claim

$$
\begin{aligned}
\mathbf{E}\left[\|h^+\|^2\right] &= (1-a)^2 \mathbf{E}\left[\|h\|^2\right] \\
&\quad + 2(1-a)^2 \mathbf{E}\left[\left\|\left(\nabla f_i(^+x) - \nabla f_i(x)\right) - \nabla f(x^+) + \nabla f(x)\right\|^2\right] \\
&\quad + 2a^2 \mathbf{E}\left[\|\nabla f_i(x) - \nabla f(x)\|^2\right] \\
&\overset{(7)}{\le} (1-a)^2 \mathbf{E}\left[\|h\|^2\right] + 2(1-a)^2 \delta^2 \mathbf{E}\left[\|x^+ - x\|^2\right] + 2a^2 \mathbf{E}\left[\|\nabla f_i(x) - \nabla f(x)\|^2\right].
\end{aligned}
$$

Finally, note that $(1-a)^2 \le (1-a) \le 1$ for $a \in [0,1]$. Therefore,

$$
\mathbf{E}\left[\|h^+\|^2\right] \le (1-a)\mathbf{E}\left[\|h\|^2\right] + 2\delta^2 \mathbf{E}\left[\|x^+ - x\|^2\right] + 2a^2 \mathbf{E}\left[\|\nabla f_i(x) - \nabla f(x)\|^2\right].
$$

We can continue upper bounding $\mathbf{E}\left[\|x^+ - x\|^2\right]$.

$$
\begin{aligned}
\mathbf{E}\left[\|x^+ - x\|^2\right] &= \eta_l^2 \mathbf{E}\left[\left\|\sum_{e=0}^{E-1} \frac{1}{|\mathcal{D}_i|} \sum_{j-1}^{|\mathcal{D}_i|} d_{i,e,j-1}\right\|^2\right] \\
&= \eta_l^2 \mathbf{E}\left[\left\|\sum_{e=0}^{E-1} (d_{i,e} - \nabla f(y_{i,e})) - (g_{i,e} - \nabla f_i(y_{i,e}))\right\|^2\right] \\
&\overset{(23)}{\le} 2\eta_l^2 E \sum_{e=0}^{E-1} \mathbf{E}\left[\|d_{i,e} - \nabla f(y_{i,e})\|^2 + \|g_{i,e} - \nabla f_i(y_{i,e})\|^2\right] \\
&\overset{(26)}{\le} 2\eta_l^2 E \sum_{e=0}^{E-1} \mathbf{E}\left[\|d_{i,e} - \nabla f(y_{i,e})\|^2 + 2\eta_l^2 L^2 \frac{\sigma_i^2}{|\mathcal{D}_i|} + \eta_l^2 L^2 \|d_{i,e}\|^2\right] \\
&\overset{(23)}{\le} 4\eta_l^4 E^2 L^2 \frac{\sigma_i^2}{|\mathcal{D}_i|} + 4\eta_l^2 E \sum_{e=0}^{E-1} \mathbf{E}\left[\|d_{i,e} - \nabla f(y_{i,e})\|^2\right] + 4\eta_l^4 L^2 E \sum_{e=0}^{E-1} \mathbf{E}\left[\|\nabla f(y_{i,e})\|^2\right] \\
&\overset{(27)}{\le} 4\eta_l^4 E^2 L^2 \frac{\sigma_i^2}{|\mathcal{D}_i|} + 12\eta_l^2 E^2 \|h\|^2 + 12\eta_l^2 E^2 a^2 \|\nabla f_i(x) - \nabla f(x)\|^2 \\
&\quad + 12\eta_l^2 E \delta^2 \sum_{e=0}^{E-1} \mathbf{E}\left[\|y_{i,e} - x\|^2\right] + 4\eta_l^4 L^2 E \sum_{e=0}^{E-1} \mathbf{E}\left[\|\nabla f(y_{i,e})\|^2\right] \\
&\overset{(28)}{\le} 4\eta_l^4 E^2 L^2 \left(1 + 48\eta_l^2 E^2 \delta^2\right) \frac{\sigma_i^2}{|\mathcal{D}_i|} \\
&\quad + 12\eta_l^2 E^2 \left(1 + 48\eta_l^2 E^2 \delta^2\right) \|h\|^2 + 12\eta_l^2 E^2 a^2 \left(1 + 48\eta_l^2 E^2 \delta^2\right) \|\nabla f_i(x) - \nabla f(x)\|^2 \\
&\quad + 4\eta_l^4 E \left(L^2 + 48\delta^2 E^2\right) \sum_{e=0}^{E-1} \mathbf{E}\left[\|\nabla f(y_{i,e})\|^2\right] \\
&\overset{(28)}{\le} 8\eta_l^4 E^2 L^2 \frac{\sigma_i^2}{|\mathcal{D}_i|} + 24\eta_l^2 E^2 \|h\|^2 + 24\eta_l^2 E^2 a^2 \|\nabla f_i(x) - \nabla f(x)\|^2 \\
&\quad + 4\eta_l^2 E \sum_{e=0}^{E-1} \mathbf{E}\left[\|\nabla f(y_{i,e})\|^2\right],
\end{aligned}
$$

where the last inequality uses the upper bound on the step size $\eta_l$. Plugging this back to previous bound yields

$$\mathbf{E}\left[\|h^+\|^2\right] \le \left(1 - a + 48\eta_l^2 E^2 \delta^2\right)\mathbf{E}\left[\|h\|^2\right] + 3a^2\mathbf{E}\left[\|\nabla f_i(x) - \nabla f(x)\|^2\right]$$
$$+ 16\eta_l^4 E^2 \delta^2 L^2 \frac{\sigma_i^2}{|\mathcal{D}_i|} + 8\eta_l^2 \delta^2 E \sum_{e=0}^{E-1} \mathbf{E}\left[\|\nabla f(y_{i,e})\|^2\right].$$

The last step used the bound on the momentum parameter that $1 \ge a \ge 1152\eta_l^2\delta^2 E^2$. Note that $\eta_l \le \frac{1}{40\delta E}$ ensures that this set is non-empty. $\qquad\square$

Now we can compute the overall progress.

**Lemma D.5.** *For any client update step with step size $\eta_l \le \min\left\{\frac{1}{4L}, \frac{1}{40\delta E}\right\}$, $a \ge 1152\eta_l^2\delta^2 E^2$ and given that Assumptions 3.2-3.5 hold with $P_i = 0$ for all $i \in [n]$ and $B = 1$ and one client is sampled with probabilities $\{w_i\}$, we have*

$$\frac{1}{RE}\sum_{r=0}^{R-1}\sum_{e=0}^{E-1}\mathbf{E}\left[\|\nabla f(y_{i^r,e}^r)\|^2\right] \le \frac{5\Psi^0}{\eta_l RE} + 25\eta_l^2 L^2\sigma^2 + 255aG^2, \tag{30}$$

*where $\Psi^r \overset{def}{=} (f(x^r) - f^\star) + \frac{288\eta_l}{23a}E\|m^r - \nabla f(x^r)\|^2$ and $\sigma^2 \overset{def}{=} \frac{1}{|\mathcal{D}|}\sum_{i=1}^n \sigma_i^2$.*

*Proof.* The assumption that $f$ is $L$-smooth implies a quadratic upper bound (15).

$$\mathbf{E}_e\left[f(y_{i,e+1})\right] - f(y_{i,e}) \le -\eta_l\mathbf{E}_e\left[\left\langle \nabla f(y_{i,e}), \frac{1}{|\mathcal{D}_i|}\sum_{j=1}^{|\mathcal{D}_i|}d_{i,e,j-1}\right\rangle\right] + \frac{L\eta_l^2}{2}\mathbf{E}_e\left[\left\|\frac{1}{|\mathcal{D}_i|}\sum_{j=1}^{|\mathcal{D}_i|}d_{i,e,j-1}\right\|^2\right]$$

$$= -\frac{\eta_l}{2}\|\nabla f(y_{i,e})\|^2 - \frac{\eta_l}{2}(1 - \eta_l L)\mathbf{E}_e\left[\left\|\frac{1}{|\mathcal{D}_i|}\sum_{j=1}^{|\mathcal{D}_i|}d_{i,e,j-1}\right\|^2\right]$$

$$+ \frac{\eta_l}{2}\mathbf{E}_e\left[\left\|\frac{1}{|\mathcal{D}_i|}\sum_{j=1}^{|\mathcal{D}_i|}d_{i,e,j-1} - \nabla f(y_{i,e})\right\|^2\right]$$

$$\le -\frac{\eta_l}{2}\|\nabla f(y_{i,e})\|^2 + \mathbf{E}_e\left[\frac{\eta_l}{2}\|(d_{i,e} - \nabla f(y_{i,e})) - (g_{i,e} - \nabla f_i(y_{i,e}))\|^2\right]$$

$$\overset{(23)}{\le} -\frac{\eta_l}{2}\|\nabla f(y_{i,e})\|^2 + \eta_l\|d_{i,e} - \nabla f(y_{i,e})\|^2 + \eta_l\mathbf{E}_e\left[\|g_{i,e} - \nabla f_i(y_{i,e})\|^2\right]$$

$$\overset{(26)}{\le} -\frac{\eta_l}{2}\|\nabla f(y_{i,e})\|^2 + \eta_l\|d_{i,e} - \nabla f(y_{i,e})\|^2 + 2\eta_l^3 L^2\frac{\sigma_i^2}{|\mathcal{D}_i|} + \eta_l^3 L^2\|d_{i,e}\|^2$$

$$\overset{(23)}{\le} -\frac{\eta_l}{2}\left(1 - 4\eta_l^2 L^2\right)\|\nabla f(y_{i,e})\|^2 + 2\eta_l\|d_{i,e} - \nabla f(y_{i,e})\|^2 + 2\eta_l^3 L^2\frac{\sigma_i^2}{|\mathcal{D}_i|}$$

$$\overset{(27)}{\le} -\frac{\eta_l}{2}\left(1 - 4\eta_l^2 L^2\right)\|\nabla f(y_{i,e})\|^2 + 2\eta_l^3 L^2\frac{\sigma_i^2}{|\mathcal{D}_i|}$$

$$+ 6\eta_l\left(\|h\|^2 + \delta^2\|y_{i,e} - x\|^2 + a^2\|\nabla f_i(x) - \nabla f(x)\|^2\right).$$

The first equality used the fact that for any $a, b$: $2\langle a, b\rangle = \|a - b\|^2 - \|a\|^2 - \|b\|^2$. The second term in the first equality can be removed since $\eta_l \le \frac{1}{L}$.

Applying this inequality recursively with full expectation over the current communication round

$$\mathbf{E}\left[f(x^+)\right] - f(x) \leq -\frac{\eta_l}{2}\left(1 - 4\eta_l^2 L^2\right) \sum_{e=0}^{E-1} \|\nabla f(y_{i,e})\|^2 + 2\eta_l^3 L^2 E \frac{\sigma_i^2}{|\mathcal{D}_i|} + 6\eta_l \delta^2 \sum_{e=0}^{E-1} \|y_{i,e} - x\|^2$$

$$+ 6E\eta_l\left(\|h\|^2 + a^2\|\nabla f_i(x) - \nabla f(x)\|^2\right)$$

$$\overset{(28)}{\leq} -\frac{\eta_l}{2}\left(1 - 4\eta_l^2 L^2 - 192\eta_l^2\delta^2 E^2\right) \sum_{e=0}^{E-1} \|\nabla f(y_{i,e})\|^2 + 2\eta_l^3 L^2 E\left(1 + 48\eta_l^2 E^2\delta^2\right)\frac{\sigma_i^2}{|\mathcal{D}_i|}$$

$$+ 6E\eta_l\left(1 + 192\eta_l^2 E^2\delta^2\right)\left(\|h\|^2 + a^2\|\nabla f_i(x) - \nabla f(x)\|^2\right)$$

$$\leq -\frac{\eta_l}{4}\sum_{e=0}^{E-1} \|\nabla f(y_{i,e})\|^2 + 4\eta_l^3 L^2 E\frac{\sigma_i^2}{|\mathcal{D}_i|} + 12E\eta_l\left(\|h\|^2 + a^2\|\nabla f_i(x) - \nabla f(x)\|^2\right).$$

Adding an extra term $12E\frac{24\eta_l}{23a}\|h^+\|^2$ term results to

$$\mathbf{E}\left[f(x^+) + 12E\frac{24\eta_l}{23a}\|h^+\|^2\right] - f(x) \overset{(29)}{\leq} -\frac{\eta_l}{4}\left(1 - 104\frac{\eta_l^2 E^2\delta^2}{a}\right) \sum_{e=0}^{E-1} \|\nabla f(y_{i,e})\|^2 + \left(4 + 208\frac{\eta_l^2 E^2\delta^2}{a}\right)E\eta_l^3 L^2\frac{\sigma_i^2}{|\mathcal{D}_i|}$$

$$+ 12E\frac{24\eta_l}{23a}\|h\|^2 + \left(12\eta_l a^2 + 39\eta_l a\right)E\|\nabla f_i(x) - \nabla f(x)\|^2$$

$$\leq -\frac{\eta_l}{5}\sum_{e=0}^{E-1} \|\nabla f(y_{i,e})\|^2 + 5E\eta_l^3 L^2\frac{\sigma_i^2}{|\mathcal{D}_i|}$$

$$+ 12E\frac{24\eta_l}{23a}\|h\|^2 + 51\eta_l aE\|\nabla f_i(x) - \nabla f(x)\|^2.$$

Taking the full expectation including the client selection step yields

$$\frac{5}{\eta_l E}\mathbf{E}\left[\Psi^+ - \Psi\right] \overset{(4)}{\leq} -\frac{1}{E}\sum_{e=0}^{E-1}\mathbf{E}\left[\|\nabla f(y_{i,e})\|^2\right] + 25\eta_l^2 L^2\sigma^2 + 255aG^2.$$

Applying this inequality recursively with the fact $\Psi \geq 0$ leads to the desired result. $\qquad\square$

Equipped with Lemma D.5, we have

$$\frac{1}{RE}\sum_{r=0}^{R-1}\sum_{e=0}^{E-1}\mathbf{E}\left[\|\nabla f(y_{i^r,e}^r)\|^2\right] \leq \frac{5\Psi^0}{\eta_l RE} + 25\eta_l^2 L^2\sigma^2 + 255aG^2,$$

where $\Psi^0 \overset{\text{def}}{=} (f(x^0) - f^\star) + \frac{288\eta_l}{23a}E\|m^0 - \nabla f(x^0)\|^2$.

Further, we need to control $\|m^0 - \nabla f(x^0)\|^2$. Cutkosky & Orabona (2019) show that by using time-varying step sizes, it is possible to directly control this error. Alternatively, Tran-Dinh et al. (2019) use a large initial accumulation for the momentum term. For the sake of simplicity, we will follow the latter approach similarly to the work of Karimireddy et al. (2020). Extension to the time-varying step size case is straightforward. Suppose that we run the algorithm for $2R$ communication rounds wherein for the first $R$ rounds, we simply compute $m^0 = \frac{1}{R}\sum_{t=1}^{R}\nabla f_{i^r}(x^0)$. With this, we have $\|m^0 - \nabla f(x^0)\|^2 \leq \frac{G^2}{R}$. Thus, we have for the first round $r = 0$

$$\Psi^0 = (f(x^0) - f^\star) + \frac{288\eta_l}{23a}E\|m^0 - \nabla f(x^0)\|^2 \leq (f(x^0) - f^\star) + \frac{288\eta_l EG^2}{23aR}.$$

Together, this gives

$$\frac{1}{RE}\sum_{r=0}^{R-1}\sum_{e=0}^{E-1}\mathbf{E}\left[\|\nabla f(y_{i^r,e}^r)\|^2\right] \leq \frac{5(f(x^0) - f^\star)}{\eta_l ER} + \frac{63G^2}{aR^2} + 25\eta_l^2 L^2\sigma^2 + 293760G^2.$$

The above equation holds for any choice of $\eta \leq \min\left(\frac{1}{4L}, \frac{1}{40\delta K}\right)$ and momentum parameter $a \geq 1152E^2\delta^2\eta_l^2$. Set the momentum parameter as

$$a = \max\left(1152E^2\delta^2\eta_l^2, \frac{1}{T}\right)$$

With this choice, we can simplify the rate of convergence as

$$\frac{5(f(x^0) - f^\star)}{\eta_l E R} + \frac{318G^2}{R} + \eta_l^2\left(25L^2\sigma^2 + 293760E^2\delta^2G^2\right).$$

Now let us pick

$$\eta_l = \min\left\{\frac{1}{4L}, \frac{1}{40\delta E}, \left(\frac{f(x^0) - f^\star}{E^3 R(58752\delta^2 G^2 + 5\sigma^2\frac{L^2}{E^2})}\right)^{1/3}\right\}.$$

For this combination of step size $\eta_l$ and $a$, the rate simplifies to

$$\frac{318G^2}{R} + 390\left(\frac{(f(x^0) - f^\star)\left(\delta^2 G^2 + \sigma^2\frac{L^2}{E^2}\right)}{R^2}\right)^{1/3} + \frac{20(L + 10\delta E)(f(x^0) - f^\star)}{E R}.$$

Since $E \geq \frac{L}{\delta}$ then $\frac{L^2}{E^2} \leq \delta^2$ and $L \leq E\delta$, which concludes the proof.

## D.1 Discussion on Theoretical Assumptions

Note that in Theorem 5.2, we assume that $P_i = 0$ for all $i \in [n]$ and $B = 1$. These are not necessary assumptions, but we rather introduce these more restrictive requirements to match the assumptions that were used to obtain previous results. Furthermore, note that we require only one client to be sampled in each round. Unfortunately, our results cannot be simply extended to the case with multiple clients sampled in each round. The problem is the aggregation step (line 15 in Algorithm 1) which involves averaging. For our analysis and all the previous works that show the improvement over non-local methods in the non-convex setting to work, one would need to assume that the average model performs not worse than the average output of the sampled client. Such a property was empirically observed in (McMahan et al., 2017); thus, is reasonable to assume. Another possibility is to replace the averaging by randomly selecting model from one of the sampled clients. We note that while the second option might be preferable in theory as it does not require an extra assumption, it might affect the client privacy and therefore not applicable in practice.

---

**Algorithm 4** FEDSHUFFLEGEN $(c, \tilde{w}, q)$

---

1: **Input:** initial global model $x^0$, global and local step sizes $\eta_g^r$, $\eta_l^r$, proper distribution $\mathcal{S}$
2: **for** each round $r = 0, \ldots, R-1$ **do**
3:      server broadcasts $x^r$ to all clients $i \in \mathcal{S}^r \sim S$
4:      **for** each client $i \in \mathcal{S}^r$ (in parallel) **do**
5:          initialize local model $y_{i,0,0}^r \leftarrow x^r$
6:          **for** $e = 1, \ldots, E_i$ **do**
7:              Sample permutation $\{\Pi_{i,e,0}^r, \ldots, \Pi_{i,e,|\mathcal{D}_i|-1}^k\}$ of $\{1, \ldots, |\mathcal{D}_i|\}$
8:              **for** $j = 1, \ldots, |\mathcal{D}_i|$ **do**
9:                  local step size $\eta_{l,i}^r = \eta_l^r / c_i$
10:                update $y_{i,e,j}^r = y_{i,e,j-1}^r - \eta_{l,i}^r \nabla f_{i\Pi_{i,e,j-1}^r}(y_{i,e,j-1}^r)$
11:              **end for**
12:              $y_{i,e+1,0}^r = y_{i,e,|\mathcal{D}_i|}^r$
13:          **end for**
14:          send $\Delta_i^r = y_{i,E_i,|\mathcal{D}_i|}^r - x^r$ to server
15:      **end for**
16:      server computes $\Delta^r = \sum_{i \in \mathcal{S}^r} \frac{\tilde{w}_i}{q_i^{S^r}} \Delta_i^r$
17:      server updates global model $x^{r+1} = x^r - \eta_g^r \Delta^r$
18: **end for**

---

## E    FEDSHUFFLEGEN: **General Shuffling Method**

In this section, we introduce and analyze FEDSHUFFLEGEN. FEDSHUFFLEGEN is a class of algorithms parametrized by local and global step sizes $\{\eta_l^r\}_{r=0}^{R-1}$ and $\{\eta_g^r\}_{r=0}^{R-1}$, step size normalization $c = \{c_i\}_{i=1}^n$, where each of its element $c_i$ is the step size normalization for client $i$, aggregation weights $\{\tilde{w}_i\}_{i=1}^n$ and the aggregation normalization constants $\left\{\{q_i^{S^r}\}_{i \in S, S^r \sim \mathcal{S}}\right\}_{r=0}^{R-1}$. Note that we allow the aggregation normalization constants to be non-deterministic. To our knowledge, such results are impossible to obtain with any known analysis, despite this being standard practice for FEDAVG (where the update of each client is scaled by $w_i / (\sum_{j \in S} w_j)$), e.g., the default way to aggregate in Tensorflow Federated and other frameworks. We include the pseudocode in Algorithm 4. Later in this section, we show that FEDSHUFFLEGEN with its parametrization covers a wide variety of the FL algorithms in the form as they are implemented in practice.

### E.1   Convergence Analysis

We start with the convergence analysis of FEDSHUFFLEGEN. We first introduce the function $\hat{f}(x)$, which we show is the true objective optimized by Algorithm 4:

$$\hat{f}(x) \overset{\text{def}}{=} \sum_{i \in [n]} \hat{w}_i f_i(x), \text{ where } \hat{w}_i \overset{\text{def}}{=} \frac{\tilde{w}_i |\mathcal{D}_i| E_i}{W q_i c_i} \text{ with } \frac{1}{q_i} = \mathbf{E}_{\mathcal{S}}\left[\frac{1}{q_i^S} 1_{i \in \mathcal{S}}\right] \text{ and } W = \sum_{i \in [n]} \frac{\tilde{w}_i |\mathcal{D}_i| E_i}{q_i c_i}. \quad (31)$$

We denote $\hat{x}^\star$ to be an optimal solution of $\hat{f}$ and $\hat{f}^\star$ to be its functional value.

Furthermore, FEDSHUFFLEGEN allows the normalization to depend on the sampled clients (line 16 of Algorithm 4, which requires more general notion of variance. For this purpose, we introduce a $n \times n$ matrix $\mathbf{H}$, where its elements $\mathbf{H}_{i,j} = \mathbf{E}_{\mathcal{S}}\left[\frac{q_i q_j}{q_i^S q_j^S} 1_{i,j \in \mathcal{S}}\right]$ and $h$ is its diagonal with $h_i = \mathbf{E}_{\mathcal{S}}\left[\frac{q_i^2}{(q_i^S)^2} 1_{i \in \mathcal{S}}\right]$. We further define vector $s \in \mathbb{R}^n$ to be such vector that it holds

$$\mathbf{H} - ee^\top \preceq \mathbf{Diag}(h_1 s_2, h_2 s_2, \ldots, h_n s_n),$$

where $e \in \mathbb{R}^n$ is all ones vector. Note that this is not an assumption as such upper-bound always exists due to Gershgorin circle theorem, e.g., for $s_i = n$ for all $i \in [n]$ this holds. Equipped with these extra quantities, we proceed with the convergence guarantees for (strongly-)convex and non-convex functions.

**Theorem E.1.** *Suppose that the Assumptions 3.2-3.4 holds. Then, in each of the following cases, there exist weights $\{v_r\}$ and local step sizes $\eta_l^r \stackrel{def}{=} \eta_l$ such that for any $\eta_g^r \stackrel{def}{=} \eta_g \geq 1$ the output of FEDSHUFFLEGEN (Algorithm 4)*

$$\bar{x}^R = x^r \text{ with probability } \frac{v_r}{\sum_\tau v_\tau} \text{ for } r \in \{0, \dots, R-1\}. \tag{32}$$

*satisfies*

- **Strongly convex:** $\{f_{ij}\}$ *satisfy* (2) *for* $\mu > 0$, $\eta_l \leq \frac{1}{4\beta L \eta_g}$, $R \geq \frac{4\beta L}{\mu}$ *and* $c_i \geq E_i|\mathcal{D}_i|$ *then*

$$\mathbf{E}\left[\hat{f}(\bar{x}^R) - \hat{f}(\hat{x}^\star)\right] \leq \tilde{\mathcal{O}}\left(\frac{M_1 G^2}{\mu R} + \frac{(1+P^2)G^2 + \sigma^2}{\mu^2 R^2 \eta_g^2} + \mu D^2 \exp(-\frac{\mu}{8\beta L}R)\right),$$

- **General convex:** $\{f_{ij}\}$ *satisfy* (2) *for* $\mu = 0$, $\eta_l \leq \frac{1}{4\beta L \eta_g}$, $R \geq 1$ *and* $c_i \geq E_i|\mathcal{D}_i|$ *then*

$$\mathbf{E}\left[\hat{f}(\bar{x}^R) - \hat{f}(\hat{x}^\star)\right] \leq \mathcal{O}\left(\frac{\sqrt{DM_1}G}{\sqrt{R}} + \frac{D^{2/3}((1+P^2)G^2 + \sigma^2)^{1/3}}{R^{2/3}\eta_g^{2/3}} + \frac{LD\beta}{R}\right),$$

- **Non-convex:** $\eta_l \leq \frac{1}{4\beta L \eta_g}$, $R \geq 1$ *and* $c_i \geq E_i|\mathcal{D}_i|$ *then*

$$\mathbf{E}\left[\|\nabla \hat{f}(\bar{x}^R)\|^2\right] \leq \mathcal{O}\left(\frac{\sqrt{FM_1 L}G}{\sqrt{R}} + \frac{F^{2/3}L^{1/3}((1+P^2)G^2 + \sigma^2)^{1/3}}{R^{2/3}\eta_g^{2/3}} + \frac{LF\beta}{R}\right),$$

*where* $\beta \stackrel{def}{=} 1 + M_2 + (1+P)B + M_1 B^2$, $P^2 \stackrel{def}{=} \max_{i \in [n]} \frac{P_i^2}{3|\mathcal{D}_i|E_i^2}$, $\sigma^2 \stackrel{def}{=} \sum_{i \in [n]} \hat{w}_i \frac{\sigma_i^2}{3|\mathcal{D}_i|E_i^2}$, $M_1 \stackrel{def}{=} \max_{i \in [n]}\{h_i s_i \hat{w}_i\}$, $M_2 \stackrel{def}{=} \left(\sum_{i \in [n]} E_i|\mathcal{D}_i|\right) \max_{i \in [n]}\left\{\frac{\hat{w}_i}{W q_i c_i}\right\}$, $D \stackrel{def}{=} \|x^0 - x^\star\|^2$, *and* $F \stackrel{def}{=} f(x^0) - f^\star$.

The above rate is a generalization of results presented in Section 5 and we refer the reader to this section for the detailed discussion.

## E.2 Special Cases

In this section, we show how FEDSHUFFLEGEN captures not only our proposed FEDSHUFFLE, but also both FEDAVG and FEDNOVA with heterogeneous data, arbitrary client sampling, random reshuffling, non-identical local steps, stateless clients and server and local step sizes. To the best of our knowledge, our work is the first to provide such comprehensive analysis.

We start with the FEDAVG algorithm. Algorithm 2 contains a detailed pseudocode of how it is usually implemented in practice. It is easy to verify that FEDSHUFFLEGEN covers this implementation with the following selection of parameters

$$\hat{w} = w, \quad c = \max_{i \in [n]}\{E_i|\mathcal{D}_i|\}e \quad \text{and} \quad q_i^{S^r} = \frac{c}{n} \sum_{j \in S^r} w_j.$$

Unfortunately, it is not guaranteed that $\hat{f} = f$. For instance, when all clients participate in each round, each client runs $E$ local epochs and weight of each client is proportional to its dataset size, i.e. $w_i = |\mathcal{D}_i|/|\mathcal{D}|$, then $\hat{w}_i = |\mathcal{D}_i|^2/\sum_{j \in [n]} |\mathcal{D}_j|^2$, which means that the objective that we end up optimizing favours clients with larger amount of data. This inconsistency is partially removed when client sampling is introduced, since the clients with the larger number of data points are in average normalized with larger numbers, i.e. $q_i$ grows. The inconsistency is only fully removed when only one client is sampled uniformly at random, which does not

reflect standard FL systems where a reasonably large number of clients is selected to participate in each round.

FEDNOVA tackles the objective inconsistency issue by re-weighting the local updates during the aggregation step. For the same example as before with the full participation, FEDNOVA uses the same parameters as FEDAVG with a difference that $\hat{w} = e/n$ that leads to $\hat{f} = f$. Apart from full participation, Wang et al. (2020) analyze a client sampling scheme with one client per communication round sampled with probability $|\mathcal{D}_i|/|\mathcal{D}|$.

In our work, apart from providing a general theory for shuffling methods in FL, we introduce FEDSHUFFLE that is motivated by insights obtained from our general theory provided in Theorem E.1. FEDSHUFFLE preserves the original objective aggregation weights, i.e., $\hat{w} = w$, it uses unbiased normalization weights during the aggregation step and it sets $c_i = E_i|\mathcal{D}_i|$. Such parameters choice guarantees that $\hat{f} = f$.

*Remark* E.2 (FEDSHUFFLE is better than FEDNOVA). FEDSHUFFLE preserves the original objective aggregation weights, i.e., $\hat{w} = w$, and it uses unbiased normalization weights during the aggregation step. In addition, contrary to FEDAVG and FEDNOVA, FEDSHUFFLE uses $c_i = E_i|\mathcal{D}_i|$ (FEDAVG and FEDNOVA require $c_i = \max_{j\in[n]}\{E_j|\mathcal{D}_j|\}$), which implies that it allows larger local step sizes than both FEDAVG and FEDNOVA, but it does not introduce any inconsistency as it still holds that $\hat{f} = f$ as for FEDNOVA. Differently from FEDNOVA, FEDSHUFFLE does not degrade the local progress made by clients by diminishing their contribution via decreased weights in the aggregation step. Still, it achieves the objective consistency by balancing the clients' progress at each step while preserving the worst-case convergence rate that can't be further improved Woodworth et al. (2020); Arjevani & Shamir (2015).

### E.3 Proof of Theorem E.1

By $\mathbf{E}_r[\cdot]$, we denote the expectation conditioned on the all history prior to communication round $r$.

**Lemma E.3. (one round progress)** *Suppose Assumptions 3.1 – 3.4 hold. For any constant step sizes* $\eta_l^r \stackrel{def}{=} \eta_l$ *and* $\eta_l^r \stackrel{def}{=} \eta_l$ *satisfying* $\eta_l \leq \frac{1}{(1+M_2+M_1B^2)4L\eta_g}$ *and effective step size* $\tilde{\eta} \stackrel{def}{=} W\eta_g\eta_l$, *the updates of* FEDSHUFFLEMVR *satisfy*

$$\mathbf{E}\left[\|x^r - \hat{x}^\star\|^2\right] \leq \left(1 - \frac{\mu\tilde{\eta}}{2}\right)\mathbf{E}\left[\|x^{r-1} - \hat{x}^\star\|^2\right] - \tilde{\eta}\mathbf{E}_{r-1}\left[\hat{f}(x^{r-1}) - \hat{f}^\star\right] + 3L\tilde{\eta}\xi^r + 2\tilde{\eta}^2 M_1 G^2,$$

*where* $\xi_r$ *is the drift caused by the local updates on the clients defined to be*

$$\xi^r \stackrel{def}{=} \frac{1}{W}\sum_{i=1}^n\sum_{e=1}^{E_i}\sum_{j=1}^{|\mathcal{D}_i|}\frac{\tilde{w}_i}{q_ic_i}\mathbf{E}_{r-1}\left[\|y_{i,e,j-1}^r - x^{r-1}\|^2\right]$$

$M_1 \stackrel{def}{=} \max_{i\in[n]}\{h_is_i\hat{w}_i\}$ *and* $M_2 \stackrel{def}{=} \left(\sum_{i\in[n]}E_i|\mathcal{D}_i|\right)\max_{i\in[n]}\left\{\frac{\tilde{w}_i}{Wq_ic_i}\right\}$.

*Proof.* For a better readability of the proofs in one round progress, we drop the superscript that represents the current completed communication round $r-1$.

By the definition in Algorithm 4, the update $\Delta$ can be written as

$$\Delta = -\eta_g\sum_{i\in\mathcal{S}}\frac{w_i}{q_i}\Delta_i = -\frac{\tilde{\eta}}{W}\sum_{i\in\mathcal{S}}\sum_{e=1}^{E_i}\sum_{j=1}^{|\mathcal{D}_i|}\frac{\tilde{w}_i}{q_i^\mathcal{S}c_i}\nabla f_{i\Pi_{i,e,j-1}}(y_{i,e,j-1}).$$

We adopt the convention that summation $\sum_{i\in\mathcal{M},e,j}$ ($\mathcal{M}$ is either $[n]$ or $\mathcal{S}$) refers to the summations $\sum_{i\in\mathcal{M}}\sum_{e=1}^{E_i}\sum_{j=1}^{|\mathcal{D}_i|}$ unless otherwise stated. Furthermore, we denote $g_{i,e,j} \stackrel{def}{=} \nabla f_{i\Pi_{i,e,j-1}}(y_{i,e,j-1})$. Using above, we proceed as

$$\mathbf{E}_{r-1}\left[\|x+\Delta-\hat{x}^\star\|^2\right] = \|x-\hat{x}^\star\|^2 \underbrace{-2\mathbf{E}_{r-1}\left[\frac{\tilde{\eta}}{W}\sum_{i\in\mathcal{S},e,j}\frac{\tilde{w}_i}{q_i^\mathcal{S}c_i}\langle g_{i,e,j}, x-\hat{x}^\star\rangle\right]}_{\mathcal{A}_1} + \underbrace{\tilde{\eta}^2\mathbf{E}_{r-1}\left[\left\|\frac{1}{W}\sum_{i\in\mathcal{S},e,j}\frac{\tilde{w}_i}{q_i^\mathcal{S}c_i}g_{i,e,j}\right\|^2\right]}_{\mathcal{A}_2}.$$

To bound the term $\mathcal{A}_1$, we apply Lemma B.1 to each term of the summation with $h = f_{ij}$, $x = y_{i,e,j-1}$, $y = \hat{x}^\star$, and $z = x$. Therefore,

$$
\mathcal{A}_1 = -\mathbf{E}_{r-1}\left[2\frac{\tilde{\eta}}{W}\sum_{i \in \mathcal{S},e,j}\frac{\tilde{w}_i}{q_i^{\mathcal{S}}c_i}\langle g_{i,e,j}, x - \hat{x}^\star\rangle\right]
$$

$$
\leq \mathbf{E}_{r-1}\left[2\frac{\tilde{\eta}}{W}\sum_{i \in \mathcal{S},e,j}\frac{\tilde{w}_i}{q_i^{\mathcal{S}}c_i}\left(f_{i\Pi_{i,e,j-1}}(\hat{x}^\star) - f_{i\Pi_{i,e,j-1}}(x) + L\|y_{i,e,j-1} - x\|^2 - \frac{\mu}{4}\|x - \hat{x}^\star\|^2\right)\right]
$$

$$
= -2\tilde{\eta}\left(\hat{f}(x) - \hat{f}^\star + \frac{\mu}{4}\|x - \hat{x}^\star\|^2\right) + 2L\tilde{\eta}\xi.
$$

For the second term $\mathcal{A}_2$, we have

$$
\mathcal{A}_2 = \tilde{\eta}^2\mathbf{E}_{r-1}\left[\left\|\frac{1}{W}\sum_{i \in \mathcal{S},e,j}\frac{\tilde{w}_i}{q_i^{\mathcal{S}}c_i}g_{i,e,j}\right\|^2\right]
$$

$$
\overset{(17)}{\leq} \tilde{\eta}^2\mathbf{E}_{r-1}\left[\left\|\frac{1}{W}\sum_{i \in \mathcal{S},e,j}\frac{w_i}{q_i^{\mathcal{S}}c_i}g_{i,e,j} - \frac{1}{W}\sum_{i \in [n],e,j}\frac{\tilde{w}_i}{q_ic_i}g_{i,e,j}\right\|^2 + \left\|\frac{1}{W}\sum_{i \in [n],e,j}\frac{\tilde{w}_i}{q_ic_i}g_{i,e,j}\right\|^2\right]
$$

$$
\overset{(34)}{\leq} \frac{\tilde{\eta}^2}{W^2}\mathbf{E}_{r-1}\left[\sum_{i \in [n]}\frac{h_is_i\tilde{w}_i^2}{q_i^2c_i^2}\left\|\sum_{e,j}g_{i,e,j}\right\|^2 + \left\|\sum_{i \in [n],e,j}\frac{\tilde{w}_i}{q_ic_i}g_{i,e,j}\right\|^2\right]
$$

$$
\overset{(23)}{\leq} \frac{2\tilde{\eta}^2}{W^2}\mathbf{E}_{r-1}\left[\sum_{i \in [n]}\frac{h_is_i\tilde{w}_i^2}{q_i^2c_i^2}\left\|\sum_{e,j}g_{i,e,j} - \nabla f_{i\Pi_{i,e,j-1}}(x)\right\|^2\right] + 2\frac{\tilde{\eta}^2}{W^2}\sum_{i \in [n]}\frac{h_is_i\tilde{w}_i^2E_i^2|\mathcal{D}_i|^2}{q_i^2c_i^2}\|\nabla f_i(x)\|^2
$$

$$
+ \frac{2\tilde{\eta}^2}{W^2}\mathbf{E}_{r-1}\left[\left\|\sum_{i \in [n],e,j}\frac{\tilde{w}_i}{q_ic_i}\{g_{i,e,j} - \nabla f_{i\Pi_{i,e,j-1}}(x)\}\right\|^2\right] + 2\tilde{\eta}^2\left\|\nabla\hat{f}(x)\right\|^2
$$

$$
\overset{(23)}{\leq} \frac{2\tilde{\eta}^2}{W^2}\mathbf{E}_{r-1}\left[\sum_{i \in [n],e,j}\frac{h_is_i\tilde{w}_i^2E_i|\mathcal{D}_i|}{q_i^2c_i^2}\left\|g_{i,e,j} - \nabla f_{i\Pi_{i,e,j-1}}(x)\right\|^2\right] + 2\frac{\tilde{\eta}^2}{W^2}\sum_{i \in [n]}\frac{h_is_i\tilde{w}_i^2E_i^2|\mathcal{D}_i|^2}{q_i^2c_i^2}\|\nabla f_i(x)\|^2
$$

$$
+ \frac{2\tilde{\eta}^2}{W^2}\sum_{i \in [n],e,j}\frac{\tilde{w}_i^2\left(\sum_{i \in [n]}E_i|\mathcal{D}_i|\right)}{q_i^2c_i^2}\mathbf{E}_{r-1}\left[\left\|g_{i,e,j} - \nabla f_{i\Pi_{i,e,j-1}}(x)\right\|^2\right] + 2\tilde{\eta}^2\left\|\nabla\hat{f}(x)\right\|^2
$$

$$
\overset{(3)}{\leq} 2M_1\tilde{\eta}^2L^2\xi + 2\tilde{\eta}^2M_1\sum_{i \in [n]}\hat{w}_i\|\nabla f_i(x)\|^2 + 2\tilde{\eta}^2L^2\xi M_2 + 2\tilde{\eta}^2\left\|\nabla\hat{f}(x)\right\|^2
$$

$$
\overset{(4)}{\leq} 2(M_2 + M_1)\tilde{\eta}^2L^2\xi + 2\tilde{\eta}^2M_1G^2 + 2\tilde{\eta}^2\left(1 + M_1B^2\right)\left\|\nabla\hat{f}(x)\right\|^2
$$

$$
\overset{(16)}{\leq} 2(M_2 + M_1)\tilde{\eta}^2L^2\xi + 2\tilde{\eta}^2M_1G^2 + 4L\tilde{\eta}^2\left(1 + M_1B^2\right)(\hat{f}(x) - \hat{f}^\star).
$$

By plugging back the bounds on $\mathcal{A}_1$ and $\mathcal{A}_2$,

$$
\mathbf{E}_{r-1}\left[\|x + \Delta - \hat{x}^\star\|^2\right] \leq \left(1 - \frac{\mu\tilde{\eta}}{2}\right)\|x - \hat{x}^\star\|^2 - (2\tilde{\eta} - 4L\tilde{\eta}^2(M_1B^2 + 1))(\hat{f}(x) - \hat{f}^\star)
$$

$$
+ (1 + (M_1 + M_2)\tilde{\eta}L)2L\tilde{\eta}\xi + 2\tilde{\eta}^2M_1G^2.
$$

The lemma now follows by observing that $4L\tilde{\eta}(M_1B^2 + M_2 + 1) \leq 1$ and that $B \geq 1$. $\qquad\square$

**Lemma E.4. (bounded drift)** *Suppose Assumptions 3.2 – 3.4 hold. Then the updates of* FEDSHUFFLEMVR *for any step size satisfying* $\eta_l \leq \frac{1}{(1+M_2+(P+1)B+M_1B^2))4L\eta_g}$ *and* $c_i \geq E_i|\mathcal{D}_i|$ *for all* $i \in [n]$ *have bounded drift:*

$$3L\tilde{\eta}\xi_r \leq \frac{9}{10}\tilde{\eta}(\hat{f}(x^r) - \hat{f}^\star) + 9\frac{\tilde{\eta}^3}{\eta_g^2}\left((1+P^2)G^2 + \sigma^2\right),$$

*where* $P^2 \stackrel{def}{=} \max_{i \in [n]} \frac{P_i^2}{3|\mathcal{D}_i|E_i^2}$, $\sigma^2 \stackrel{def}{=} \sum_{i \in [n]} \hat{w}_i \frac{\sigma_i^2}{3|\mathcal{D}_i|E_i^2}$ *and* $M_1 \stackrel{def}{=} \max_{i \in [n]}\{h_i s_i \hat{w}_i\}$.

*Proof.* We adopt the same convention as for the previous proof, i.e., dropping superscripts, simplifying sum notation and having $g_{i,e,j} \stackrel{def}{=} \nabla f_{i\Pi_{i,e,j-1}}(y_{i,e,j-1})$. Therefore,

$$\xi = \frac{1}{W}\sum_{i \in [n],i,j}\frac{\tilde{w}_i}{q_i c_i}\mathbf{E}_{r-1}\left[\|y_{i,e,j-1}^r - x^{r-1}\|^2\right]$$

$$= \frac{\eta_l^2}{W}\sum_{i \in [n],e,j}\frac{\tilde{w}_i}{q_i c_i^2}\mathbf{E}_{r-1}\left[\left\|\sum_{l=0}^{e-1}\sum_{c=1}^{\mathcal{D}_i}g_{i,l,c} + \sum_{c=1}^{j}g_{i,e,c}\right\|^2\right]$$

$$\stackrel{(22)}{\leq} \frac{2\eta_l^2}{W}\sum_{i \in [n],e,j}\frac{\tilde{w}_i}{q_i c_i^3}\mathbf{E}_{r-1}\left[\left\|\sum_{l=0}^{e-1}\sum_{c=1}^{\mathcal{D}_i}g_{i,l,c} - (e-1)|\mathcal{D}_i|\nabla f_i(x) + \sum_{c=1}^{j}g_{i,e,c} - \nabla f_{i\Pi_{i,e,c-1}}(x)\right\|^2\right]$$

$$+ \frac{2\eta_l^2}{W}\sum_{i \in [n],e,j}\frac{\tilde{w}_i}{q_i c_i^3}\mathbf{E}_{r-1}\left[\left\|(e-1)|\mathcal{D}_i|\nabla f_i(x) + \sum_{c=1}^{j}\nabla f_{i\Pi_{i,e,c-1}}(x)\right\|^2\right]$$

Next, we upper bound each term separately. For the first term, we have

$$\frac{2\eta_l^2}{W}\sum_{i \in [n],e,j}\frac{\tilde{w}_i}{q_i c_i^3}\mathbf{E}_{r-1}\left[\left\|\sum_{l=0}^{e-1}\sum_{c=1}^{\mathcal{D}_i}g_{i,l,c} - (e-1)|\mathcal{D}_i|\nabla f_i(x) + \sum_{c=1}^{j}g_{i,e,c} - \nabla f_{i\Pi_{i,e,c-1}}(x)\right\|^2\right]$$

$$\stackrel{(23)}{\leq} \frac{2\eta_l^2}{W}\sum_{i \in [n],e,j}\frac{\tilde{w}_i((e-1)|\mathcal{D}_i| + j)}{q_i c_i^3}\mathbf{E}_{r-1}\left[\left(\sum_{l=0}^{e-1}\sum_{c=1}^{\mathcal{D}_i}\|g_{i,l,c} - \nabla f_{i\Pi_{i,e,j-1}}(x)\|^2 + \sum_{c=1}^{j}\|g_{i,e,c} - \nabla f_{i\Pi_{i,e,c-1}}(x)\|^2\right)\right]$$

$$\stackrel{(3)}{\leq} 2\eta_l^2 L^2 \xi$$

For the second term,

$$\frac{2\eta_l^2}{W}\sum_{i\in[n],e,j}\frac{\tilde{w}_i}{q_i c_i^3}\mathbf{E}_{r-1}\left[\left\|(e-1)|\mathcal{D}_i|\nabla f_i(x)+\sum_{c=1}^{j}\nabla f_{i\Pi_{i,e,c-1}}(x)\right\|^2\right]$$

$$\overset{(17)}{\leq}\frac{2\eta_l^2}{W}\sum_{i\in[n],e,j}\frac{\tilde{w}_i}{q_i c_i^3}\mathbf{E}_{r-1}\left[\|((e-1)|\mathcal{D}_i|+j)\nabla f_i(x)\|^2+\left\|\sum_{c=1}^{j}\nabla f_{i\Pi_{i,e,c-1}}(x)-j\nabla f_i(x)\right\|^2\right]$$

$$\overset{(21)}{\leq}\frac{2\eta_l^2}{W}\sum_{i\in[n],e,j}\frac{\tilde{w}_i}{q_i c_i^3}\left(((e-1)|\mathcal{D}_i|+j)^2\|\nabla f_i(x)\|^2+\frac{j(|\mathcal{D}_i|-j)}{(|\mathcal{D}_i|-1)}\frac{1}{|\mathcal{D}_i|}\sum_{c=1}^{\mathcal{D}_i}\|\nabla f_{ic}(x)-\nabla f_i(x)\|^2\right)$$

$$\overset{(6)}{\leq}\frac{2\eta_l^2}{W}\sum_{i\in[n],e,j}\frac{\tilde{w}_i}{q_i c_i^3}\left(((e-1)|\mathcal{D}_i|+j)^2\|\nabla f_i(x)\|^2+\frac{j(|\mathcal{D}_i|-j)}{(|\mathcal{D}_i|-1)}(\sigma_i^2+P^2\|\nabla f_i(x)\|^2)\right)$$

$$\leq 2\eta_l^2\sum_{i\in[n]}\frac{|\mathcal{D}_i|^2 E_i^2}{c_i^2}\hat{w}_i\|\nabla f_i(x)\|^2+\frac{|\mathcal{D}_i+1|)}{6c_i^2}\hat{w}_i(\sigma_i^2+P_i^2\|\nabla f_i(x)\|^2)$$

$$\leq 2\eta_l^2\sum_{i\in[n]}\left(1+\frac{P_i^2}{3|\mathcal{D}_i|E_i^2}\right)\hat{w}_i\|\nabla f_i(x)\|^2+\hat{w}_i\frac{\sigma_i^2}{3|\mathcal{D}_i|E_i^2}$$

$$\overset{(5)}{\leq}4LB^2\eta_l^2(1+P^2)(\hat{f}(x)-\hat{f}^\star)+2\eta_l^2((1+P^2)G^2+\sigma^2).$$

Combining the upper bounds

$$\xi\leq 2\eta_l^2 L^2\xi+4LB^2\eta_l^2(1+P)(\hat{f}(x)-\hat{f}^\star)+2\eta_l^2((1+P^2)G^2+\sigma^2).$$

Since $2\eta_l^2 L^2\leq\frac{1}{8}$ and $4LB^2\eta_l^2(1+P^2)\leq\frac{1}{4L}$, therefore

$$3L\tilde{\eta}\xi\leq\frac{9}{10}\tilde{\eta}(\hat{f}(x)-\hat{f}^\star)+9\tilde{\eta}\eta_l^2\left((1+P^2)G^2+\sigma^2\right),$$

which concludes the proof. $\qquad\square$

Adding the statements of the above Lemmas E.3 and E.4, we get

$$\mathbf{E}\left[\|x^r-\hat{x}^\star\|^2\right]\leq\left(1-\frac{\mu\tilde{\eta}}{2}\right)\mathbf{E}\left[\|x^{r-1}-\hat{x}^\star\|^2\right]-\frac{\tilde{\eta}}{10}\mathbf{E}_{r-1}\left[\hat{f}(x^{r-1})-\hat{f}^\star\right]+2\tilde{\eta}^2 M_1 G^2+\frac{9\tilde{\eta}^3}{\eta_g^2}\left((1+P^2)G^2+\sigma^2\right)$$

$$=\left(1-\frac{\mu\tilde{\eta}}{2}\right)\mathbf{E}\left[\|x^{r-1}-\hat{x}^\star\|^2\right]-\frac{\tilde{\eta}}{10}\mathbf{E}_{r-1}\left[f(x^{r-1})-\hat{f}^\star\right]+2\tilde{\eta}^2\left(M_1 G^2+\frac{9\tilde{\eta}}{2\eta_g^2}\left((1+P^2)G^2+\sigma^2\right)\right),$$

Moving the $(f(x^{r-1})-f(x^\star))$ term and dividing both sides by $\frac{\tilde{\eta}}{10}$, we get the following bound for any $\tilde{\eta}\leq\frac{1}{(1+M_2+(P+1)B+M_1 B^2))4L}$

$$\mathbf{E}_{r-1}\left[f(x^{r-1})-f^\star\right]\leq\frac{10}{\tilde{\eta}}\left(1-\frac{\mu\tilde{\eta}}{2}\right)\mathbf{E}\left[\|x^{r-1}-x^\star\|^2\right]+20\tilde{\eta}\left(M_1 G^2+\frac{9\tilde{\eta}}{2\eta_g^2}\left((1+P^2)G^2+\sigma^2\right)\right).$$

If $\mu=0$ (weakly-convex), we can directly apply Lemma B.3. Otherwise, by averaging using weights $v_r=(1-\frac{\mu\tilde{\eta}}{2})^{1-r}$ and using the same weights to pick output $\bar{x}^R$, we can simplify the above recursive bound (see proof of Lemma B.2) to prove that for any $\tilde{\eta}$ satisfying $\frac{1}{\mu R}\leq\tilde{\eta}\leq\frac{1}{(1+M_2+(P+1)B+M_1 B^2))4L}$

$$\mathbf{E}\left[f(\bar{x}^R)\right]-f(x^\star)\leq\underbrace{10\|x^0-x^\star\|^2}_{\overset{\text{def}}{=}d}\mu\exp(-\frac{\tilde{\eta}}{2}\mu R)+\tilde{\eta}\underbrace{\left(4M_1 G^2\right)}_{\overset{\text{def}}{=}c_1}+\tilde{\eta}^2\left\{\underbrace{\frac{18}{\eta_g^2}\left((1+P^2)G^2+\sigma^2\right)}_{\overset{\text{def}}{=}c_2}\right\}$$

Now, the choice of $\tilde{\eta} = \min\left\{\frac{\log(\max(1,\mu^2 Rd/c_1))}{\mu R}, \frac{1}{(1+M_2+(P+1)B+M_1B^2))4L}\right\}$ yields the desired rate.

For the non-convex case, one first exploits the smoothness assumption (Assumption 3.2) (extra smoothness term $L$ in the first term in the convergence guarantee) and the rest of the proof follows essentially in the same steps as the provided analysis. The only difference is that distance to an optimal solution is replaced by functional difference, i.e., $\|x^0 - \hat{x}^\star\|^2 \to \hat{f}(x^0) - \hat{f}^\star$. The final convergence bound also relies on Lemma B.3. For completeness, we provide the proof below.

We adapt the same notation simplification as for the prior cases, see proof of Lemma E.3. Since $\{f_{ij}\}$ are $L$-smooth then $f$ is also $L$ smooth. Therefore,

$$
\begin{aligned}
\mathbf{E}_{r-1}\left[\hat{f}(x+\Delta)\right] &\le \hat{f}(x) + \mathbf{E}_{r-1}\left[\left\langle \nabla\hat{f}(x), \Delta\right\rangle\right] + \frac{L}{2}\mathbf{E}_{r-1}\left[\|\Delta\|^2\right] \\
&= \hat{f}(x) - \mathbf{E}_{r-1}\left[\left\langle \nabla\hat{f}(x), \frac{\tilde{\eta}}{W}\sum_{i\in[n],e,j}\frac{\tilde{w}_i}{q_i c_i}g_{i,e,j}\right\rangle\right] + \frac{L\tilde{\eta}^2}{2}\mathbf{E}_{r-1}\left[\left\|\frac{1}{W}\sum_{i\in\mathcal{S},e,j}\frac{\tilde{w}_i}{q_i^{\mathcal{S}} c_i}g_{i,e,j}\right\|^2\right] \\
&\overset{(22)}{\le} \hat{f}(x) - \frac{\tilde{\eta}}{2}\left\|\nabla\hat{f}(x)\right\|^2 + \frac{\tilde{\eta}}{2}\mathbf{E}_{r-1}\left[\left\|\frac{1}{W}\sum_{i\in[n],e,j}\frac{\tilde{w}_i}{q_i c_i}\left(g_{i,e,j} - \nabla f_{i\Pi_{i,e,j-1}}(x)\right)\right\|^2\right] \\
&\quad + \frac{L\tilde{\eta}^2}{2}\mathbf{E}_{r-1}\left[\left\|\frac{1}{W}\sum_{i\in\mathcal{S},e,j}\frac{\tilde{w}_i}{q_i^{\mathcal{S}} c_i}g_{i,e,j}\right\|^2\right] \\
&\overset{(3)+(23)}{\le} \hat{f}(x) - \frac{\tilde{\eta}}{2}\left\|\nabla\hat{f}(x)\right\|^2 + \frac{\tilde{\eta}L^2}{2}M_2\xi + \frac{L\tilde{\eta}^2}{2}\mathbf{E}_{r-1}\left[\left\|\frac{1}{W}\sum_{i\in\mathcal{S},e,j}\frac{\tilde{w}_i}{q_i^{\mathcal{S}} c_i}g_{i,e,j}\right\|^2\right].
\end{aligned}
$$

We upper-bound the last term using the bound of $\mathcal{A}_2$ in the proof of Lemma C.1 (note that this proof does not rely on the convexity assumption). Thus, we have

$$
\mathbf{E}_{r-1}\left[\hat{f}(x+\Delta)\right] \le \hat{f}(x) - \frac{\tilde{\eta}}{2}\left\|\nabla\hat{f}(x)\right\|^2 + \frac{\tilde{\eta}L^2}{2}M_2\xi + (M_1+M_2)\tilde{\eta}^2 L^3\xi + \tilde{\eta}^2 M_1 G^2 L + \tilde{\eta}^2\left(1+M_1B^2\right)L\left\|\nabla\hat{f}(x)\right\|^2.
$$

The bound on the step size $\tilde{\eta} \le \frac{1}{(1+M_2+(P+1)B+M_1B^2))4L}$ implies

$$
\mathbf{E}_{r-1}\left[f(x+\Delta)\right] \le f(x) - \frac{\tilde{\eta}}{4}\|\nabla f(x)\|^2 + \frac{3\tilde{\eta}L^2}{4}\xi + \tilde{\eta}^2 M_1 G^2 L.
$$

Next, we reuse the partial result of Lemma C.2 that does not require convexity, i.e., we replace $\hat{f}(x) - \hat{f}^\star$ with $\frac{1}{2L}\|\nabla\hat{f}(x)\|^2$. Therefore,

$$
\mathbf{E}_{r-1}\left[\hat{f}(x+\Delta)\right] \le \hat{f}(x) - \frac{\tilde{\eta}}{8}\left\|\nabla\hat{f}(x)\right\|^2 + \frac{9\tilde{\eta}^3 L}{4\eta_g^2}\left((1+P^2)G^2 + \sigma^2\right) + \tilde{\eta}^2 M_1 G^2 L.
$$

By adding $\hat{f}^\star$ to both sides, reordering, dividing by $\tilde{\eta}$ and taking full expectation, we obtain

$$
\mathbf{E}\left[\left\|\nabla\hat{f}(x^r)\right\|^2\right] \le \frac{8}{\tilde{\eta}}\left(\mathbf{E}\left[\hat{f}(x^r) - \hat{f}^\star\right] - \mathbf{E}\left[\hat{f}(x^{r+1}) - \hat{f}^\star\right]\right) + \tilde{\eta}\underbrace{8M_1 G^2 L}_{c_1} + \tilde{\eta}^2\underbrace{\frac{18L}{\eta_g^2}\left((1+P^2)G^2 + \sigma^2\right)}_{c_2}.
$$

Applying Lemma B.3 concludes the proof.

### E.4  General Variance Bound

In this section, we provide a more general version of Lemma B.4. We recall the definition of the indicator function used in the proof of the aforementioned lemma, where $1_{i \in S} = 1$ if $i \in S$ and $1_{i \in S} = 0$ otherwise and, likewise, $1_{i,j \in S} = 1$ if $i, j \in S$ and $1_{i,j \in S} = 0$ otherwise. Further, let $\mathbf{H}$ be $n \times n$ matrix, where $\mathbf{H}_{i,j} = \mathbf{E}_S \left[ \frac{q_i q_j}{q_i^S q_j^S} 1_{i,j \in S} \right]$ and $h$ is its diagonal with $h_i = \mathbf{E}_S \left[ \frac{q_i^2}{(q_i^S)^2} 1_{i \in S} \right]$. We are ready to proceed with the lemma.

**Lemma E.5.** *Let $\zeta_1, \zeta_2, \ldots, \zeta_n$ be vectors in $\mathbb{R}^d$ and $w_1, w_2, \ldots, w_n$ be non-negative real numbers such that $\sum_{i=1}^n w_i = 1$. Define $\tilde{\zeta} \stackrel{def}{=} \sum_{i=1}^n \frac{w_i}{q_i} \zeta_i$. Let $S$ be a proper sampling. If $s \in \mathbb{R}^n$ is such that*

$$\mathbf{H} - ee^\top \preceq \mathbf{Diag}(h_1 s_2, h_2 s_2, \ldots, h_n s_n), \tag{33}$$

*then*

$$\mathbf{E} \left[ \left\| \sum_{i \in S} \frac{w_i \zeta_i}{g_i^S} - \tilde{\zeta} \right\|^2 \right] \leq \sum_{i=1}^n h_i s_i \left\| \frac{w_i \zeta_i}{q_i} \right\|^2, \tag{34}$$

*where the expectation is taken over $S$ and $e$ is the vector of all ones in $\mathbb{R}^n$.*

*Proof.* Let us first compute the mean of $X \stackrel{\text{def}}{=} \sum_{i \in S} \frac{w_i \zeta_i}{q_i^S}$:

$$\mathbf{E}[X] = \mathbf{E} \left[ \sum_{i \in S} \frac{w_i \zeta_i}{q_i^S} \right] = \mathbf{E} \left[ \sum_{i=1}^n \frac{w_i \zeta_i}{q_i^S} 1_{i \in S} \right] = \sum_{i=1}^n w_i \zeta_i \mathbf{E} \left[ \frac{1}{q_i^S} 1_{i \in S} \right] = \sum_{i=1}^n \frac{w_i}{q_i} \zeta_i = \tilde{\zeta}.$$

Let $\boldsymbol{A} = [a_1, \ldots, a_n] \in \mathbb{R}^{d \times n}$, where $a_i = \frac{w_i \zeta_i}{q_i}$. We now write the variance of $X$ in a form which will be convenient to establish a bound:

$$\begin{aligned}
\mathbf{E} \left[ \|X - \mathbf{E}[X]\|^2 \right] &= \mathbf{E} \left[ \|X\|^2 \right] - \|\mathbf{E}[X]\|^2 \\
&= \mathbf{E} \left[ \left\| \sum_{i \in S} \frac{w_i \zeta_i}{q_i^S} \right\|^2 \right] - \|\tilde{\zeta}\|^2 \\
&= \mathbf{E} \left[ \sum_{i,j} \frac{w_i \zeta_i^\top}{q_i^S} \frac{w_j \zeta_j}{q_j^S} 1_{i,j \in S} \right] - \|\tilde{\zeta}\|^2 \\
&= \sum_{i,j} \mathbf{H}_{ij} \frac{w_i \zeta_i^\top}{q_i} \frac{w_j \zeta_j}{q_j} - \sum_{i,j} \frac{w_i \zeta_i^\top}{q_i} \frac{w_j \zeta_j}{q_j} \\
&= \sum_{i,j} (\mathbf{H}_{ij} - 1) a_i^\top a_j \\
&= e^\top ((\mathbf{H} - ee^\top) \circ \boldsymbol{A}^\top \boldsymbol{A}) e.
\end{aligned} \tag{35}$$

Since, by assumption, we have $\mathbf{H} - ee^\top \preceq \mathbf{Diag}(h \circ s)$, we can further bound

$$e^\top ((\mathbf{H} - ee^\top) \circ \boldsymbol{A}^\top \boldsymbol{A}) e \leq e^\top (\mathbf{Diag}(h \circ s) \circ \boldsymbol{A}^\top \boldsymbol{A}) e = \sum_{i=1}^n h_i s_i \|a_i\|^2.$$

$\square$

Table 4: Baselines.

| | Local Steps/ Epochs | W/ or W/O Replacement Gradient Sampling[a] | Pseudocode |
|---|---|---|---|
| FEDAVGRR | Epochs | W/O | Algorithm 2 |
| FEDAVGMIN | Steps | W/ | Reddi et al. (2020) Algorithm 1 with $K = K_{\min}^b$ |
| FEDAVGMEAN | Steps | W/ | Reddi et al. (2020) Algorithm 1 with $K = K_{\mathrm{mean}}^c$ |
| FEDNOVA | Epochs | W/ | Wang et al. (2020) |
| FEDNOVARR | Epochs | W/O | Algorithm 4 with FEDNOVA aggregation |
| FEDSHUFFLE | Epochs | W/O | Algorithm 1 |

[a] For real-world datasets, as it is commonly implemented in practice, all methods use without replacement sampling, i.e., random reshuffling.

[b] $K = K_{\min}$ corresponds to the minimal number of steps across clients for FEDAVGRR.

[c] $K = K_{\mathrm{mean}}$ corresponds to the average number of steps across clients for FEDAVGRR, i.e., FEDAVGMEAN and FEDAVGRR perform the same total computation.

## F   Experimental Setup and Extra Experiments

As mentioned in the previous section, we compare three algorithms – FEDAVG, FEDNOVA, and our FED-SHUFFLE. For FEDAVG, we include an extra baseline that fixes objective inconsistency by running the same number of local steps per each client. To avoid extra computational burden on clients, i.e., to avoid stragglers, we select the number of local steps to be the minimal number of steps across clients. We refer to this baseline as FEDAVGMIN. Another FEDAVG-type baseline that we compare to is FEDAVGMEAN. Similarly to FEDAVGMIN, FEDAVGMEAN runs the same number of local steps per each device with a difference that the number of steps is selected such that the total computation performed by FEDAVGMEAN is equal to FEDAVG with local epochs. We include this theoretical baseline to see the effect of heterogeneity in local steps and whether it is beneficial to run full epochs. We consider three extensions: (i) random reshuffling for FEDAVG, and FEDNOVA, since these methods were not originally analysed using biased random reshuffling but rather with unbiased stochastic gradients obtained by sampling with replacement, (ii) global momentum. Our implementation of FEDAVG and FEDNOVA follows (Reddi et al., 2020) and (Wang et al., 2020), respectively. In all of the experiments, the reported performance is average with one standard deviation over 3 runs with fixed seeds across methods for a fair comparison.

**Hyperparameters.** The server learning rate for all methods is kept at its default value of 1. For all momentum methods, we pick momentum 0.9. The rest of the parameters for the simple quadratic objective are selected based on the theoretical values for all the methods. For the other experiments (Shakespeare and CIFAR100), the local learning rate is tuned by searching over a grid $\{0.1, 0.01, 0.001\}$ and we use a standard learning schedule, where the step size is decreased by 10 at 50% and 75% of all communication rounds. We note that FEDSHUFFLE uses different local step sizes for each client $\eta_{l,i}^r = \eta_i^r/|\mathcal{D}_i|$. For a fair comparison, we take the local learning rate for FEDSHUFFLE to be the local learning rate for the client with the largest number of local steps as this follows directly from the theory. The global momentum as defined in (13) and (14) is only used in this form for the toy experiment. For the real-word experiments, we ignore the last term and we approximate $\nabla f_i(x^r) \approx \frac{1}{E|\mathcal{D}_i|\eta_{l,i}^r}\Delta_i^r$ to avoid extra communication and computation. Note that for both FEDNOVA and FEDSHUFFLE, the estimator of $\nabla f(x^r)$ is proportional to $\Delta^r$ and, therefore, the server can directly update the global momentum using only this update. For FEDAVG, this is not the case and it is closely related to the objective inconsistency issue. Despite this fact, we provide FEDAVG with the proper momentum estimator that improves its performance as it reduces the objective inconsistency. Lastly, for neural network experiments we allow each method to benefit from the weight decay $\{0, 1e-4\}$ and the gradient clipping $\{5, \infty\}$, i.e., we do a grid search over all the combinations of step sizes, weight decays and gradient clippings. For the toy experiment, the batch size is 1 and for neural nets, we select batch size to be 32.

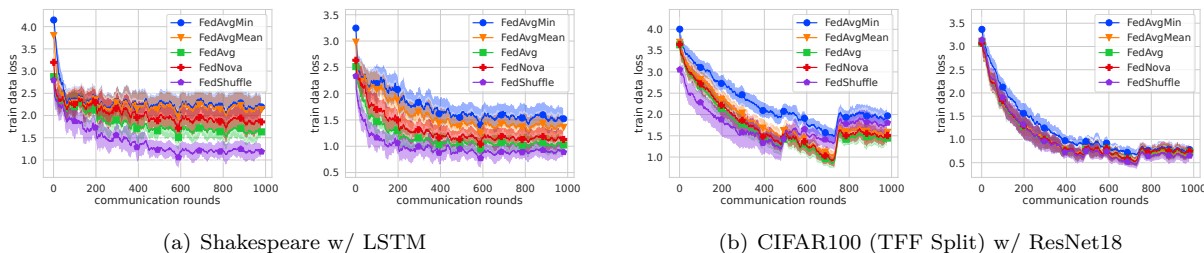

(a) Shakespeare w/ LSTM                     (b) CIFAR100 (TFF Split) w/ ResNet18

Figure 3: Comparison of the moving average with slide 20 of the train data loss on FEDAVGMIN, FEDAVG, FEDNOVA, FEDSHUFFLE on real-world datasets. Partial participation: in each round 16 client is sampled uniformly at random. All methods use random reshuffling. For Shakespeare, number of local epochs is 2 and for CIFAR100, it is 2 to 5 sampled uniformly at random at each communication round for each client. Left: Plain methods. Right: Global momentum 0.9. *Note that the displayed loss is only for the train data, and it does not include the l2 penalty; therefore, its informative value is limited.*

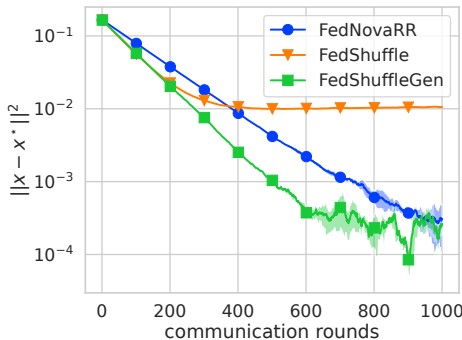

Figure 4: Quadratic objective as defined in (36). Each client runs one local epoch. A comparison of FEDNOVA with reshuffling (FEDNOVA RR) and FEDSHUFFLE and FEDSHUFFLE with FEDNOVA-like aggregation weights to remove objective inconsistency caused by unfinished local rounds (FEDSHUFFLEGEN) with full participation.

**Datasets and models.** We run experiments on three datasets with three corresponding models. First, we consider a simple quadratic objective

$$\min_{x \in \mathbb{R}^6} \frac{1}{12} \sum_{i=1}^{6} \|x - e_i\|^2, \tag{36}$$

where $e_i$ is the canonical basis vector with 1 at $i$-th coordinate and 0 elsewhere. We partition data into 3 clients, where the first client owns the first data point, the second is assigned $e_2$ and $e_3$ and the rest belongs to the third client.

For the second task, we evaluate a next character prediction model on the Shakespeare dataset (Caldas et al., 2018a). This dataset consists of 715 users (characters of Shakespeare plays), where each example corresponds to a contiguous set of lines spoken by the character in a given play. For the model, an input character is first transformed into an 8-dimensional vector using an embedding layer, and this is followed by an LSTM (Hochreiter & Schmidhuber, 1997) with two hidden layers and 512-dimensional hidden state.

Last, we consider an image classification task on the CIFAR100 dataset (Krizhevsky et al., 2009) with a ResNet18 model (He et al., 2016), where we replace batch normalization layers with group normalization following (Hsieh et al., 2020). The data is partitioned across 500 clients, where each client owns 100 data points using a hierarchical Latent Dirichlet Allocation (LDA) process (Li & McCallum, 2006). We use equivalent splits to those provided in TensorFlow Federated datasets (TFF, 2021).

Our implementation is in PyTorch (Paszke et al., 2019).

For the first experiment, we only consider performance on the train set (i.e., training loss), while for the real-world datasets we report performance on the corresponding validation datasets (validation accuracy).

**Baselines.** For the experimental evaluation, we compare three methods — FedAvg, FedNova, and our FedShuffle— with different extensions such as random reshuffling or momentum

### F.1 Hybrid approach to tackle system challenges

As discussed in Section 4.3, FEDSHUFFLEGEN allows us to run and analyze hybrid approaches of mixing step size scaling with update scaling to overcome the objective inconsistency caused by system challenges. In this example, we assume that each client does not finish the last iteration of the local training. To fix this issue, we employ FEDSHUFFLE and FEDNOVA-like aggregation that fixes objective inconsistency (we refer to this method as FEDSHUFFLEGEN since it is its special case); see (31) for the selection of weights. We use the quadratic objective as introduced above with full participation, and each client runs one local epoch. As it can be seen from Figure 4, as expected, FEDSHUFFLE suffers from objective inconsistency since some clients do not finish the predefined number of local steps. However, our FEDSHUFFLEGEN based technique fixes objective consistency caused by system challenges while providing superior performance when compared to the plain FEDNOVA method with reshuffling (FEDNOVA RR).

