# OpenReview forum: "FedShuffle:  Recipes for Better Use of Local Work in Federated Learning"
_TMLR — Accepted by TMLR_

### Review · Reviewer_T6NJ · 2022-07-30

**Summary Of Contributions:**

This paper provides detailed convergence analysis for FedShuffle, which includes several ingredients commonly used in federated learning but not rigorously analyzed, including random reshuffling, heterogeneous data, non-identical local steps, global momentum, etc.


**Broader Impact Concerns:**

The authors did not discuss the broader impact. This is okay as it is a theoretical paper, but general discussion on federated learning might be good.

**Requested Changes:**

1. Page 2, 4th paragraph: Why do we suddenly talk about random reshuffling? Could you add some motivation for this technique?
2. Page 2 before contributions: why would you say "a unified view"? Accounting for all these tricks does not sound like a unified view but more like a simple composition.
3. Typo: Page 4, 3rd paragraph: "in  Mishchenko"
4. Section 3, Assumption 3.1: maybe it's worthwhile to (re)introduce $(j, x)$.
5. Assumptions 3.3, 3.4: is it a well-known assumption in the field? If so please add relevant references.
6. Page 5 typo: $s_i = n - b/n - 1$
7. Page 5: "Finally, we note that a fixed arbitrary participation framework is only for an ease of exposition and our framework can handle non-fixed distributions with minimal adjustments in the analysis." Could you explain how the minimal adjustments can be made?
8. Page 5: what is the meaning of the vector $s$?
9. Section 4.1, eq. (9): this example is adapted from FedNova and some reference needs to be made.
10. Before Sec 4.2: this paragraph should be moved to the experimental section;
11. Sec 4.2 2nd paragraph: the derivation is not quite straightforward. The authors might want to explain the example separately and show the details;
12. Sec 4.2: "This is a very standard aggregation scheme". Any reference?
13. Question: Which algorithm do you recommend the most, FedShuffle, FedShuffleMVR or FedShuffleGen? Any theoretical comparison on the convergence rates?
14. Sec 5, before Thm 5.1: (re)explain $s_i, p_i, P_i, B$.
15. Page 9, line 1: why would $M = 0$ will full participation?
16. Page 9 line 5: how can we have $\eta_g \to \infty$? This is not practical.
17. Page 9: the terminology "importance sampling" has its special meaning. Is your method the same as the standard terminology?
18. Page 9: Before Theorem 5.2: (typo) we "refer reader to"
19. General typo: \citet vs. \citep: Page 2 (Karimireddy et al., 2020); Page 4: (Mishchenko et al, 2021), (Yun et al., 2021)
20. Sec 6.1 typo: see (36) in appendix; due to decreased amount
21. Baseline comparison in Table 2/3: "FedAvgMin" and "FedAvgMean" are from the current paper and may not be regarded as strong baselines. The authors should implement FedLin, Mime, FedRR, FedProx, VRLSGD and SCAFFOLD that are mentioned in the related work.
22. Large blanks in the appendix: Page 20; page 23.




**Strengths And Weaknesses:**

Strengths:
1. Careful theoretical analysis of important ingredients of federated learning, including heterogeneous local steps, random reshuffling and momentum variance reduction.
2. This work proposes several algorithms, FedShuffle, FedShuffleMVR and FedShuffleGen. Convergence analysis of each algorithm is given.
3. Experiments on CIFAR100 and Shakespeare, which show the improvement of FedShuffle over baseline algorithms.
4. Good literature survey of recent related work.


Weaknesses:
1. Although I appreciate the detailed convergence analysis of FedShuffle, the main algorithm seems like a direct combination of many tricks used for FL, which makes the work sounds a bit incremental. In fact, as shown in Table 1, all of the tricks have appeared in previous works, including FedNova, FedLin, Mime and FedRR. This work seems like a chowder of these various tricks.
2. Some notations are not clear for non-experts. (See the next section)
3. Some paragraphs are not clearly written and further motivations/explanations are needed. (See the next section)
4. In the experimental section, the comparison wrt the baselines is limited. More related algorithms are needed for demonstrating the effectiveness of FedShuffle.

---

> ### Author Response · Authors · 2022-08-12
> **Authors' response**
>
> We thank the reviewer for their valuable time reviewing our manuscript and for insightful suggestions. In particular, we appreciate that the reviewer acknowledges our careful theoretical analysis of several popular tricks for FL, the proposal of new methods accompanied by theoretical analysis, our experiments, and a good survey of recent related work.
>
> We carefully address all the raised concerns below. In a nutshell, we believe that they can be easily addressed mainly by adding more explanations to make the statements more transparent.
>
> Please let us know whether all your concerns were sufficiently addressed.

---

> > ### Author Response · Authors · 2022-08-12
> > **Authors' response (Weaknesses)**
> >
> > > 1. Although I appreciate the detailed convergence analysis of FedShuffle, the main algorithm seems like a direct combination of many tricks used for FL, which makes the work sounds a bit incremental. In fact, as shown in Table 1, all of the tricks have appeared in previous works, including FedNova, FedLin, Mime and FedRR. This work seems like a chowder of these various tricks.
> >
> > Thank you for this comment. The statement is indeed correct. We do not claim any novelty when it comes to individual techniques/components used in the algorithmic construction. As discussed in the paper, FedShuffle reflects the common implementation of local methods in practice. However, a rigorous understanding of its convergence properties was not available, and our work is the first to provide those guarantees, which is the main contribution of our paper; see the third bullet point in contributions. As we mentioned, the provided analysis is highly non-trivial and captures complex interactions of all these tricks. Particularly challenging steps of our analysis are incorporating random reshuffling and non-deterministic aggregation rules based on client sampling.
> > If this is unclear, we would appreciate any suggestion by the reviewer about how to improve our writing here. We would be happy to incorporate such changes in the final version of our manuscript. Our careful analysis reveals the “optimal” combination of these tricks and leads to the proposal of the FedShuffle algorithm.
> >
> > > 2. Some notations are not clear for non-experts. (See the next section)
> >
> > Thank you for this suggestion. We have incorporated your suggestions, which are reflected in the updated manuscript.
> >
> > > 3. Some paragraphs are not clearly written and further motivations/explanations are needed. (See the next section)
> > In the experimental section, the comparison wrt the baselines is limited. More related algorithms are needed for demonstrating the effectiveness of FedShuffle.
> >
> > Thank you for this suggestion. We have incorporated your suggestions, which are reflected in the updated manuscript.
> >
> > > 4. Thank you for this suggestion. We have incorporated your suggestions, which are reflected in the updated manuscript.
> >
> > We think there is a slight misunderstanding with the baselines that we compare. We believe that our considered baselines are current state-of-the-art, as shown in previous works ([FedNova](https://arxiv.org/abs/2007.07481), [Mime](https://openreview.net/forum?id=FMPuzXV1fR)). We indeed do not compare to FedLin, SCAFFOLD, or VRLSGD. Still, we do compare against stronger baselines, namely Mime, MimeMVR, and FedNova, as these were shown to lead to strictly better empirical results than the aforementioned baselines on the tasks that we consider; see [FedNova](https://arxiv.org/abs/2007.07481), [Mime](https://openreview.net/forum?id=FMPuzXV1fR). Below, we provide the exact mapping:
> >
> > - Mime $\approx$ MimeLite = FedAvg (as run in practice), FedAvgMin (Mean) (as analyzed in theory).
> > - MimeMVR \approx MimeLiteMVR = FedAvg \w Momentum (as run in practice), FedAvgMin (Mean) w/ (as analyzed in theory).
> > - FedRR = FedAvg (as run in practice), FedAvgMin (Mean) (as analyzed in theory). Note here that we also extend FedRR with momentum to obtain a stronger baseline, which was not considered in the original paper.
> > - FedNova = FedNova. Note here that we also extend FedNova with momentum and local shuffling to obtain a stronger baseline, both of which were not considered in the original paper.
> >
> > We do not consider communication compression or adaptive methods, e.g., Adam and Adagrad, as our work's primary goal and contribution is to provide a better understanding of local work in federated learning. Both communication compression and adaptive methods are orthogonal techniques to improve convergence guarantees further. Adding both of these techniques is an interesting direction for future work.

---

> > > ### Author Response · Authors · 2022-08-12
> > > **Authors' response (requested changes 1/2)**
> > >
> > > > 1. Page 2, 4th paragraph: Why do we suddenly talk about random reshuffling? Could you add some motivation for this technique?
> > >
> > > We mention this in the first sentence. The main motivation for this technique is its “variance-reducing effect.” Please let us know whether this is sufficient or if you think it requires more explanation / better reasoning.
> > >
> > > > 2. Page 2 before contributions: why would you say "a unified view"? Accounting for all these tricks does not sound like a unified view but more like a simple composition.
> > >
> > > This is correct when one talks about the implementation, but the story is entirely different when we look at the theoretical guarantees. While it is easy to combine all of these techniques in practice, theoretical understanding requires careful analysis because of non-trivial interactions among different tricks. We discuss this briefly in the contributions (“Theoretical analysis”).
> > >
> > > > 3. Typo: Page 4, 3rd paragraph: "in Mishchenko"
> > >
> > > Thank you, we fixed this.
> > >
> > > > 4. Section 3, Assumption 3.1: maybe it's worthwhile to (re)introduce $(j,x)$.
> > >
> > > Thank you, we added an extra sentence.
> > >
> > > > 5. Assumptions 3.3, 3.4: is it a well-known assumption in the field? If so, please add relevant references.
> > >
> > > Yes, these are standard assumptions. To the best of our knowledge, these are the weakest assumptions commonly considered in the literature. We added the requested references.
> > >
> > > > 6. Page 5 typo: $s_i=\frac{n−b}{n−1}$
> > >
> > > Thank you, we fixed this.
> > >
> > > > 7. Page 5: "Finally, we note that a fixed arbitrary participation framework is only for an ease of exposition and our framework can handle non-fixed distributions with minimal adjustments in the analysis." Could you explain how the minimal adjustments can be made?
> > >
> > > Assume we have access to the current distribution of the client sampling. In that case, we can reuse the analysis of the single communication round provided in the paper. The only difference would be to combine the communication rounds with different client samplings. A naive solution is to use a worst-case bound across the communication rounds. Another option would be to devise a more accurate bound by considering some weighted average.
> > >
> > > > 8. Page 5: what is the meaning of the vector $s$?
> > >
> > > One can view $s$ as a price for client sampling, i.e., client variance. Larger $s$ implies worse sampling, thus, slower convergence.
> > >
> > > > 9. Section 4.1, eq. (9): this example is adapted from FedNova, and some reference needs to be made.
> > >
> > > Thank you. This is indeed correct. We added the reference.
> > >
> > > > 10. Before Sec 4.2: this paragraph should be moved to the experimental section;
> > >
> > > We kindly disagree with this statement. We believe it is beneficial to introduce the considered alternatives to FedShuffle earlier so the reader has a better picture of the development of our method. We would be grateful if the reviewer could provide us with the reasoning to move the paragraph to the experimental section.
> > >
> > > > 11. Sec 4.2 2nd paragraph: the derivation is not quite straightforward. The authors might want to explain the example separately and show the details;
> > >
> > > Thank you for this comment. For the first client with a single data point, it is not selected with probability ⅓. It is selected with the second client with probability ⅓, and the same probability holds for being selected with the third client. Therefore, the expected contribution is
> > > $$
> > > \frac{1}{3} * 0 + \frac{1}{3} * \frac{\frac{1}{6}}{\frac{1}{6} + \frac{2}{6}} + \frac{1}{3} *  \frac{\frac{1}{6}}{\frac{1}{6} + \frac{3}{6}}  = \frac{7}{36}
> > > $$
> > >
> > > The derivation for other clients can be obtained analogously.
> > >
> > > > 12. Sec 4.2: "This is a very standard aggregation scheme." Any reference?
> > >
> > > Thank you. We added the references that use such aggregation in different domains.
> > >
> > > > 13. Question: Which algorithm do you recommend the most, FedShuffle, FedShuffleMVR, or FedShuffleGen? Any theoretical comparison on the convergence rates?
> > >
> > > We want to clarify that FedShuffleGen is instead an algorithmic framework, and it encapsulates FedShuffle as a particular case (the precise choice of parameters is discussed in the appendix, Section E.2), where FedShuffle is obtained directly from FedShuffleGen by optimizing convergence rate with the constraint of no objective inconsistency. However, it suffers from large client drift since the local updates use no global information. FedShuffleMVR addresses this issue by employing momentum variance reduction, leading to the best convergence rate, surpassing even non-local-update methods. However, despite the superior convergence rate, FedShuffleMVR may be practically more complicated to implement (at least, as precisely analyzed), and it requires additional computation due to momentum. In addition, the analysis only covers cases where only a single client contributes to the global model in each round. To sum up, we recommend using FedShuffleMVR the most, but there might be specific scenarios in which FedShuffle could be preferred.

---

> > > > ### Author Response · Authors · 2022-08-12
> > > > **Authors' response (requested changes 2/2)**
> > > >
> > > >
> > > > > 14. Sec 5, before Thm 5.1: (re)explain $s_i, p_i, P_i, B.$
> > > >
> > > > Thank you for this comment. We added a link to the assumptions to improve readability.
> > > >
> > > > > 15. Page 9, line 1: why would $M=0$ will full participation?
> > > >
> > > > The full participation admits $s_i=0$ (Lemma B.4), because there is no approximation error. Therefore, $M=0$ by definition.
> > > >
> > > > > 16. Page 9 line 5: how can we have $\eta_g \to \infty$? This is not practical.
> > > >
> > > > While taking $\eta_g \rightarrow \infty$ is not practical, it is straightforward to implement if one considers that $\eta_l \rightarrow 0$ with the same speed. Thus, $\eta_g \eta_l$ is a constant. Therefore, the case $\eta_g \rightarrow \infty$ is equivalent to running distributed gradient descent. Note that our convergence rate exactly recovers the rate of distributed gradient descent in case $\eta_g \rightarrow \infty$.
> > > >
> > > > > 17. Page 9: the terminology "importance sampling" has its special meaning. Is your method the same as the standard terminology?
> > > >
> > > > Yes, this means that we sample more informative clients more often. In our case, this would be clients with more data.
> > > >
> > > > > 18. Page 9: Before Theorem 5.2: (typo) we "refer reader to"
> > > >
> > > > Thank you, we fixed this.
> > > >
> > > > > 19. General typo: \citet vs. \citep: Page 2 (Karimireddy et al., 2020); Page 4: (Mishchenko et al, 2021), (Yun et al., 2021)
> > > >
> > > > Thank you, we fixed this.
> > > >
> > > > > 20. Sec 6.1 typo: see (36) in appendix; due to decreased amount
> > > >
> > > > Thank you, we fixed this.
> > > >
> > > > > 21. Baseline comparison in Table 2/3: "FedAvgMin" and "FedAvgMean" are from the current paper and may not be regarded as strong baselines. The authors should implement FedLin, Mime, FedRR, FedProx, VRLSGD and SCAFFOLD that are mentioned in the related work.
> > > >
> > > > Thank you for this comment. Please see our response to weakness 4 that you mentioned.
> > > >
> > > > > 22. Large blanks in the appendix: Page 20; page 23.
> > > >
> > > >  Thank you, we fixed this.

---

### Review · Reviewer_y9tc · 2022-08-05

**Summary Of Contributions:**

The paper analyzes the convergence of federated learning for heterogeneous local data, in particular when reshuffling local data in each epoch and rescaling local learning rates by dataset size. Moreover, the paper proposes to use momentum and variance reduction in local updates to improve convergence. It then derives a general form of these approaches that also is a generalization of classical FedAvg and FedNOVA. It provides convergence rates for these approaches, as well as the general version.

**Broader Impact Concerns:**

I do not have concerns on the ethical implications of this work and thus do not see a requirement for a broader impact statement.

**Requested Changes:**

Critical:
- Clarify the issues of existing methods and how this paper overcomes them, in particular wrt. to their original description. Claiming that (i) FedAvg would not perform reshuffling and that this is a major issue, (ii) choosing a fixed number of updates between communication rounds is hard, and (iii) communicating local dataset sizes is problematic without substantiating these claims runs the risk of being a strawman argument.
- Clarify whether the convergence analysis assumes reshuffling or for drawing gradients iid. The proof in the appendix even does not mention reshuffling, but rather seems to analyse the convergence of FedAvg.
- relate convergence results to existing ones: explain the differences in assumptions and proof techniques and differences in the rates to existing convergence rate results. This includes discussing the novelty over the proof in Karimireddy et al., 2019b.
- analyze the impact of learning rate rescaling on model quality in deep learning
- please discuss the assumption of gradient similarity / Hessian similarity to the non-iid setting. Since this work is mainly motivated by the problems of FedAvg under heterogeneous local data, assuming gradient similarity seems counter-intuitive.

Non-critical:
- in the example computation in 4.2, I got $\frac{10}{36}$ for $i=1$ instead of  $\frac{7}{36}$, but I might have made a mistake. Please provide the computation, e.g., in the appendix.



**Strengths And Weaknesses:**

Strength:
- The paper presents a generalization of FedAvg and FedNova and FedAvg with momentum variance reduction, including convergence results for this general algorithm.
- The paper presents a range of convergence rate results.

Weaknesses:
- The main motivation for FedShuffle is that FedAvg would sample data with replacement and that reshuffling is beneficial. Yet, it seems the convergence analysis in this paper (appendix C) assumes gradients are drawn iid.
- How can the objective function of FedAvg with drawing data with replacement be wrong, when there are convergence results for this setup (e.g., [1], or [2] with a similar gradient similarity assumption)?
- The claim that FedAvg does not perform reshuffling is false. In the original paper, local data is partitioned into batches in each epoch (cf. Algo 1 in [3]) and reshuffling is a common practice in most implementations.
- the authors claim that fixing a number of update steps or epochs between communication rounds is hard, mainly because determining it requires sending local dataset sizes, which might infringe privacy. This claim seems unsubstantiated, since standard FedAvg requires sending local dataset sizes to normalize updates, anyways. Moreover, many practical implementations (including the original FedAvg) perform a fixed number of updates between communication rounds.
- The scaling of local learning rates is unnecessary if local dataset sizes are roughly equal, so their main use should be when local dataset sizes vary greatly. In this scenario, choosing a normal learning rate means that the effective local learning rate (i.e., chosen learning rate divided by local dataset size) is miniscule for clients with larger datasets. Choosing a large learning rate instead (so that for clients with large local datasets the effective learning rate is normal) means that for clients with small local datasets the learning rates will be enormous. Since learning rates in non-convex optimization - in particular deep learning - impact the minima that are found, this could deteriorate the model quality substantially (even if it does not impact the convergence rate).

[1] Li, Xiang, et al. "On the Convergence of FedAvg on Non-IID Data." International Conference on Learning Representations. 2020.

[2] Yu, Hao, Sen Yang, and Shenghuo Zhu. "Parallel restarted SGD with faster convergence and less communication: Demystifying why model averaging works for deep learning." Proceedings of the AAAI Conference on Artificial Intelligence. Vol. 33. No. 01. 2019.

[3] McMahan, Brendan, et al. "Communication-efficient learning of deep networks from decentralized data." Artificial intelligence and statistics. PMLR, 2017.

---

> ### Author Response · Authors · 2022-08-12
> **Authors' response**
>
> We would like to thank the reviewer for their valuable time spent reviewing our manuscript and for insightful comments and suggestions. We appreciate that the reviewer recognized our approach's generality and provided convergence guarantees. Below, we address all your concerns. We believe we have provided a detailed explanation that sufficiently addresses all reviewer's concerns. Please let us know whether you are satisfied with our response. We are happy to elaborate more if needed.

---

> > ### Author Response · Authors · 2022-08-12
> > **Authors' response (Weaknesses)**
> >
> > > The main motivation for FedShuffle is that FedAvg would sample data with replacement and that reshuffling is beneficial. Yet, it seems the convergence analysis in this paper (appendix C) assumes gradients are drawn iid.
> >
> > We would kindly disagree with this statement. Our analysis is for sampling with random reshuffling, please see line 7 in Algorithm 3, and corresponding notion $\Pi^r_{i,e,j}$, which represents $j$-th sample on $i$-th client during $e$-th local epoch at $r$-th communication round obtained by random reshuffling of local data. This means that for each local epoch $e$, we see all the local samples with probability $1$.
> >
> > > How can the objective function of FedAvg with drawing data with replacement be wrong, when there are convergence results for this setup (e.g., [1], or [2] with a similar gradient similarity assumption)?
> >
> > Thank you for this important comment. Our results do not contradict the results obtained by  [1] and [2]. The issue that we tackle is a subtle difference in how the algorithms are analyzed and run in practice. In theory, the researchers assume that each client runs a predefined number of local steps. In each local step, a local stochastic gradient is sampled using with-replacement sampling (iid). However, in practice, developers run local epochs and use random reshuffling. We discuss this in the fourth paragraph of the introduction section. This subtle difference can cause objective inconsistency (algorithms converge to the solution of a slightly different problem, see (31) in Appendix E.1) as initially observed by Wang et al. 2020 (FedNova).
> >
> > > The claim that FedAvg does not perform reshuffling is false. In the original paper, local data is partitioned into batches in each epoch (cf. Algo 1 in [3]) and reshuffling is a common practice in most implementations.
> >
> > Thank you for raising this issue. We do realize and also acknowledge (in the fourth paragraph of the introduction section) that reshuffling is a common practice in virtually all implementations. The gap that we address in this work is that despite shuffling being common in real-world deployments, it is often overlooked in the theoretical analysis. To the best of our knowledge, our work is the first to provide rigorous theoretical analysis in the most similar setup to what practitioners would implement.
> >
> > > The authors claim that fixing a number of update steps or epochs between communication rounds is hard, mainly because determining it requires sending local dataset sizes, which might infringe privacy. This claim seems unsubstantiated since standard FedAvg requires sending local dataset sizes to normalize updates, anyways. Moreover, many practical implementations (including the original FedAvg) perform a fixed number of updates between communication rounds.
> >
> > Thank you for this important comment. We wanted to point out that this might be a problem (“hard”) if clients are not willing to reveal their dataset size. We do not claim this is hard per se. We only claim that a privacy leak might be connected with this procedure, i.e., clients reveal their local dataset size. Despite this, we note that FedShuffle surpasses all the baselines, even our proposed alternatives, i.e., FedAvgMean and FedAvgMin.
> >
> > > The scaling of local learning rates is unnecessary if local dataset sizes are roughly equal, so their main use should be when local dataset sizes vary greatly. In this scenario, choosing a normal learning rate means that the effective local learning rate (i.e., chosen learning rate divided by local dataset size) is miniscule for clients with larger datasets. Choosing a large learning rate instead (so that for clients with large local datasets the effective learning rate is normal) means that for clients with small local datasets the learning rates will be enormous. Since learning rates in non-convex optimization - in particular deep learning - impact the minima that are found, this could deteriorate the model quality substantially (even if it does not impact the convergence rate).
> >
> > Thank you for this insightful comment. We note that in our experiments, we do experiment with different datasets (balanced--CIFAR100, unbalanced--Shakespeare), and for both datasets, we see that FedShuffle improves upon the baselines. We want to point out that for FedShuffle (or generally in FL), local models are not fully trained locally but are periodically averaged. Based on our experiments, such training seems to have no adverse effect on the quality of the obtained solution, and, in addition, FedShuffle enjoys faster convergence. However, we think looking at this phenomenon more in-depth might be an interesting future direction. Thank you for the suggestion.

---

> > > ### Author Response · Authors · 2022-08-12
> > > **Authors's response (Requested changes)**
> > >
> > > > Clarify the issues of existing methods and how this paper overcomes them, in particular wrt. to their original description. Claiming that (i) FedAvg would not perform reshuffling and that this is a major issue, (ii) choosing a fixed number of updates between communication rounds is hard, and (iii) communicating local dataset sizes is problematic without substantiating these claims runs the risk of being a strawman argument.
> > >
> > > We think there is a slight misunderstanding about our claims. We believe these are now well clarified in our response to the weaknesses you mentioned. Please let us know whether we have addressed all your concerns.
> > >
> > > > Clarify whether the convergence analysis assumes reshuffling or for drawing gradients iid. The proof in the appendix even does not mention reshuffling, but rather seems to analyze the convergence of FedAvg.
> > >
> > > We think there is a slight misunderstanding about iid sampling. We believe that this is now well clarified in our response to the first weakness.
> > >
> > > > Relate convergence results to existing ones: explain the differences in assumptions and proof techniques and differences in the rates to existing convergence rate results. This includes discussing the novelty of the proof in Karimireddy et al., 2019b.
> > >
> > > Regarding comparison to existing convergence results, we provide this below each theorem. Please let us know if we missed something. We are happy to add this to the final version of our manuscript.
> > >
> > > Regarding the novelty compared to the proof of Karimireddy et al., 2019b, although the high-level scheme follows similar steps, we include several non-trivial extensions that require careful novel analysis, e.g., random reshuffling, arbitrary sampling, and non-deterministic aggregation rules. In addition, our results are tight in the special cases, e.g., we recover the rate of distributed gradient descent with partial participation for $\eta_g \rightarrow \infty$ and linear speed with respect to the expected cohort size $b$ for client sampling probabilities proportional to local datasets sizes, which is also a non-trivial task considering an extra layer of complexity. We summarize this in the contributions--theoretical analysis. Please let us know whether more explanation is needed.
> > >
> > > > Analyze the impact of learning rate rescaling on model quality in deep learning
> > >
> > > As discussed in our response to the mentioned weakness, the FedShuffle training scheme seems to have no adverse effect on the quality of the obtained solution. We believe this is mainly due to the difference that local models are not fully trained locally but are periodically averaged. We again thank the reviewer for this valuable suggestion; we see this as an excellent direction to pursue future research.
> > >
> > > > Please discuss the assumption of gradient similarity / Hessian similarity to the non-iid setting. Since this work is mainly motivated by the problems of FedAvg under heterogeneous local data, assuming gradient similarity seems counter-intuitive.
> > >
> > > We use standard assumptions considered in the literature (Karimireddy et al., 2019b [Scaffold]; 2020 [Mime]; Wang et al., 2020 [FedNova]; Mitra et al., 2021 [FedLin]). We have added the reference to other works that use similar assumptions. To the best of our knowledge, our considered assumptions are the least restrictive among the ones standardly used in the literature. We note that assuming gradient similarity is standard practice to show convergence of local methods. This is not a very restrictive assumption since $G$, and $B$ can be large numbers if data are very heterogeneous. We do not assume clients to have identical data; we only consider a particular weak notion of similarity. We want to point out that bearing some similarity is very beneficial. For instance, in the case of hessian similarity, this allows us to show that if clients are similar, they can run many local steps as one can show significant improvements in communication complexity; see Theorem 5.2.
> > >
> > > > Non-critical:
> > > > In the example computation in 4.2, I got $\frac{10}{36}$ for $i=1$ instead of $\frac{7}{36}$, but I might have made a mistake. Please provide the computation, e.g., in the appendix.
> > >
> > > Thank you for spotting this. There was a typo in the formula, which we corrected in the updated manuscript. For the first client with a single data point, it is not selected with probability $\frac{1}{3}$. It is selected with the second client with probability $\frac{1}{3}$, and the same probability holds for being selected with the third client. Therefore, the expected contribution is
> > > $$
> > > \frac{1}{3} * 0 + \frac{1}{3} * \frac{\frac{1}{6}}{\frac{1}{6} + \frac{2}{6}} + \frac{1}{3} *  \frac{\frac{1}{6}}{\frac{1}{6} + \frac{3}{6}}  = \frac{7}{36}
> > > $$
> > >
> > > The derivation for other clients can be obtained analogously.

---

### Review · Reviewer_5ooG · 2022-08-08

**Summary Of Contributions:**

The paper studies using random reshuffling in federated learning algorithms, along with other techniques coping with data heterogeneity and client sampling. A general algorithmic framework is proposed and convergence analyses are provided. Some new algorithms using random reshuffling are proposed and experiments show these algorithms have good performance.

**Broader Impact Concerns:**

No concerns.

**Requested Changes:**

Current version looks good to me. A  few suggestions are 1) adding more insights discussion on random reshuffling since this is the main difference that separate this work with existing works 2) add more experiments comparing algorithms with and without random reshuffling to provide readers an idea on how much random reshuffling could help.



**Strengths And Weaknesses:**

Strengths:
1. Very comprehensive analyses are provided for the framework and the algorithms. These analyses could be of interest for future research.
2. Random reshuffling is commonly used in practice while less attention is spent on it in federated learning research. This work fills the gap in the area.

Weaknesses:
1. The new algorithm is an ensemble of commonly used techniques and is a straight forward design, I assume some practitioners might have been already using it. Thus the algorithm design is less interesting itself.

---

> ### Author Response · Authors · 2022-08-12
> **Authors' response**
>
> We would like to thank the reviewer for a very positive evaluation of our work and the valuable time spent reviewing our paper. We particularly appreciate that the reviewer acknowledges our comprehensive convergence analysis and the importance of our research focusing on the role of random reshuffling in the federated learning framework.
>
>
> > The new algorithm is an ensemble of commonly used techniques and is a straightforward design, I assume some practitioners might have been already using it. Thus the algorithm design is less interesting itself.
>
> Thank you for this comment. The statement is indeed correct. We do not claim any novelty when it comes to algorithmic construction. As discussed in the paper, FedShuffle reflects the common implementation of local methods in practice. However, a rigorous understanding of its convergence properties was not available, and our work is the first to provide those guarantees, which is the main contribution of our paper; see the third bullet point in contributions. As we mentioned, the provided analysis is highly non-trivial and captures complex interactions of all these tricks. Particularly challenging steps of our analysis are incorporating random reshuffling and non-deterministic aggregation rules based on client sampling.
>
>
> > 1) adding more insights discussion on random reshuffling since this is the main difference that separate this work with existing works
>
> Thank you for this valuable comment. We include a detailed discussion on random reshuffling in the related work section. We have added an extra reference to the introduction part to make this clear and more accessible for the reader.
>
> > 2) add more experiments comparing algorithms with and without random reshuffling to provide readers an idea on how much random reshuffling could help.
>
> Thank you for this suggestion. We show that using random reshuffling leads to significant improvement in Section 6.1 to give readers an idea about the benefits of random reshuffling. For Section 6.2, we focus on showing that FedShuffle is the best performing method for using local work in federated learning. We do not focus on directly showing the benefits of random reshuffling for deep learning experiments, as this was consistently demonstrated in previous works; please see the experimental setup in virtually all the methods we compare in our paper for evidence of the superiority of random reshuffling. We note this is consistent with what the researchers observed for centralized learning. Please let us know whether it is sufficient to address your suggestion this way.

---

### Decision · Action_Editors · 2022-09-20

**Recommendation:** Accept with minor revision

**Comment:**

The authors provide a unified theoretical analysis of a few implementation practices in federated learning (FL), including random shuffling (widely used but rarely analyzed until lately), client sampling, adaptive step size, momentum and variance reduction. While this work built on some results from previous works, I believe its analysis on random shuffling and more general subsampling in FL is timely and interesting. The authors also carefully compared to other recent baselines and showed some improvements. A few confusions were clarified during the discussion, thanks to the authors' efforts. As the authors acknowledged, the proposed algorithms are perhaps not too surprising from a practical perspective, but the simplified and unified analysis (which is the focus here) may still prove useful and inspiring to researchers in FL. Overall, we conclude that the presented results are great additions to FL.

Please consider making the following changes in the final version:
- Abstract in the submitted pdf differs from the one on openreview.
- The analysis of the example in Section 4.1 is deferred to Appendix E but it is not clear where this analysis can be found. If the argument is not long, I'd suggest moving the details to the main body. Comparing to $E=\infty$ on this example would also be interesting. $N$ in Eq (9) should be $n$?
- Is it possible to measure Hessian similarity during the iterations? This empirical quantity might provide more insights on the experimental results.
- As the reviewers suggested, it may be beneficial to explicitly point out the novelties in the proof against existing work. The statement in the contribution list is a bit too high level while the appendix (where proofs are) did not highlight the key parts.

---

> ### Author Response · Authors · 2022-09-25
> **Thank you for the positive feedback**
>
> Dear AE,
>
> Thank you for recommending the acceptance of our paper for publication in TMLR. We have uploaded the camera-ready version and included the changes as requested.
>
> Below, we address the AE’s comments that are not directly addressed in the manuscript.
>
> > Comparing $E \to \infty$ on this example would also be interesting.
>
> Thank you for this suggestion. In the provided example, the sub-optimal solution $\tilde{x}$ to which FedAvg converges doesn’t depend on $E$ as long as each client runs the same amount of epochs $E$. Therefore, taking $E$ very large does not affect anything.
> > Is it possible to measure Hessian similarity during the iterations? This empirical quantity might provide more insights into the experimental results.
>
> Thank you for this great suggestion. It could be possible to measure hessian similarity using the fact that hessian similarity is equivalent to the smoothness of $f_i(x) - f(x)$. Therefore, we could use an update from the client $i$ as an approximation to the local gradient, and the averaged update as an approximation to the full gradient, and estimate Hessian similarity $\delta$ as follows.
> $$
> \delta \approx \frac{\|\nabla f_i(x) - \nabla f(x) - \nabla f_i(x^-) + \nabla f(x^-)\|^2}{\|x - x^-\|^2},
> $$
> Where $x$ and $x^-$ are the current and previous global iterations, respectively. This is an exciting direction to investigate in future work.
>
> Best wishes,
>
> Authors